# Evolutionary footprints of a cold relic in a rapidly warming world

Eva Wolf[1]*, Emmanuel Gaquerel[1†], Mathias Scharmann[2], Levi Yant[3], Marcus A Koch[1]*

[1]Centre for Organismal Studies, University of Heidelberg, Heidelberg, Germany; [2]Department of Ecology and Evolution, University of Lausanne, Lausanne, Switzerland; [3]Future Food Beacon and School of Life Sciences, the University of Nottingham, Nottingham, United Kingdom

*For correspondence:
eva.wolf@cos.uni-heidelberg.
de (EW);
marcus.koch@cos.uni-
heidelberg.de (MAK)

Present address: †Institut de
Biologie Moléculaire des Plantes
CNRS UPR2357, Université de
Strasbourg, Strasbourg, France

Competing interest: The authors
declare that no competing
interests exist.

Reviewing Editor: Daniel J
Kliebenstein, University of
California, Davis, United States

**Abstract** With accelerating global warming, understanding the evolutionary dynamics of plant adaptation to environmental change is increasingly urgent. Here, we reveal the enigmatic history of the genus *Cochlearia* (Brassicaceae), a Pleistocene relic that originated from a drought-adapted Mediterranean sister genus during the Miocene. *Cochlearia* rapidly diversified and adapted to circum-Arctic regions and other cold-characterized habitat types during the Pleistocene. This sudden change in ecological preferences was accompanied by a highly complex, reticulate polyploid evolution, which was apparently triggered by the impact of repeated Pleistocene glaciation cycles. Our results illustrate that two early diversified Arctic-alpine diploid gene pools contributed differently to the evolution of this young polyploid genus now captured in a cold-adapted niche. Metabolomics revealed central carbon metabolism responses to cold in diverse species and ecotypes, likely due to continuous connections to cold habitats that may have facilitated widespread adaptation to alpine and subalpine habitats, and which we speculate were coopted from existing drought adaptations. Given the growing scientific interest in the adaptive evolution of temperature-related traits, our results provide much-needed taxonomic and phylogenomic resolution of a model system as well as first insights into the origins of its adaptation to cold.

## Editor's evaluation

This work has the potential to be of broad interest to scientists seeking to understand the evolutionary dynamics of plants during past periods of rapid climate change. Specifically, within the target genus of *Cochlearia*, the results indicate increased rates of speciation and diversification in response to pronounced glacial cycles. Future work to establish more direct mechanistic links between the results and conclusions will improve our understanding of adaptation and speciation.

## Introduction

Vast spatiotemporal variation across natural environments subjects all organisms to abiotic stressors (*Gienapp et al., 2008*). Dynamic shifts in these stressors lead to migration, adaptation, or extinction (*Aitken et al., 2008*). Thus, the current acceleration of global warming and climate volatility demands a better understanding of evolutionary dynamics resulting from climate change (*Root et al., 2003*; *Thomas et al., 2004*; *Jump and Peñuelas, 2005*; *Visser, 2008*; *Franks et al., 2014*). Further, there is a strong economic rationale for understanding the consequences of environmental change on plants, which typically lack the option of rapidly migrating away from changing conditions (*Xoconostle-Cazares et al., 2010*; *Olsen and Wendel, 2013*). An especially powerful natural laboratory for the study of climate change adaptation is represented by the recurrent cycles of glaciation and

deglaciation during the Pleistocene. Thus, looking backwards in time by investigating the evolutionary footprints of this epoch can provide valuable insight for our understanding of adaptive evolution.

The genus *Cochlearia* L. represents a promising study system for the evolutionary genomics of adaptation not only because of proximity to *Arabidopsis* and other Brassicaceae models, but also because of distinctive ecotypic traits which evolved within a short time span (*Koch et al., 1996*; *Koch et al., 1998*; *Koch et al., 1999*; *Koch, 2012*). Among these are adaptations to extreme bedrock types (dolomite versus siliceous), heavy metal-rich soils, diverse salt habitats, high alpine regions, and life cycle variation. This diversity is accompanied by a remarkably dynamic cytogenetic evolution within the genus. Two base chromosome numbers exist (n=6 and n=7) and out of the 20 accepted taxa, two-thirds are neopolyploids, ranging from tetraploids to octoploids (previous phylogenetic hypotheses are given in *Appendix 1—figure 1*; see *Supplementary file 1* and Appendix 1 for details). The connecting element between the various cytotypes and ecotypes is the cold character of the diverse habitat types standing in sharp contrast to the preferences of the sole outgroup sister genus *Ionopsidium*, which occurs only in arid Mediterranean habitats (*Koch, 2012*). These two genera constitute the monophyletic tribe Cochlearieae with a stem group age of approx. 18.9 million years ago (*Walden et al., 2020*) and which forms with various other tribes from Brassicaceae the rapidly emerging evolutionary lineage II with highest net diversification rates 16–23 million years ago (*Walden et al., 2020*). In total, the genus *Cochlearia* comprises 16 accepted species and 4 subspecies (*Kiefer et al., 2014*, *Supplementary file 1*).

While on species-level it has been shown in *Arabidopsis thaliana* that drought- and temperature-adaptive genetic variants are shared among Mediterranean and Nordic regions (*Exposito-Alonso et al., 2017*), the separation of *Cochlearia* from *Ionopsidium* is much deeper, dating to the mid-Miocene (*Koch, 2012*). However, the formation of the genus *Cochlearia* as we see it today first started much more recently, during the middle (0.77–0.13 mya) and late (0.13–0.012 mya) Pleistocene (*Koch, 2012*). This long lag between the divergence of the sole outgroup *Ionopsidium* and *Cochlearia* raises the hypothesis of a long-lasting footprint of drought adaptation in *Cochlearia*. The strong association with cold habitats shown by almost all *Cochlearia* species may therefore be interpreted as a cold preference that was acquired rapidly in adaptation to the intense climatic fluctuations which characterized this epoch.

There is a growing interest in the genus *Cochlearia* from diverse fields (*Reeves, 2019*; *Brock et al., 2006*; *Dauvergne et al., 2006*; *Brandrud et al., 2017*; *Mandáková et al., 2017*; *Nawaz et al., 2017*; *Bray et al., 2020*), but the evolutionary history of the genus has been highly recalcitrant. Thus it is still unknown how the genus managed the rapid transition from Mediterranean to circum-Arctic or high-alpine habitat types in combination with a highly dynamic cytogenetic evolution. Here, we overcome the first obstacle, presenting the first genus-wide picture, using comprehensive cytogenetic data and highly resolving phylogenomic analyses, complemented by insights into the *Cochlearia* metabolome response to cold. Herein the metabolome is primarily used as a complex phenotype, which might characterize potentially different bioclimatically defined biomes or species distribution ranges. However, since it has been shown for *A. thaliana* that the cold metabolome can reflect continental-scale biogeographically defined clines along temperature gradients (*Pritchard et al., 2000*; *Weiszmann et al., 2020*), it can be assumed that on evolutionary scales past footprints of diversification might be detectable, and that in principle genomic analyses (e.g., GWAS) may allow to identify candidate genes involved in pathway regulation and metabolic reaction plasticity. Our phylogenetic and cytogenetic analysis uncovers a recurrent boosting of speciation by glaciation cycles in this cytotypically very diverse genus and indicates that, despite clear challenges brought by global warming, the genus survives evolutionarily while, we speculate, rescuing its species diversity with reticulate and polyploid evolution.

## Results

### Cytogenetic analyses show geographic structuring and parallel evolutionary trends toward shrinking haploid genomes in higher polyploids

In order to first resolve its cytogenetic evolution, we generated a comprehensive survey of 575 georeferenced chromosome counts and/or genome sizes across the *Cochlearia* genus (*Supplementary file*

2) based on our novel cytogenetic data (*Supplementary file 3*) and a review of published literature over the last century (*Supplementary file 4*). This survey revealed a clear continental-scale geographical partitioning of diploid cytotypes (2n=12 and 2n=14; *Figure 1a*).

We observed highly dynamic genome compositions throughout the genus, with widespread aneuploidies (aberrations from typical species-specific chromosome numbers) and DNA content variation. Aneuploidies are frequently found in polyploid *Cochlearia* taxa, especially along the coasts (*Figure 1b*), but they are only rarely spotted in diploids and therefore are nearly absent from Arctic regions, where only diploids are observed (*Appendix 1—figure 2*). In order to further investigate the cytogenetic dynamics within *Cochlearia*, we analyzed the relationships of (1) chromosome number versus genome size and (2) chromosome number versus DNA content per chromosome via rank correlation tests (*Supplementary file 5*) and, if normality of data was given, via linear regression analyses (*Figure 1d*, *Appendix 1—figure 3*). Both analyses revealed that (1) the high frequency of polyploidization events is accompanied by increasing genome sizes (*Appendix 1—figure 4*), while there is (2) a slight but significant negative correlation of chromosome size with increasing chromosome numbers (*Figure 1c and d*). This trend was independent of inclusion or exclusion of the annual species *C. danica* and the short-lived Arctic diploids. These taxa were treated as putative outliers because a relationship between lower genome size and annuality was shown as a significant trend for the Brassicaceae as a whole (*Hohmann et al., 2015*).

## Organellar phylogenies provide evidence of recurrent glacial speciation boosting

We next assessed spatiotemporal patterns of genetic variation first using cytoplasmic and maternally inherited genomes. Extensive past hybridization and reticulate evolution of polyploid taxa results in complex evolutionary scenarios, which are often not well resolved by strict phylogenetic reconstruction using nuclear data. Therefore, conclusions herein are restricted mostly to diploid taxa and respective gene pools. To provide a highly resolved organellar phylogeny, we generated genome sequence data for 65 *Cochlearia* accessions, representing all accepted *Cochlearia* species, as well as three species from the sister genus *Ionopsidium* (*Appendix 1—figure 5* and *Supplementary file 6*). Using these data, complete plastid genomes were assembled de novo for all samples. A maximum likelihood (ML) analysis using RAxML based on our whole chloroplast genome alignment (122,798 bp, excluding one copy of the inverted repeat) covering a total of 5292 SNPs (1003 within *Cochlearia*) revealed six well-supported major lineages within the genus (*Appendix 1—figure 6*, congruent to lineages as illustrated in BEAST chronogram, *Figure 2*). A radiation of plastome diversity was indicated by the existence of several polytomies, despite the generally high resolution of the ML tree. To test if the phylogenetic scenario as revealed by plastid genome analyses was also supported by the mitochondrial genome, we generated a mitochondrial ML phylogeny based on a combination of de novo assembly and referenced-based mapping (*Appendix 1—figure 7*). The two maternal phylogenies are largely consistent, and after collapsing all branches below a bootstrap support of 95%, no incongruences remained (illustrated by a tanglegram in *Appendix 1—figure 8*).

Divergence time estimates generated with BEAST based on our whole plastid genome alignment revealed diversification bursts that closely coincide with high glacial periods. We used two secondary calibration points (*Ionopsidium*/*Cochlearia* split: 10.81 mya; *Cochlearia* crown age: 0.71 mya) taken from a large-scale age estimation analysis performed by *Hohmann et al., 2015* which included five *Cochlearia* samples and one *Ionopsidium* sample that are also included in the present study. The revealed tree topology (*Figure 2*; full tree given in *Appendix 1—figure 9* and *Appendix 1—figure 10*) is congruent with the topology of the ML tree. In accordance with *Hohmann et al., 2015*, our BEAST analysis shows a diversification of the entire genus within the last ~660 kya, after a long period of evolutionary stasis and zero net diversification following a deep split from the genus *Ionopsidium* ~9.25 mya (see also *Koch, 2012*), and in concert with the beginning of the Pleistocene's major climatic fluctuations, which are dated to 700 kya (*Webb and Bartlein, 1992*; *Comes and Kadereit, 1998*). Thus, diversification times in the six major chloroplast lineages as revealed via both ML and BEAST analysis closely coincide with high glacial periods (*Petit et al., 1999*; see timeline in *Figure 2*; *Augustin et al., 2004*).

The most basal *Cochlearia* chloroplast and mitochondrial haplotypes (yellow lineage – Arctic I) were found in Eastern Canadian *C. tridactylites*, a species of unknown ploidy (Appendix 2), in a region

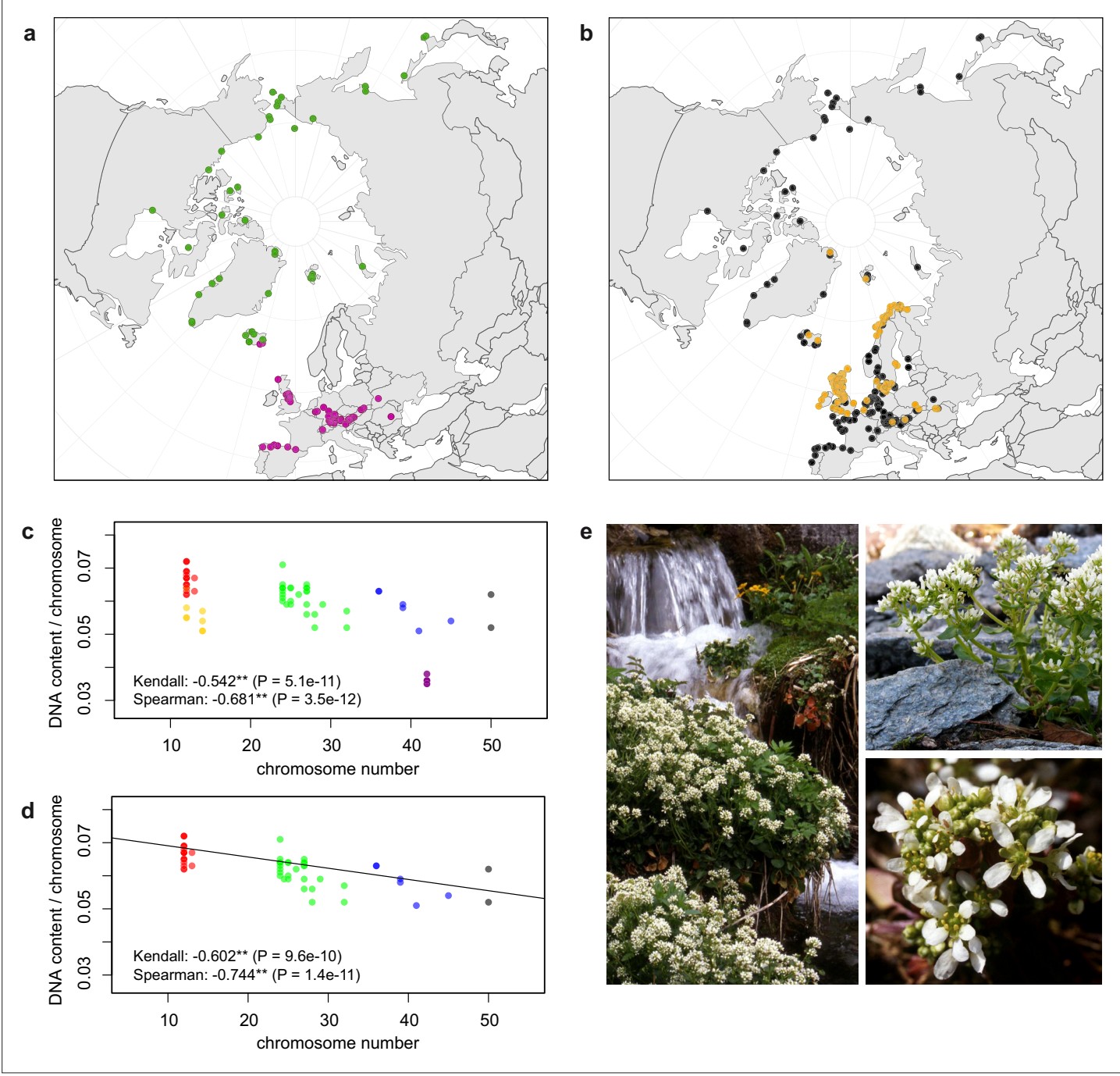

**Figure 1.** Distribution and cytogenomic flexibility of *Cochlearia*. (**a**) Geographic distribution of chromosome counts for diploid *Cochlearia* accessions (n=169; *Supplementary file 1* and *Figure 1—source data 1*), showing a clear separation of 2n=12 (European, green) and 2n=14 (Arctic, pink). (**b**) Geographic distribution of aneuploidies (orange, n=138) and euploidies (black, n=376) in diploid and polyploid *Cochlearia* (n=514; *Figure 1—source data 2*). (**c**) Measured DNA content per chromosome (given in picograms; *Figure 1—source data 3*) relative to respective total chromosome numbers (red [21 counts]: 2n=2x [non-Arctic], yellow [nine counts]: 2n=2x [Arctic], green [30 counts]: 2n=4x, blue [six counts]: 2n=6x [excluding *C. danica*], purple [10 counts]: 2n=6x [*C. danica*], dark grey [two counts]: 2n=8x) showing a significant decline of genome size per chromosome with increasing total chromosome numbers as revealed by Kendall and Spearman rank correlation analyses (78 individuals from 38 accessions representing 14 taxa analyzed in total) (data are not distributed normally and linear regression has not been performed). (**d**) Measured DNA content per chromosome (given in picograms; *Figure 1—source data 4*) relative to respective total chromosome numbers excluding Arctic diploids (yellow) and *C. danica* (purple) as putative outliers (59 individuals, 29 accessions, 11 taxa analyzed in total), showing a significant decline with increasing chromosome numbers via both rank correlation analyses and linear regression analysis (data are normally distributed and linear regression is significant with p=$7.11e^{-10}$; R²=0.48; QQ-plot given with *Appendix 1—figure 3*). (**e**) Images of three *Cochlearia* species (left: *C. pyrenaica* [2n=2x=12], top right: *C. tatrae* [2n=6x=42], bottom

*Figure 1 continued on next page*

Figure 1 continued

right: *C. anglica* [2n=8x=48]) (data are normally distributed and linear regression is significant with p=7.11e⁻10; R²=0.48).

The online version of this article includes the following source data for figure 1:

**Source data 1.** Coordinates of diploid Cochlearia records in the survey of cytogenetic evolution.

**Source data 2.** Coordinates of Cochlearia accessions with documented euploidies and aneuploidies in the survey of cytogenetic evolution.

**Source data 3.** Measured DNA content per chromosome (given in picograms; full data set).

**Source data 4.** Measured DNA content per chromosome (given in picograms; excluding Arctic diploids and *C. danica* as putative outliers).

where ice-free areas putatively occurred during the Penultimate Glacial Period (~140 kya; *Colleoni et al., 2016*). The earliest diverged organellar genomes of known diploids were found in the Arctic species *C. groenlandica* and *C. sessilifolia* (collected in British Columbia, Canada, and Kodiak Island, Alaska, respectively; pink lineage (Arctic II) in *Figure 2*, see *Appendix 1—figure 6* for details), with distribution ranges covering areas such as Beringia that were thought to have served as ice-free Pleistocene refugia (*Abbott and Brochmann, 2003*). European diploids are found in the green (Arctic-European) and purple (western European) lineages only. Some of the European polyploids, however, harbor early diverged haplotypes from the otherwise pink (Arctic II) lineage. Thus, except for the eastern European *C. borzaeana* with 2n=8x=48, all taxa from the pink lineage, as well as several taxa from the early diverged blue (Coastal Western European) and orange (Eastern European) lineages, have a base chromosome number of n=7 (see *Figure 2*).

## Genomic data and demographic modeling of diploid gene pools indicate glacial expansion

In order to analyze the nuclear fraction of our data, we mapped reads of each sample to our *C. pyrenaica* transcriptome reference (total length: 58,236,171 bp; *Lopez et al., 2017*). For 63 samples with sufficient nuclear sequence data quality (62 *Cochlearia* samples and *Ionopsidium megalospermum*), we generated a phylogenetic network using SplitsTree (*Huson, 1998*; *Huson and Bryant, 2006*) based on 447,919 biallelic SNPs. Concordant with our cytogenetic results, the network shows a clear separation of Arctic and European diploid taxa (*Figure 3a and b*; see *Appendix 1—figure 11* for detailed SplitsTree output). Close associations of both *C. tridactylites* and *C. danica* with *Ionopsidium* support the picture as revealed from organellar phylogenies. Further support for the early divergence of these two species came from an ML analysis based on 298,978 variant sites (same set of samples) performed with RAxML using an ascertainment bias correction and a general time-reversible substitution model assuming gamma distribution (*Appendix 1—figure 12*). Referring to single polyploid taxa, organellar and nuclear phylogenies are not congruent (*Appendix 1—figure 6* and *Appendix 1—figure 12*), which is best explained by the often allopolyploid origin of the tetraploids (e.g., *C. bavarica*, *Appendix 1—figure 13* and *Appendix 1—figure 14*), which may be even complicated by multiple and polytopic origin and further reticulation. Therefore, we do not discuss here the individual polyploid taxa (but see Appendix 1), instead focusing on the ancestral diploid gene pools.

A STRUCTURE analysis of *Cochlearia* samples only (same variant calling, 400,071 variants after excluding *Ionopsidium*) with K=3 (optimal K following Evanno Method; *Evanno et al., 2005*) revealed a pattern very similar to that obtained via SplitsTree, showing the two diploid clusters and a third cluster comprising *C. danica* samples (*Figure 3b* and *Appendix 1—figure 15*). We note potential signatures of admixture between the diploid clusters, especially in Icelandic 2n=12 and 2n=14 diploids and in several polyploid samples such as *C. bavarica* (discussed in detail in Appendix 1). Interestingly, *C. tridactylites*, the earliest diverged lineage according to the plastome analysis, was modeled as a mix of the Arctic gene pool and *C. danica*. In a separate TreeMix (*Pickrell and Pritchard, 2012*) analysis of all *Cochlearia* accessions and *Ionopsidium* (447,919 variants; up to 10 migration events), the bootstrapped graph for m=1 (optimal number of migration events according to Evanno Method; *Appendix 1—figure 16*) likewise indicates that *C. tridactylites* is admixed, with a majority ancestry from near the base of the European (excepting hexaploid *C. danica*) and Arctic groups, and a minority ancestry from the 2n=14 group of Arctic diploids (see *Appendix 1—figures 17–19*).

In order to elucidate the early stages of the *Cochlearia* species complex formation that might have facilitated the general cold association of the genus, we tested hypotheses regarding the evolutionary history of the diploid lineages from the Arctic (ARC) and European (EUR) distribution ranges by

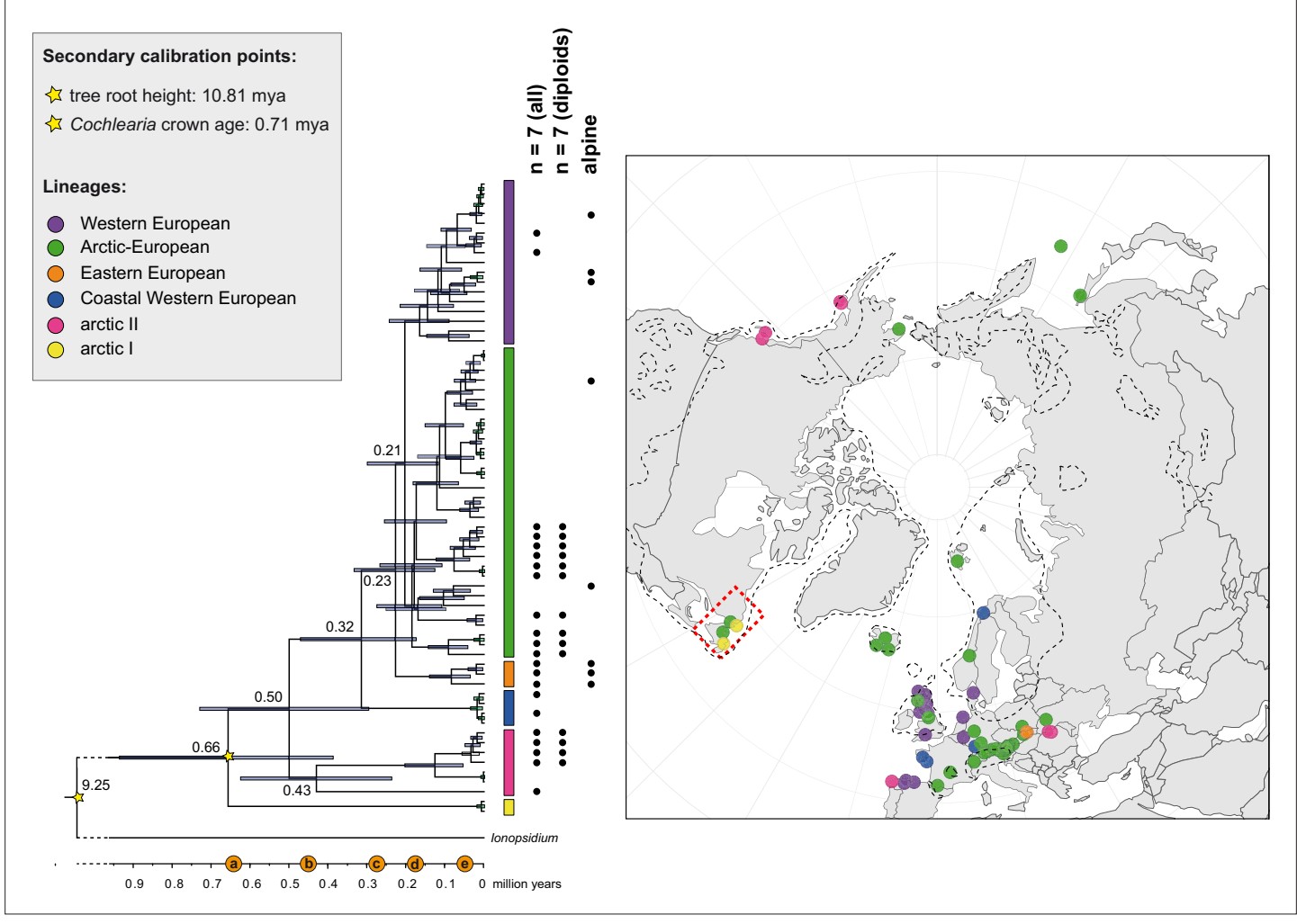

**Figure 2.** The maternal footprint of recurrent glacial speciation boosting. BEAST chronogram (*Figure 2—source data 1*) based on complete plastid genome sequence data supplemented by a geographical distribution pattern of the six main phylogenetic lineages (displayed as colored bars next to the tree; dots in the map are colored accordingly; *Figure 2—source data 2*). The *Ionopsidium* outgroup lineage is collapsed and condensed. The tree topology is congruent to the topology as revealed from maximum likelihood (ML) analysis. The full BEAST chronogram and the ML tree (incl. bootstrap support values) are given in *Appendix 1—figures 6, 9 and 10*. Individuals with a base chromosome number of n=7 (shown for all ploidy levels and diploids only) and accessions with an alpine or subalpine habitat type are marked with black dots next to respective tips. Letters (a)–(e) as displayed on the timeline indicate high glacial periods: (a) 640 kya, end of Günz glacial; (b) 450 kya, beginning of Mindel glacial; (c) 250–300 kya, Mindel-Riss inter-glacial; (d) 150–200 kya, Riss glacial; (e) 30–80 kya, Würm glacial. The black dashed line indicates the extent of the Last Glacial Maximum (LGM ~21 kya; based on *Ehlers and Gibbard, 2007*). The red dashed rectangle highlights a region with evidence for ice-free areas during the Penultimate Glacial Period (~140 kya; based on *Colleoni et al., 2016*).

The online version of this article includes the following source data for figure 2:

**Source data 1.** BEAST chronogram in NEXUS format.

**Source data 2.** Geographical distribution of phylogenetic lineages (plastid genome).

modeling possible histories using a coalescent framework with Approximate Bayesian Computation (ABC; *Tavaré et al., 1997*; *Beaumont et al., 2002*). For a data set of 22 European (2n=12) and 12 Arctic (2n=14) individuals (*Supplementary file 7* for sampling), we analyzed 2140 fourfold degenerate SNPs (Materials and methods). The sampling of Arctic accessions covers different taxa with deep evolutionary splits as exemplified by plastome analyses and the entire Arctic range is covered, there-fore unequal sampling size is expected to have a minor effect if any. Overall, the EUR metapopulation exhibits much higher genetic diversity than the ARC metapopulation (*Supplementary file 8*), which is in accordance with population-based analyses (*Koch et al., 1998*). In both populations, 2n=12 and 2n=14, we found an excess of rare alleles (negative Tajima's D, *Supplementary file 8*), indicating that

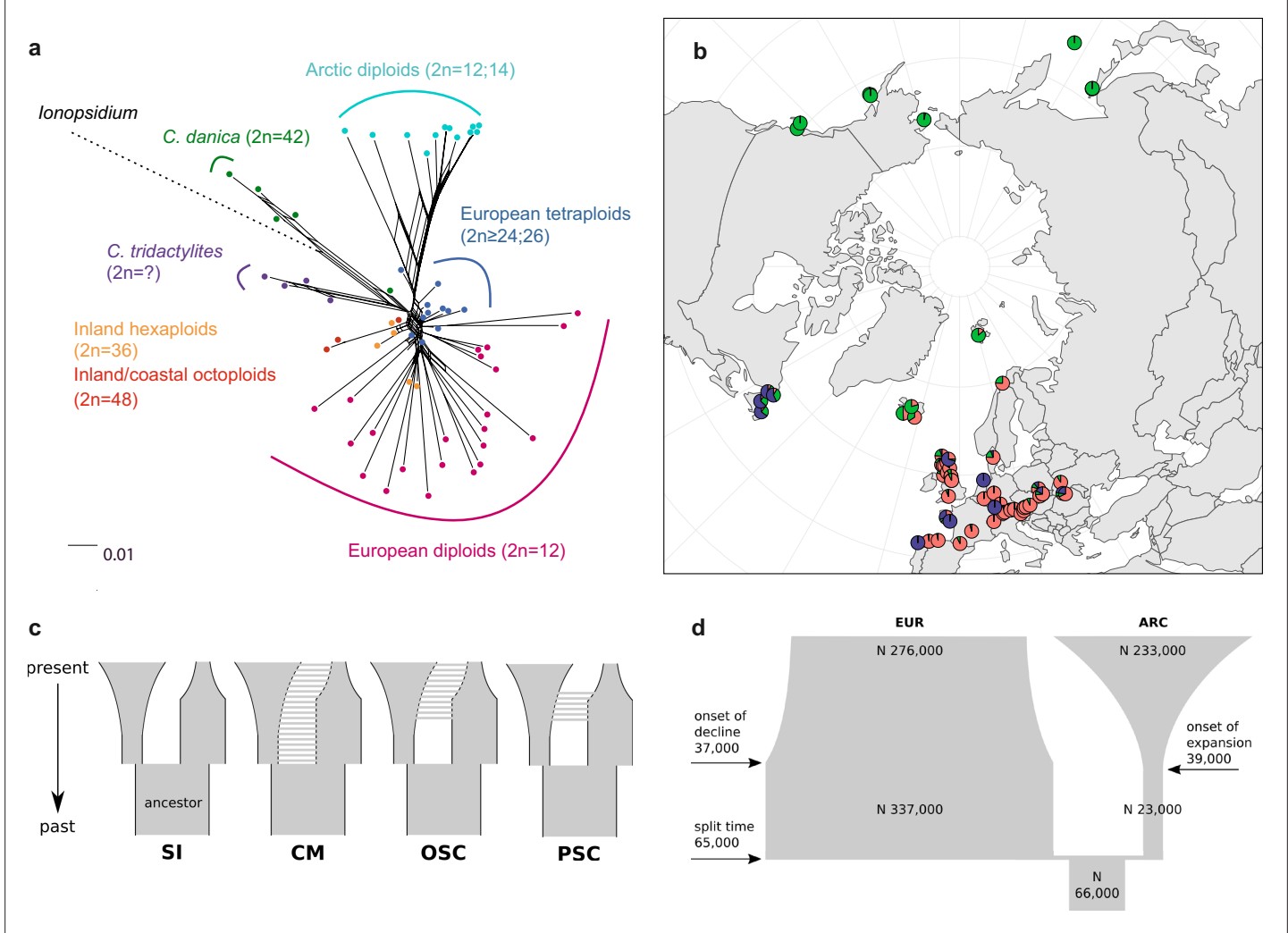

**Figure 3.** Demographic structure and history of the *Cochlearia* genus based on nuclear genome sequence data. (**a**) SplitsTree analysis of 62 *Cochlearia* samples and *Ionopsidium* (outgroup) using the NeighborNet algorithm based on uncorrected p-distances (***Figure 3—source data 1***; network with tip labels is given with ***Appendix 1—figure 11***). (**b**) Geographic distribution of 62 *Cochlearia* samples. Chart colors correspond to STRUCTURE results (62 *Cochlearia* samples; 400,071 variants) at K=3 (***Figure 3—source data 2***); green: Arctic gene pool, red: European gene pool, purple: *C. danica*-specific cluster (STRUCTURE result with tip labels given with ***Appendix 1—figure 15***). (**c**) Coalescent models for diploid populations explored with Approximate Bayesian Computation (ABC). SI=strict isolation, CM=continuous migration from the population split to the present, OSC=ongoing secondary contact with gene flow starting after population split and continuing to the present, PSC=past secondary contact with gene flow starting after population split and stopping before the present. (**d**) Most likely demographic history of diploid EUR (2n=12) and ARC (2n=14) *Cochlearia* populations from coalescent modeling (based on 22 EUR individuals and 12 ARC individuals; 2140 SNPs at fourfold degenerate sites) and ABC of models without gene flow and upper bound of the population size (N) prior as 400,000 (***Supplementary file 10***). N is in number of diploid individuals, and time in number of generations ago.

The online version of this article includes the following source data for figure 3:

**Source data 1.** SplitsTree network in NEXUS format.

**Source data 2.** STRUCTURE result at K=3 (georeferenced) and Delta K result (Evanno Method).

they are not in mutation-drift equilibrium. Differentiation and divergence between ARC and EUR were overall very low (Fst ~0.098, dxy ~0.0036, ***Supplementary file 8***).

Given the dynamic nature of their ice-age-associated speciation histories, we hypothesized that after the EUR and ARC metapopulations separated, they underwent dramatic changes in effective population sizes ($N_e$) over time, and that they experienced gene flow. We first tested four different hypotheses regarding the occurrence and relative timing of gene flow between ARC and EUR, because failure to account for gene flow can confound the inference of population size changes. Our gene

flow hypotheses were formulated as different model categories (*Figure 3c*), and random forest ABC (ABC-RF; *Marin and Pudlo, 2015*) was used to test under which model the observed data was most probable to have arisen. However, discriminating between the four gene flow models was ambiguous (see *Supplementary file 9*), as the most probable model depended on the priors, particularly on the upper bound of the $N_e$ priors. When allowing $N_e$ up to 400,000, a model of ongoing secondary contact (OSC) prevailed over a model without any gene flow (model SI; posterior probability >0.75, i.e., Bayes Factor >3), but OSC was not clearly better than a model with continuous gene flow (CM) or a past secondary contact (PSC). Yet, when choosing a less informative $N_e$ prior with a greater upper bound of three million, all four models were similarly in agreement with the observed data. In the absence of strong prior information for $N_e$, we could not establish the occurrence of gene flow between ARC and EUR metapopulations with confidence (see Appendix 3 for further information on ABC model choice).

To estimate changes in $N_e$ through time, we fitted parameters of a model without any gene flow (SI) and a model of OSC, considering both high and low $N_e$ upper prior bounds, amounting to a total of four model fits (*Figure 3c*; see *Supplementary file 10*). The general pattern of changes in $N_e$ through time were always modeled such that each of ARC and EUR had an older phase of constant $N_e$ followed by a recent phase of exponential expansion or decline.

The model without gene flow (SI) contained the fewest parameters, and this model with a smaller $N_e$ prior bound of maximal 400,000 provided the best fit to the data (smallest Euclidean distances between observed and predicted values from posterior predictive checks; see *Supplementary file 11*). This model fit (*Figure 3d*) is consistent with the remaining three model fits. Importantly, all model fits agreed about the relative $N_e$ of EUR and ARC: they evolved drastically differently, with the EUR metapopulation having risen to 4–12-fold the $N_e$ of their common ancestral population followed by a moderate decline (0.4–0.8-fold in three out of four models), or constant size up to the present (OSC with smaller $N_e$ upper bound). In contrast, the ARC metapopulation experienced a bottleneck after splitting from the common ancestor (0.2–0.5-fold), followed by a dramatic expansion of 9–52-fold (*Supplementary file 10*). Estimated $N_e$ for the ancestral population and for ARC during the ancient phase were robust to the choice of model and priors, but other $N_e$ parameters, in particular the ancient phase of EUR and the recent phase of ARC (i.e., the phases in which their $N_e$ were largest), increased when the prior's upper bound was increased. However, the relative $N_e$ trends through time were robust to these uncertainties, as mentioned above.

If EUR and ARC did not experience gene flow (SI model fit), they must have separated only about 65,000–73,000 generations ago, corresponding to 0.2–3 $N_e$ units. This estimate was robust to the choice of priors and may coincide with the last interglacial period (considering a 2-year average generation time; *Abs, 1999*). If gene flow is assumed (OSC), this split could have occurred earlier (119,000–227,000 generations ago; *Supplementary file 10*). Further parameters were poorly estimable as indicated by large prediction error, and little deviation between prior and posterior. These include the timing of the transitions from ancient to recent phases of N, the timing of migration, and the migration rates. Considering our BEAST analyses from plastome data, a split-time of 65,000–73,000 generations ago is more likely (e.g., dating of splits within the green evolutionary lineage). An important implication of this is that polyploid inland taxa, such as alpine hexaploid *C. tatrae* from the High Tatra mountains, showing footprints of both diploid gene pools cannot have evolved earlier than during the Last Interglacial. The example of hexaploid *C. bavarica* showed a postglacial origin and footprint of both diploid gene pools dated with approximately 12–15,000 years ago predating rapid Holocenic temperature increase.

## Genus-wide cold response characterized by metabolic profiling indicates an ancient origin of cold temperature tolerance

We hypothesized that a very early evolved tolerance to cold facilitated the observed widespread adaptation to alpine and subalpine habitats across the *Cochlearia* genus. To test this hypothesis, we performed metabolite profiling of central carbon metabolism using gas chromatography-mass spectrometry (GC-MS) for 28 worldwide *Cochlearia* accessions, including 14 taxa and 5 outgroup accessions (genus *Ionopsidium;* sampling details in *Supplementary file 11*).

Principal component analysis (PCA) based on 19 WorldClim temperature and precipitation-based bioclimatic variables distribute 46.8%, 29.6%, and 12.4% of variance with components 1, 2, and 3, respectively (*Appendix 1—figure 20*). A KMO test showed significant difference between the

correlation matrix of variables and an identity matrix (KMO=0.53, df 171, p<0.001; *Supplementary file 12*). A PCA using WorldClim variables 1–11 (temperature-related variables only) resulted in a similarly structured scree plot compared to the analysis using all 19 variables and recognizing four major clusters. Herein 44.3%, 39.5%, and 8.3% of variance are distributed with components 1, 2, and 3 (*Appendix 1—figure 21*). A KMO test also showed significant difference between the correlation matrix of variables and an identity matrix (KMO=0.725, df 55, p<0.001). For the *Cochlearia* accessions, we found the same four bioclimatically defined clusters based on a hierarchical cluster analysis of nine WorldClim bioclimatic variables chosen as most important for species' growth and fitness according to our field and cultivation experiences (*Figure 4a and b*). Cluster 1 combines coastal polyploid accessions from the northern UK; Cluster 2 includes European inland accessions (diploids and polyploids); Cluster 3 comprises European accessions with Arctic/alpine habitat types (Norway, Carpathians, High Tatra Mountains, and Austrian Alps); and cluster four includes Arctic accessions (Iceland, Alaska) and those from alpine habitat types from the UK. Thus, the various habitat types and major ecological features defining the taxa of the genus are well represented and the defined clustering may indicate the value of ecological parameters defining species boundaries (*Koch et al., 1998*; *Koch et al., 1999*). *Ionopsidium* represented a fifth, Mediterranean, bioclimatically defined group.

For all groups, metabolic profiles were collected before and after a 20-day treatment under cold (5°C) or control conditions (18°C/20°C; *Figure 4c*). From these profiles, we exported peak areas for a set of 40 compounds consistently detected across our samples that were annotated as intermediates within central carbon metabolic pathways (see *Supplementary file 13* for a list of detected compounds; raw/normalized data provided in *Supplementary file 14*).

As has been best seen in the *Arabidopsis* cold stress metabolome (*Cook et al., 2004*; *Kaplan et al., 2004*; *Kaplan et al., 2007*), we generally found increases in the relative levels of many of the targeted compounds after the cold treatment, especially for carbohydrates and amino acids, exemplifying a similar physiological mechanism (data grouped by bioclimatic clusters, analyzed by one-way ANOVAs; *Appendix 1—figures 22–27*). Significant increases in carbohydrate levels as a well-known reaction to cold stress in plants were detected in all herein analyzed carbohydrates except for sucrose. Important functions of various carbohydrates as cryoprotectants and/or signaling molecules in the plant cold response have been suggested before, for glucose (reviewed by *Janská et al., 2010*) and mannose (*Davey et al., 2008*). Among the analyzed amino acids showing increased levels after cold treatment, proline is well known for its role in the plant response to various abiotic stresses including low temperatures (e.g., reviewed by *Ashraf and Foolad, 2007*). Likewise, increased levels of glutamic acid and aspartic acid, both associated with the citric acid cycle, display a typical stress response consistent with the cold metabolomes of *A. thaliana* (*Kaplan et al., 2004*).

Initially, we ran PCA analyses with different group priors for the 40 compound matrix, yet for most of the pre-defined clusters these analyses revealed little discriminating structure (see *Appendix 1—figure 28*, *Appendix 1—figure 29*). Thus, to increase power, we performed discriminant analyses of principal components (DAPC), thereby maximizing the variation between clusters while minimizing the variation within (*Jombart et al., 2010*). When grouping the data by temperature treatment, DAPC illustrates a metabolomic response of all clusters to temperature stress (cold treatment at 5°C; *Figure 4d*; *Appendix 1—figure 30*). Although our data set comprises different species from two genera spanning a large distribution range from the Mediterranean toward the Artic no demarcation of the cold responses could be revealed either among the four bioclimatically defined groups of species and accessions of *Cochlearia* or even between *Cochlearia* and *Ionopsidium* (*Figure 4e*).

The putative high pleiotropy inherent in metabolism may also force compensatory variants, which are then private in evolutionary lineages, species, or accessions. Crossing experiments in *A. thaliana* accession have shown that progenies have a much wider diversity of primary cold metabolism phenotypes, which is suggesting that when crossing the harmony of primary and compensatory mutations is broken and then a wider distribution of phenotypes will emerge. This concept may also contribute to explain the evolutionary success of the putative hybrid and allopolyploid *Cochlearia* taxa, and pointing toward parallel evolution. However, in particular with the polyploid *Cochlearia* taxa detailed knowledge on past and recent genetic admixture is essential to infer parallel evolution, and herein presented metabolomic data cannot contribute to shed light on parallel and adaptive evolution of *Cochlearia* polyploids. This will need detailed and comparative analyses of the individual evolutionary and biogeographic history of any polyploid *Cochlearia* taxon.

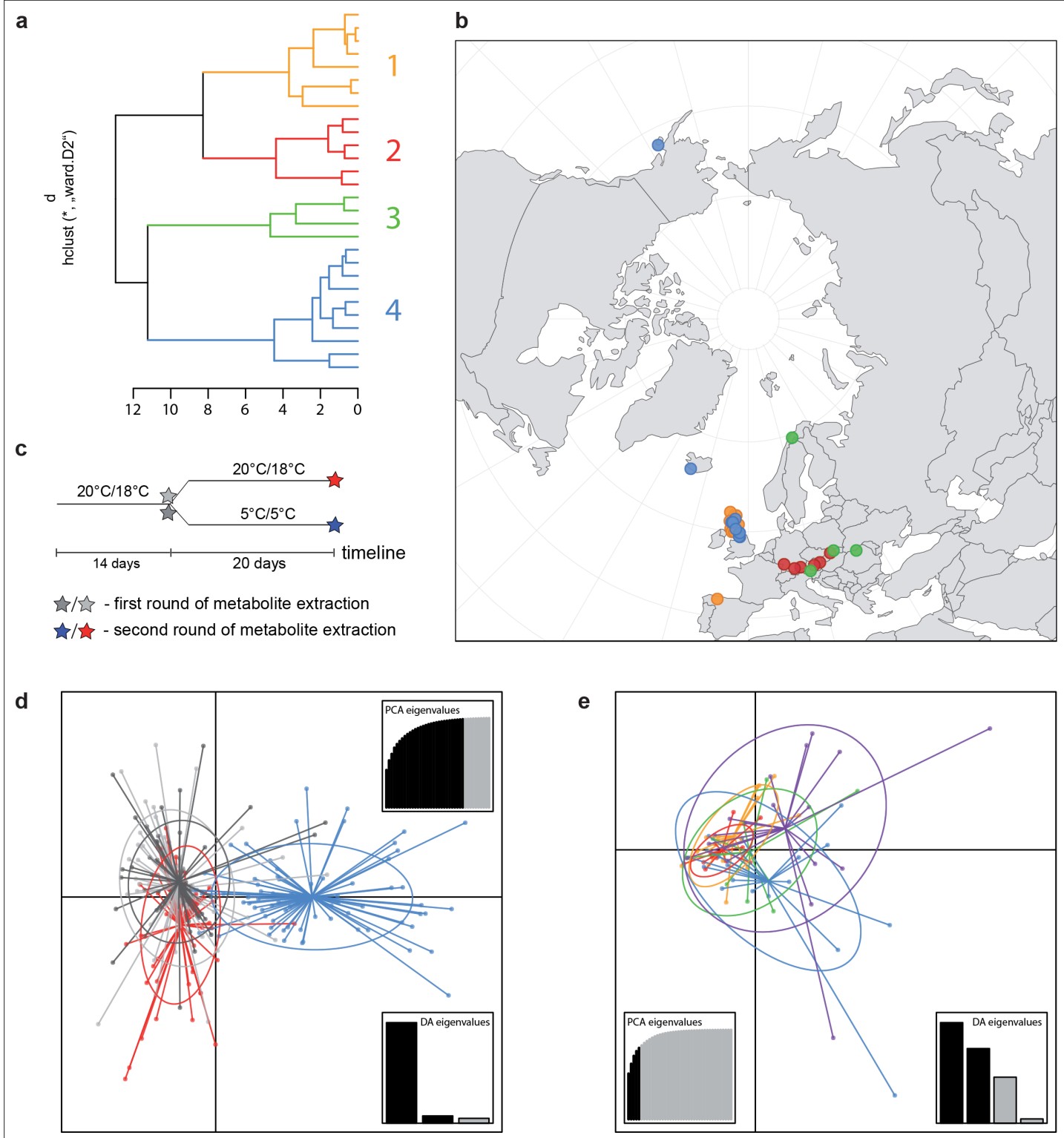

**Figure 4.** A common cold response indicated by metabolic profile clustering. (**a**) Hierarchical cluster analysis of 28 *Cochlearia* accessions based on five temperature-related and four temperature/precipitation-related bioclimatic variables (WorldClim; given with **Figure 4—source data 1**) using Euclidian distances and the Ward's method (**Ward Jr JH, 1963**; **Murtagh and Legendre, 2014**). Cluster 1: coastal accessions of polyploids from the northern UK; Cluster 2: European inland accessions (diploids and polyploids); Cluster 3: European accessions with Arctic/alpine habitat types (Norway, Carpathians, High Tatra Mountains, and Austrian Alps); Cluster 4: Arctic accessions (Iceland, Alaska) and alpine habitat types from the UK. (**b**) Geographical distribution of bioclimatic clusters as discovered via hierarchical cluster analysis (colors are representing clusters 1–4, see (**a**); **Figure 4—**

*Figure 4 continued on next page*

*Figure 4 continued*

*source data 2*). (**c**) Experimental setup of temperature treatment with a timeline for metabolite extractions. (**d**) DAPC based on all metabolic profiling measurements, grouped by treatment (colors are representing metabolite extractions as illustrated in **c**) (*Figure 4—source data 3*). (**e**) DAPC of metabolite measurements after cold treatment grouped by four bioclimatic clusters (colors as in (**a**) and (**b**)) and *Ionopsidium* (purple) (*Figure 4—source data 1*_SourceData4).

The online version of this article includes the following source data for figure 4:

**Source data 1.** 9 WorldClim bioclimatic variables selected for hierarchical cluster analysis.

**Source data 2.** Geographical distribution of bioclimatic clusters.

**Source data 3.** DAPC result based on all metabolic profiling measurements, grouped by treatment.

## Discussion

Previous studies provided first insights into the complex evolutionary history of the genus *Cochlearia* (e.g., *Koch et al., 1996*; *Koch et al., 1998*; *Koch et al., 1999*; *Koch, 2002*; *Koch et al., 2003*; *Kochjarová et al., 2006*; *Cieslak et al., 2007*; *Koch, 2012*; *Brandrud et al., 2017*; *Bray et al., 2020*). Yet, a genus-wide picture has been missing. Here, we incorporate cytogenetics, phylogenomics, and metabolomics to present the complex and dynamic evolutionary past of this cold adapted, dramatically radiating genus. Today, living in a warming world, almost all of these taxa are declining since they strongly depend on the existence of cold habitats, which are among the first to be influenced by globally rising temperatures and the resultant transformations in abiotic conditions (*Walther et al., 2002*; *Parmesan, 2006*; *Alsos et al., 2012*; *Barrett et al., 2015*).

As revealed from the continental-scale geographical partitioning of diploid cytotypes (2n=12 and 2n=14; *Figure 1a*), we see an early split of two diploid gene pools followed by separation of the two diploid metapopulations in Arctic regions (2n=14) and in Europe (2n=12). Only in Iceland do both diploid cytotypes co-occur. Both cytotypes underwent genome and chromosome size reduction within this Arctic environment, which may be seen as a metabolic advantage (e.g., meiotic cell division under cold stress *Bomblies et al., 2015*, or triggered by shorter life cycles). As shown below from a phylogenetic point of view, the most parsimonious scenario recognizes 2n=14 and thus a base chromosome number of n=7 as ancestral state (*Figure 2*). A possible, but far from causal, hypothesis is that the ancestral and exclusively circum-Arctic 2n=14 taxa may have also benefitted from one extra centromere, which increases total recombination rate and thus may also be favorable under harsh conditions and drastic environmental changes (*Stapley et al., 2017*).

Whereas the presence of chromosomal aberrations in *Cochlearia* has been postulated to be caused by B-chromosomes (*Gill, 1971b*; *Gill, 1973*; *Gill et al., 1978*; *Nordal, 1988*), these aneuploidies could also be the result of hybridization/polyploidization events followed by mitotic/meiotic difficulties or independent ascending/descending dysploidies via chromosome rearrangements (*Lysak, 2014*), although the exact mechanisms in this genus are yet unclear. Karyotypic variation and aneuploidy following hybridization in plants have been described as adjustment processes within the allopolyploid genome putatively connected to the beginning of re-diploidization (*Mandáková et al., 2016*). For *Cochlearia*, the high frequency of chromosomal aberrations in polyploids likely reflects a highly dynamic genome evolution, given the near absence of interspecific fertility barriers between cytotypes (*Gill, 1971a*; *Gill, 1973*; *Koch et al., 1998*). Moreover, a trend for reducing genomic redundancy in the young polyploids is suggested by the negative correlation of chromosome size with increasing chromosome numbers. Combined with the stable diploid karyotypes, these patterns are well in accordance with the hypothesis of a necessary balance between both genomic flexibility and stability allowing for further adaptive evolution to a continuously changing environment (*Hohmann et al., 2015*).

Despite the highly fluid genome dynamics across the genus, we did observe a tight concurrence of high glacial periods and diversification times within the six major lineages (*Figure 2*). This suggests a boosting impact of these climatic fluctuations on the diversification of the genus. Accordingly, the observed polytomies, especially within the green and the purple lineage, likely reflect events of rapid, reticulate radiation during the last two glacial periods, where we speculate that polyploidization and hybridization may have facilitated adaptation to dramatically changing environments (*Hohmann et al., 2015*).

Based on the organellar phylogenies and divergence-time estimates (and contradicting former hypotheses [*Gill, 1971a*; *Gill, 1973*; *Koch et al., 1996*]), an evolutionary scenario with an ancestral base chromosome number of n=7 in diploid Arctic ancestors followed by reduction to n=6 in European diploids is most parsimonious. This view is supported by the finding that also within the green lineage (Arctic-European), basal groups are confined to Arctic regions. However, in accordance with former hypotheses (*Koch, 2002*; *Koch et al., 2003*), we can also postulate that additional glacial refugia must have existed in Europe north of the primary refugia for European species (Iberian Peninsula, Southern Italy, Balkans; *Taberlet et al., 1998*), as also described for other taxa (*Stewart and Lister, 2001*; *Tribsch et al., 2002*; *Tribsch and Schönswetter, 2003*; *Provan and Bennett, 2008*). The putative Arctic background of the entire genus *Cochlearia* might have (a) promoted the general association with cold-characterized habitat types throughout the genus and (b) facilitated later adaptation to different high alpine regions in Europe. As revealed from the plastid phylogeny (*Appendix 1—figure 6*), the latter has taken place repeatedly in the High Tatra Mountains (*C. tatrae*, orange lineage, Eastern Europe), in the Austrian Alps (*C. excelsa*, green lineage, Arctic-European), in alpine habitat types in Great Britain (*C. alpina*/*C. micacea*, purple lineage, western European), and in several other European habitats with a more subalpine character, namely in the Pyrenees or in Switzerland.

With respect to the putative Arctic origin of the genus in its current shape with an ancestral base chromosome number of n=7, the very strong admixture between both diploid genetic clusters of different chromosome numbers (as seen in diploid Arctic Icelandic accessions) provides evidence of genetic coherence. Considering the numerous polyploids and their independent, multiple origins in *Cochlearia* (see Appendix 1) and the exceptional case of sympatry of both different diploid cytotypes in an arctic environment in Iceland, it seems that the two early separated gene pools serve as a continuous source for the evolution of new, often transient, and currently highly endangered species. Regarding the demographic histories of the two gene pools, our ABC data suggest that ARC and EUR metapopulations are only weakly differentiated and share most of their genetic variation, either because of recent separation or gene flow. They currently have large and similar effective population sizes, but ARC has expanded after a bottleneck whereas EUR has remained constant, or recently declined.

When exposed to low but non-freezing temperatures, plants commonly activate cold acclimation responses involving changes in gene expression, enzymatic activities and ultimately, concentrations of many central carbon intermediates (*Kaplan et al., 2004*; *Kaplan et al., 2007*; *Hoermiller et al., 2017*). As part of this low temperature response, particular accumulations of carbohydrates, polyols, and amino acids are detected (*Kaplan et al., 2004*; *Krasensky and Jonak, 2012*). All *Cochlearia* and *Ionopsidium* species analyzed herein showed this response, too. However, despite covering a broad phylogenetic and taxonomic diversity and clustering taxa according to bioclimatically defined clusters, low temperature response did not differentiate among those, as it has been demonstrated for *A. thaliana* across a gradient from the Mediterranean toward Scandinavia (*Weiszmann et al., 2020*).

Given that all *Cochlearia* species, regardless of origin and habitat type, exhibited similar responses to the temperature treatment performed under controlled conditions for central carbon intermediates, we hypothesize that these similar responses to cold within the genus reflect continuous connections to cold-characterized habitats acquired over the course of a migration to northern/circum-Arctic regions during the early evolution of the genus. Apparently, a constitutive cold tolerance has not been lost secondarily within *Cochlearia*. Future analyses of the cold-induced metabolome of *Cochlearia* will require untargeted metabolomics analyses to systematically account for the contribution of species/ecotype-specific metabolic characters in the overall cold response. In this respect, previous studies on glucosinolate and tropane alkaloid chemotypes, which are key to biotic stress adaptation, have shown rapid structural diversification and ecotype specificities within the *Cochlearia* genus (*Brock et al., 2006*; *Blažević et al., 2020*). Regarding the overall similar response of central carbon metabolism to the cold treatment of annual species from the genus *Ionopsidium,* which are adapted to at least seasonally dry habitats and/or salt habitats, we may conclude that commonalities between cold and salinity/drought stress would enable a putative recruitment of the cold tolerance from existing adaptations to drought and/or salt as hypothesized for other plant taxa (*Böndel et al., 2018*; *Exposito-Alonso et al., 2017*). We speculate that a potential mechanism for this conserved response may be hinted at by the pleiotropic nature of metabolic networks, potentially exposing these networks to constraining selection.

Because both drought, salt and cold stress strongly involve osmotic challenge, our data also support the idea that an evolution to one of these stressors in the common ancestor not only allowed for the survival under Mediterranean conditions (*Ionopsidium*), but also may have preadapted the nascent *Cochlearia* genus to habitats characterized by cold or salt, and with Arctic environments often characterized by drought, too.

## Conclusion

Our results indicate that the immense karyotypic diversity of *Cochlearia* reflects not only postglacial or even recent hybridization, but also speciation boosting during Pleistocene glaciation cycles. This in turn appears to have triggered the parallel exploration of new ecological niches on a continental scale ranging from the Arctic toward lower latitude alpine regions. We speculate that species richness and thereby genetic diversity was meanwhile rescued in reticulate and polyploid gene pools, often resulting in sometimes porous species boundaries. Irrespective of their habitat contrasts, both *Cochlearia* and its Mediterranean sister genus *Ionopsidium* show a pronounced response to cold stress based on metabolite profiling. This leads to the idea that a shared ancestral adaptation to drought enabled not only the survival under dry Mediterranean conditions, but also may have preadapted the nascent *Cochlearia* genus to habitats characterized by cold or salt: all of these stressors have at their cores osmotic challenge. As colonization of these habitat types occurred multiple times independently, *Cochlearia* represents a powerful system to predict the fate of this and other taxa in a world marked by climate change.

## Materials and methods

### Plant material and taxon sampling for cytogenetic analyses

Our cytogenetic analyses (chromosome counts and flow cytometry) were performed on a large collection of living plants including different ploidy levels and covering big parts of the genus' distribution range. The living plants, either collected in the wild or grown from seeds in the Botanical Garden Heidelberg, were cultivated under greenhouse conditions in a substrate composed of seedling potting soil, quartz sand, and either composted earth or Ökohum's peat-based substrate (Ökohum GmbH, Herbertingen, Germany). During the summer season, we transferred parts of the plant collection to a plant room with a controlled 16/8 hr day/night cycle under 20°C.

Aside from own data, we performed a literature survey for published chromosome counts/genome size measurements considering literature from the last 100 years (*Supplementary file 4*). Published and own datapoints were finally merged in a comprehensive database (*Supplementary file 2*) and if possible, coordinates of the respective population localities were added as listed in the literature or carefully approximated based thereon. Every population listed by one author was treated as an individual datapoint. This way, we collected 575 georeferenced database entries and visualized these using the R package 'ggplot2' (*Wickham, 2016*; R version 3.3.1).

### Flow cytometry

In order to minimize the risk of contaminating factors such as fungi and to reduce the generally high amount of endopolyploidy, we selected only very young and healthy leaves for flow cytometry sample preparation. Nuclei extraction and staining were performed on ice using the Partec Two-Step Kit CyStain PI Absolute P (Partec GmbH, Münster, Germany) following the manufacturer's instructions with minor modifications. Namely, 15 mM β-mercaptoethanol and 1% w/v polyvinylpyrrolidone 25 (PVP) were added to the staining buffer. After chopping leaf material of each sample together with the chosen internal standard plant in 500 µl of the extraction buffer, the suspensions were filtered through 50 µm CellTrics filter (Sysmex Partec GmbH, Görlitz, Germany). 2000 µl of the staining buffer was added to each flow-through and samples were incubated on ice for 30–60 min.

A list of standard plants used for flow cytometry experiments together with respective 2C-values is given with *Supplementary file 15*. In order to ensure consistent measurements, 2C-values of all standard plants were finally readjusted to *Solanum*, which was used as a reference.

For several samples, instead of using the Partec Kit, an alternative lysis buffer LB01 as specified by *Dpoolezel et al., 1989* was used, containing 15 mM Tris, 2 mM Na2EDTA, 0.5 mM spermine tetra-hydrochloride, 80 mM KCl, 20 mM NaCl, and 0.1% v/v Triton X-100; 15 mM β-mercaptoethanol and

0.1 mg/ml RNAse A added right before preparation. Both buffers resulted in similar peak qualities, generated in a Partec CyFlow Space flow cytometer (Sysmex Partec GmbH) using a 30 mW green solid-state laser ($\lambda$ =532 nm). Gating and peak analysis were performed using the Partec FloMax software version 2.4 (as exemplified in *Appendix 1—figure 31*; detailed flow cytometry results given with *Supplementary file 16*).

We performed both simple linear regression analyses and rank correlation tests (Spearman Rank and Kendall Tau correlations) based on measured 2C values in R version 3.3.1 (*R Development Core Team, 2013*), testing the relationships of (1) chromosome number versus genome size and (2) chromosome number versus DNA content per chromosome.

## Plant material and taxon sampling for genome resequencing

The taxon sampling for the genome resequencing analyses aimed at covering all ploidy levels and the whole distribution range of the genus *Cochlearia*, that way including all different ecotypic variants (e.g., montane vs. alpine, or limestone vs. siliceous bedrock, salt vs. sweet water). We selected a total of 65 samples (*Supplementary file 2*). *C. islandica* was treated as a separate species, tetraploid populations of British *C. pyrenaica* were referred to as *C. pyrenaica* subsp. *alpina* (abbreviated as *C. alpina* hereafter) and sub-species levels in *C. officinalis* samples from Scandinavia were elided. According to this taxonomic treatment, 19 taxa were included in the study, comprising all accepted species as listed in BrassiBase (*Kiefer et al., 2014*; see *Supplementary file 1* for a list of accepted species and sub-species) and representing also respective species ranges. Every population/accession is represented by one sample and three samples/taxa from the sister genus *Ionopsidium* were selected to serve as outgroup.

Leaf samples were either selected from own collections (stored as herbarium vouchers, silica-dried material, or as living plants) or they were received from other institutions in the form of herbarium specimen, silica-dried samples, and/or seed material, which was then grown at the Botanical Garden Heidelberg.

A detailed summary of all sequenced samples with information on sample locations and the respective type of leaf material are given with *Supplementary file 6*.

## DNA extraction and NGS sequencing

The Invisorb Spin Plant Mini Kit (STRATEC Biomedical AG, Birkenfeld, Germany) was used for extractions of DNA from either herbarium, silica-dried, or fresh leaf material. Five samples were enriched for chloroplasts prior DNA extraction using a Percoll step-gradient centrifugation (modified from *Jansen et al., 2005*).

Library preparation (total genomic DNA) and sequencing were performed at the CellNetworks Deep Sequencing Core Facility (Heidelberg) with library insert sizes of 200–400 bp. DNA fragmentation was performed on a Covaris S2 instrument. Either the TruSeq Kit (Illumina Inc, San Diego, CA) or the NEBNext Ultra DNA Library Prep Kit for Illumina (formerly NEBNext DNA Library Prep Kit for Illumina; New England Biolabs Inc, Ipswich, MA) and the NEBNext Multiplex Oligos for Illumina were used for library preparations. Sequencing of multiplexed libraries (6–12 samples per lane) was performed on an Illumina HiSeq 2000 system in paired-end mode (100 bp).

## NGS data analysis

### K-mer analysis for ploidy estimation of *C. tridactylites*

We performed a *k*-mer analysis using Jellyfish (*Marçais and Kingsford, 2011*) for all sequenced samples of *C. tridactylites* in order to predict the species-specific genome size. After adapter and quality trimming of raw reads using Trimmomatic version 0.32 (*Bolger et al., 2014*; LEADING:20, TRAILING:20, SLIDINGWINDOW:4:15, and MINLEN:50), the distribution of the *k*-mers 17 and 25 were estimated using Jellyfish. The resulting histograms were analyzed and visualized in R version 3.3.1 (see *Appendix 2—figure 1*).

## Phylogenomic analyses

### Chloroplast genome assemblies and annotation

A combination of both de novo assemblies and reference-based mappings was used for the reconstruction of complete chloroplast genomes of 68 samples. We used the CLC Genomics Workbench

version 6.0.4 (CLC Bio, Aarhus, Denmark) for quality and adapter trimming with a quality score limit of 0.001 (corresponding to a phred score of 30) and the minimum read length set to 50 bp. De novo assemblies were performed in CLC for paired trimmed reads (length and similarity fractions set to 0.8). Chloroplast contigs were identified via BLASTn analysis (default settings) and aligned manually using PhyDE v0.9971 (*Muller et al., 2010*). Gaps between non-overlapping contigs were filled from a complete chloroplast genome that served as a reference, and trimmed reads were mapped back against the so-created preliminary genomes using the CLC *Map Reads to Reference* tool with length and similarity fraction set to 0.9. After manually checking the mapping quality, we performed a CLC variant detection with a variant probability of 0.1 and the remaining mapping errors were corrected manually. As an additional quality control, we ran reference-based mappings using the *bwa-mem* algorithm as implemented in BWA version 0.7.8 (*Li, 2013*; default parameters for matching score (1), mismatch penalty (4), gap open penalty (6), gap extension penalty (1) and clipping penalty (5); penalty for an unpaired read pair set to 15). Ambiguously mapped reads and putative PCR duplicated were removed via SAMtools version 0.1.19 (*Li et al., 2009*; *Li, 2011*) and GATK (*McKenna et al., 2010*) was applied for a local realignment and an evaluation of per base mapping quality. Finally, consensus sequences were extracted via GATK and masked based on the respective mapping quality using the *maskfasta* tool implemented in BEDtools version 2.19 (*Quinlan and Hall, 2010*; *Quinlan, 2014*).

For the annotation of complete or nearly complete chloroplast genomes, the annotated chloroplast genome of *A. thaliana* (NC_000923) served as an initial reference and was therefore aligned to a *Cochlearia* chloroplast genome using MAFFT v7.017 (*Katoh et al., 2002*; *Katoh et al., 2005*; *Katoh and Standley, 2013*) as implemented in Geneious version 7.1.7 (Biomatters Ltd, Auckland, New Zealand). We used the FFT-NS-I x1000 algorithm with a 200PAM/k = 2 scoring matrix, a gap open penalty of 1.53 and offset value of 0.123 and transferred annotations from the reference in Geneious via the *Transfer Annotation* tool (required similarity 65%). We then checked and adjusted all transferred annotations manually. After the first annotation, the remaining chloroplast samples were aligned to and annotated from the most closely related *Cochlearia* chloroplast genome.

## Chloroplast genome alignments and phylogenetic trees

Sequences of 68 annotated and quality masked cp genomes were aligned in Geneious using MAFFT v7.017 with settings as above. We used Gblocks v0.91b (*Castresana, 2000*) to subject alignment blocks of exons, introns, and intergenic spacers to an automatic alignment quality control (minimum block length of 5; gap positions allowed in up to 50% of the samples). Four putative pseudo-genes (trnH, ndhK, rrn16S, and rrn23S) were excluded manually as well as three AT-rich regions of poor mapping quality (within trnE-trnT intron, rpl16 intron, and trnH-psbA intergenic region). A search for the best partitioning schemes and substitution models for the final 258 blocks was performed using PartitionFinder v1.1.1 (*Lanfear et al., 2012*; *Lanfear et al., 2014*) with branch lengths set to be unlinked.

We used RAxML version 8.1.16 (*Stamatakis, 2014*) for phylogenetic tree-building (1000 bootstrap replicates) using the GTR+$\Gamma$ model for the partitioning schemes as determined by PartitionFinder and with three *Ionopsidium* samples set as outgroup.

## Divergence time estimation based on whole chloroplast genome alignment

In the absence of reliable fossil records within the Brassicaceae family (*Franzke et al., 2016*), a subset of five *Cochlearia* samples and one *Ionopsidium* sample were included in a large-scale divergence time estimation analysis on taxa from the whole Superrosidae clade by *Hohmann et al., 2015*. The analysis was based on 73 conserved chloroplast genes (51 protein-coding genes, 19 tRNAs, and 3 rRNAs) and allowed for four reliable fossil constraints (also used by *Njuguna et al., 2013*; *Magallón et al., 2015*), to be placed as primary calibration points along the tree (see *Hohmann et al., 2015* for details on taxon sampling and data analysis). Namely, minimum ages of calibration points were the *Prunus/Malus* split set to 48.5 mya (*Benedict et al., 2011*), the *Castanea/Cucumis* split set to 84 mya (*Sims et al., 1999*), the *Mangifera/Citrus* split set to 65 mya (*Knobloch and Mai, 1986*) and the *Oenothera/Eucalyptus* split set to 88.2 mya (*Takahashi et al., 1999*). The root age of the tree was set to 92–125 mya (uniform distribution) in accordance with *Magallón et al., 2015*.

Based on this foregoing analysis, we selected secondary calibration points for divergence time estimation within the whole Cochlearieae chloroplast data set as follows: the *Ionopsidium*/*Cochlearia* split was set to 10.81 mya and the *Cochlearia* crown age was set to 0.71 mya (normal distributions). Divergence time estimation was performed in BEAST 1.7.5 (*Drummond et al., 2012*) based on the alignment of whole chloroplast genomes (122,798 bp), with partitioning schemes as received from PartitionFinder (see above). Independent site and clock models and a combined partition tree were selected for the two partitions (GTR+$\Gamma$) and in order to allow for varying rates among branches, an uncorrelated lognormal relaxed clock model with estimated rates was applied (*Drummond et al., 2006*). We performed two independent MCMC runs with 100 million generations each (samples taken every 10,000 generations). After combining the two tree files in LogCombiner version 1.5.5 (*Drummond et al., 2012*) with a burn-in of 50,000,000 generations, treeAnnotator version 1.7.5 (*Drummond et al., 2012*) was used to combine the 18,000 generated trees to a maximum clade credibility tree which was finally visualized in FigTree version 1.4.1 (*Drummond et al., 2012*).

## Mitochondrial genome data analysis

Given the challenges in assembling plant mitochondrial genomes de novo, mainly caused by the high amount of repetitive regions (*Schatz et al., 2010*; *Straub et al., 2012*), we followed an approach by *Straub et al., 2011*, which combines the de novo assembly of mitochondrial consensus sequences and reference-based mappings to these contigs, serving as a partial mitochondrial reference genome. Therefore, eight long mitochondrial contigs of one *Cochlearia* sample (Cmica_0979; *Supplementary file 17*) were retrieved from a de novo assembly performed with CLC (same settings as specified above for chloroplast genome de novo assemblies) and identified via BLASTn. The CLC *Extract Consensus Sequence* tool was used to extract contigs (minimum coverage threshold of 10×) and annotation of genes and coding sequences was performed with the Mitofy Webserver (*Alverson et al., 2010*) under default settings. The final reference had a total length of 307,510 bp covering 32 protein-coding genes (some of them partial) and 15 tRNA sequences (13 different tRNAs).

Sequencing reads were trimmed using Trimmomatic (see above) and reference-based mappings were performed with BWA-MEM (mapping settings as specified above for chloroplast genomes). A *Cochlearia* chloroplast genome as well as the nuclear genome of *A. thaliana* (NC_003070, NC_003071, NC_003074, NC_003075, and NC_003076) were included as additional references in order to reduce mis-mapping of reads originating from pseudogenes. After a mapping quality improvement (same steps as described for chloroplast genome assemblies), the GATK tool *CallableLoci* with a minimum coverage of 20× and a minimum mapping quality of 30 was used to identify regions of high mapping quality. Hereafter, we extracted contig sequences using the GATK tools *UnifiedGenotyper* (output mode EMIT_ALL_SITES) and *FastaAlternateReferenceMaker*. Regions that had not passed the *CallableLoci* quality filters were masked in the final fasta files using BEDtools version 2.19 (*Quinlan and Hall, 2010*; *Quinlan, 2014*) *maskfasta*. Seven *Cochlearia* samples and two *Ionopsidium* outgroup samples with more than 50% missing data were excluded from the mitochondrial phylogenetic analysis.

Geneious was used to concatenate the eight contig alignments and after manually excluding putative pseudogenes, regions of poor alignment quality were removed via Gblocks (minimum block length set to 10). After excluding positions that were masked in all samples, the final alignment had a length of 232,036 bp covering 306 parsimony informative SNPs.

An ML analysis was performed with RAxML version 8.1.16 (*Stamatakis, 2014*) with 1000 rapid bootstrap replicates and the GTR+$\Gamma$+I model, selected as the most appropriate substitution model with jModelTest version 2.1.7 (*Posada, 2008*; *Darriba et al., 2012*).

In order to compare the two organellar phylogenies, we generated a tanglegram using dendroscope version 3.7.2 (*Huson and Scornavacca, 2012*) after collapsing branches with bootstrap support below 95%. The plastid phylogeny was reduced to match the taxon set of the mitochondrial phylogeny using the function 'drop.tip' as implemented in the R package 'ape' version 5.1 (*Paradis and Schliep, 2019*).

## Nuclear genome data analysis
### Mapping approach and SNP calling
In order to generate a comprehensive SNP data set for the nuclear genome, we performed reference-based mappings against the published transcriptome of *C. pyrenaica* (total length: 58,236,171 bp;

*Lopez et al., 2017*). Therefore, trimmed reads were mapped to the reference using BWA-MEM (Trimmomatic and mapping settings as specified above). Mapping quality was improved and investigated using SAMtools version 0.1.19 and the GATK tools *RealignerTargetCreator*, *IndelRealigner*, *DepthOfCoverage*. For further analyses, the minimum coverage was set to 2× and an upper 2% of coverage cutoff was used for masking high coverage sites for every sample individually, thereby excluding organellar and rDNA transcripts. Three *Cochlearia* samples (Caes_0160, Coff_1289, and Ctat_1017) and two *Ionopsidium* samples (Iabu_1074 and Iacau_1072) were excluded from nuclear genome data analysis due to low overall mapping quality (less than 40% of the reference-based mapping positions fulfilling quality and coverage requirements). The *I. megalospermum* sample (Imega_1776) also failed the 40% cutoff (~37% sites callable) but was kept in the analysis in order to have an outgroup sample included.

Callable sites of the respective samples were combined via the *multiIntersectBed* command implemented in BEDtools version 2.19 and SNP callings were performed in sites passing the chosen quality requirements in all samples. GATK's *UnifiedGenotyper* was used for transcriptome-wide SNP callings in both coding and non-coding regions of the remaining 63 samples. Since ploidy levels were unknown for some of the samples, all samples were treated as diploids. While this approach should not significantly affect the SNP calling in autopolyploids, it is likely to cause some allele drop-outs in allopolyploids. Yet, compared to the general drop-out caused by the low coverage of the sequencing data, this effect will probably be small and preliminary tests using ploidy settings adjusted to respective (known) ploidy levels (data not shown) did not significantly improve the respective SNP callings with regard to the number of called heterozygous sites.

The initial SNP calling revealed 1,250,109 raw SNPs. We first used vcftools to remove variants with a minor allele count less than 3, thereby excluding sequencing errors and private SNPs which are undistinguishable in low sequencing depth data. The remaining 492,531 variant sites were filtered using GATKs *VariantFiltration* according to the GATK Best Practices quality recommendations (QD<2.0 || FS>60.0 || MQ<40 || MQRankSum<−12.5 || ReadPosRankSum<−8.0) and keeping only biallelic sites. This resulted in a total of 447,919 hard-filtered variant sites (all samples).

## STRUCTURE analysis

Genetic clustering within the data set was investigated using STRUCTURE version 2.3.4 (*Pritchard et al., 2000*; *Falush et al., 2003*). The analysis was restricted to the 62 *Cochlearia* samples and so *I. megalospermum* was excluded from the generated 'all samples' vcf file, leaving 400,071 variants to be analyzed. STRUCTURE was run with a burn-in of 5000 cycles followed by 5000 iterations per run under an admixture model. We performed 10 runs for each K from K=1 to K=10 and every subset was analyzed two times in order to test both the *correlated* and the *independent allele frequencies* model. The *correlated allele frequencies* model is supposed to be more sensitive to discrete population structure, yet it might possibly lead to over-estimates of K (*Pritchard et al., 2010*). Therefore, both models were tested and compared. We used the structure-sum script (*Ehrich, 2006*) in R to infer the optimal number of clusters for each analysis according to the Evanno method (*Evanno et al., 2005*). Results of the different STRUCTURE analyses were processed using the python script structureHarvester.py version 0.6.94 (*Earl and vonHoldt, 2011*) and CLUMPP version 1.1 (*Jakobsson and Rosenberg, 2007*) was used to summarize replicate runs of the optimal *K*.

As an example for the evolution of a putatively allopolyploid species within the genus *Cochlearia*, we performed a separate STRUCTURE analysis for *Cochlearia bavarica* and its putative parental species *C. pyrenaica* and *C. officinalis*. We therefore selected the two *C. bavarica* samples as well as three samples of *C. pyrenaica* and *C. officinalis* respectively from the 'all samples' vcf file using GATK's *SelectVariants* tool. After respective SNP filtering steps, 103,874 variant sites remained, and two STRUCTURE analyses (*correlated/independent allele frequencies*) were performed with settings as described above for K from K=1 to K=6. The optimal K was then determined as described above.

## RAxML analysis

To further analyze phylogenetic relationships based on the nuclear genomic data, we performed an ML tree reconstruction with RAxML version 8.1.16 (*Stamatakis, 2014*). In order to run RAxML with an ascertainment bias correction, ambiguous sites had to be removed from the 'all variants' vcf file, resulting in 298,978 remaining variant sites. JModelTest version 2.1.10 was utilized to determine the

best-fit nucleotide substitution model. The RAxML analysis was carried out under the GTR+ Γ substitution model with 1000 rapid bootstrap replicates, and FigTree version 1.4.1 (*Drummond et al., 2012*) was used for visualization of the best final ML tree.

## SplitsTree analysis

Aside from the ML tree search, we used SplitsTree version 4.15.1 (*Huson, 1998*; *Huson and Bryant, 2006*) to investigate conflicting or reticulate phylogenetic relationships. The input file was generated from the 'all samples' vcf file (447,919 hard-filtered variant sites) using the Python script 'vcf2phylip' (*Ortiz, 2019*). We used the NeighborNet algorithm based on uncorrected p-distances and equal angles to compute the split network.

## TreeMix analysis

Moreover, we used TreeMix version 1.13 (*Pickrell and Pritchard, 2012*) to further investigate the historic relationships among *Ionopsidium* and the analyzed *Cochlearia* accessions. The software is designed for the estimation of population trees with admixture, yet it can also be used with only a single individual representing a population. We therefore turned off the correction for sample size (-noss) as this could lead to an overcorrection with only one sample per population. The analysis was based on 447,919 hard-filtered variant sites (see above) and we allowed for 0–10 migration edges (m). *Ionopsidium* was set as root and we used a SNP window size of 100 (-k) for all analyses. 10 initial runs were performed for every m and the R package 'optM' (*Fitak, 2021*) was used to identify the optimal number of migration events based on the Evanno method (*Evanno et al., 2005*). For the best m, we then performed 100 TreeMix runs as bootstrap replicates and a consensus tree was inferred from the generated 100 ML trees using the program SumTrees version 4.10 (*Sukumaran and Holder, 2010*).

## ABC modeling—demographic history of diploid Arctic and central European *Cochlearia*

The genetic history of diploid lineages of *Cochlearia* from the Arctic (ARC) and central European (EUR) distribution ranges was analyzed in a coalescent modeling framework with Approximate Bayesian Computation (ABC; *Tavaré et al., 1997*; *Beaumont et al., 2002*). We first evaluated a set of four models differing in the history of gene flow (see *Figure 3c*), and in a second step fitted parameters for relevant models.

Observed data were prepared by filtering all individuals for read coverage with a minimum coverage of 4× and an upper coverage cutoff of 2%. In order to restrict the data to sites that follow a model of neutral evolution as close as possible, only silent (fourfold degenerate) sites were retained. These included bi-allelic SNPs as well as monomorphic sites, to characterize neutral genetic variation in an absolute sense. Contigs with any SNPs displaying excessive heterozygosity (Hardy-Weinberg test; *Danecek et al., 2011*) indicative of paralogs were excluded. This resulted in a data set with 22 diploid EUR individuals, 12 diploid ARC individuals, and 6387 sites located in 5601 independent contigs with a minimum length of one base pair. Summary statistics for the observed data were calculated with the same code as used for simulations.

Coalescent simulations were carried out in a custom pipeline based on the work by *Roux et al., 2013*, using the simulator msnsam (*Hudson, 2002*; *Ross-Ibarra et al., 2008*). We used a set of 114 population genetic summary statistics, among them a folded two-dimensional site frequency spectrum (*Gutenkunst et al., 2009*).

The sampling scheme of observed data, that is, number of samples, contigs, and their lengths, was replicated identically in the simulations. Additional fixed parameters were the mutation and recombination rates, which were both set to $6.51548*10^{-9}$ per site per generation, the silent site mutation rate for Brassicaceae (*De La Torre et al., 2017*). Although we truncated the original contigs to their silent sites only, recombination rates (but not mutation rates) were specified such that they reflected the original contig lengths with all sites. Free parameters of the four models were sampled from uniform prior distributions and are listed in *Supplementary file 18*. All of these models put the history of effective population sizes of ARC and EUR populations in two phases, one ancient phase during which Ne was constant, and one recent phase in which Ne either remained the same as before, or exponentially grew or exponentially declined toward the present.

To evaluate the observed data against four alternative models, we used the R package 'abcrf' (*Pudlo et al., 2016*) with 40,000 simulations per model and 1000 trees per random forest, subsampling 100,000. Whereas traditional rejection ABC model choice approximates models posterior probability based on the relative frequency of simulations under the model that are similar to the observed data, the random forest ABC (ABC-RF) posterior probability is an estimate of the classification error, that is, the probability that the resulting classification is correct. Model parameters were also estimated using 'abcrf' with 10,000 simulations and separate regression random forests (500 trees) for each parameter. Goodness of fit was evaluated by posterior predictive checks, with 40,000 new simulations generated from the full approximated posterior distributions, and the standardized Euclidean distances between observed data and posterior simulations as done by the rejection-ABC function from the R package 'abc' (*Csilléry et al., 2012*).

## GC-MS primary metabolite profiling
### Plant material and taxon sampling for metabolite profiling
For the GC-MS-based metabolite profiling, we selected 14 *Cochlearia* taxa from a total of 28 populations/accessions representative of the total ecotypic variation within the genus (*Supplementary file 11*). Additionally, five populations/species of the genus *Ionopsidium* were included as an outgroup. Whenever possible, at least four plants per accession were analyzed but, in few cases, smaller sample sizes were accepted due to limited seeds/living plant material. A total of 141 plants, either collected in the wild or grown from seed material was considered for metabolomic analyses. Prior to the experiments, all plants were cultivated under greenhouse conditions in a substrate consisting of seedling potting soil, quartz sand, and either composted earth or a peat-based substrate (Ökohum GmbH, Herbertingen, Germany).

## Habitat characterization and ecotype definition
For all 28 *Cochlearia* populations, bioclimatic variables were downloaded from the high-resolution climate data WorldClim grids (http://www.worldclim.org; *Hijmans et al., 2005*) at a resolution of 30″ (~1 km²/pixel). Since geographical coordinates were not available for most of the *Ionopsidium* accessions included in the metabolite profiling, we excluded *Ionopsidium* from the habitat characterization and instead treated it as a Mediterranean ecotype in downstream analyses. For *Cochlearia*, all 19 standard topo-climatic variables were selected for PCA using SPSS version 28 (IBM Corp, Armonk, NY) (SupplFig20_SourceData1). A correlation table for all variables was computed (*Supplementary file 19*) and the significance was tested that the correlation matrix is different from an identity matrix using a Kaiser-Meyer-Olkin test (*Kaiser, 1970*; *Supplementary file 12*). We further ran a PCA using the first temperature-related 11 WorldClim variables only (SupplFig21_SourceData1). Based on congruent results in both PCA recognizing four clusters, 9 out of these 19 standard topo-climatic variables were selected for hierarchical cluster analysis in order to assign the populations to common climatic ecotypes (*Figure 4—source data 1*). PCA scree plots were generated using MVSP 3.2 (Kovach Computing Services, Anglesey, UK).

Based on our field and cultivation experiences, these nine climatic variables are the most important for *Cochlearia* growth and survival rate and include the temperature-related bioclimatic variables Annual Mean Temperature (BIO1), Max Temperature of Warmest Month (BIO5), Min Temperature of Coldest Month (BIO6), Mean Temperature of Warmest Quarter (BIO10), and Mean Temperature of Coldest Quarter (BIO11) as well as four temperature/precipitation-related variables, namely Mean Temperature of Wettest Quarter (BIO8), Mean Temperature of Driest Quarter (BIO9), Precipitation of Warmest Quarter (BIO18), and Precipitation of Coldest Quarter (BIO19).

A hierarchical cluster analysis based on the selected population-specific bioclimatic variables was performed in R version 3.3.1. Prior to scaling and log transformation, a constant value (+150) was added to the original variables in order to avoid negative values. The dist() function was used to compute euclidian distances and a hierarchical clustering was performed using hclust() ('stats' package) according to the Ward method ('ward.D2;' *Ward Jr JH, 1963*; *Murtagh and Legendre, 2014*) which has been applied for climate data cluster analyses before (*Unal et al., 2003*). The best number of clusters in the data set was evaluated using the package 'NbClust' (*Charrad et al., 2014*) with all 26 indices being computed, and the final cluster dendrogram was visualized using the R package 'factoextra' (*Kassambara and Mundt, 2016*).

## Temperature treatment

The metabolomic experiments were performed in two rounds (first batch: 27 *Cochlearia* accessions; second batch: 1 *Cochlearia* accession, 5 *Ionopsidium* accessions) with identical experimental setup (*Figure 4c*). After an initial phase of acclimatization for two weeks in a plant room with a 16/8 hr day/night, 20°C/18°C day/night cycle, leaf samples were harvested from all 141 plants, frozen in liquid nitrogen and transferred to –80°C until metabolite extraction. Hereafter, a cold treatment was performed on half of the plants of each accession in a cold chamber with a 16/8 hr day/night, 5°C day/night cycle, while the remaining plants stayed under control conditions. Leaves from all plants (cold and control) were collected again after 20 days and treated as described above.

## GC-MS-based metabolite profiling

Metabolite profiling was performed using GC-MS and primary metabolite extraction and analysis steps as described by *Roessner et al., 2001*. Briefly, 15–40 mg of the previously collected and frozen leaf material were homogenized by grinding in liquid nitrogen and hereafter mixed with 360 µl cold methanol. 20 µg of ribitol were added as an internal normalizing standard. After extracting the sample for 15 min at 70°C, it was mixed thoroughly with 200 µl chloroform and 400 µl water and centrifuged subsequently. 200 µl of the methanol-water upper phase containing polar to semi-polar metabolites were collected and concentrated to dryness in a vacuum concentrator. A two-step derivatization procedure including methoximation of the dried residue followed by silylation was performed (*Lisec et al., 2006*). To this end, the residue was first re-suspended in a methoxyamine-hydrochloride/pyridine solution for a methoxymization of the carbonyl groups. The sample was then heated for 90 min at 37°C and further silylated with N-methyl-N-trimethylsilyltrifloracetamide at 37°C for 30 min.

GC-MS analysis was performed on a gas chromatograph system equipped with quadrupole mass spectrometer (GC-MS-QP2010, Shimadzu, Duisburg, Germany). For this, 1 µl of each sample was injected in split mode with a split ratio of 1:20 and the separation of derivatized metabolites was carried out on a RTX-5MS column (Restek Corporation, Bellefonte, PA). Metabolites were detected using optimized instrumental settings (*Lisec et al., 2006*).

## GC-MS data processing

A two-pronged approach was employed for metabolite annotation. Briefly, obtained raw data files were first converted into an ANDI-MS universal file format for spectrum deconvolution and compound identification via the reference collection of the Golm Metabolome Database (GMD, http://gmd.mpimp-golm.mpg.de/) using the AMDIS program (Automated Mass Spectral Deconvolution and Identification System; https://www.amdis.net/). Kovats retention indices were calculated for deconvoluted mass spectra from measurements of an alkane mixture and hereafter compared with best hits obtained via the GMD database. The Shimadzu GCMS solutions software (v2.72) interface was further used for manual curation of metabolite annotation versus an in-house library of authentic standards analyzed under the above analytical conditions.

CSV output files were exported for each measurement batch with peak areas obtained for quantifier ions selected for a set of 40 compounds (annotated as known compounds by the above annotation approach or considered as unknown compounds) consistently detected in all analyzed samples. Peak areas (*Supplementary file 14*) were scaled on a sample-basis according to the extracted amount of leaf tissue and further in percent of the peak area of the ribitol internal standard, the latter to account for putative extraction and analytical performance variations across the different measurement batches. Cross-sample variations of the ribitol peak area did not differ significantly between the different measurement batches (between-batch F=1.998, one-way ANOVA p=0.1586) and relative standard deviation of this internal standard did not differ more than 15% between measurement batches. While not providing absolute quantification information on individual compounds, the normalized compound table allows for cross-condition statistical analysis of metabolite relative changes. To this end, normalized tables were concatenated in one matrix prior to subsequent univariate and multivariate statistics (*Supplementary file 14*).

## Multivariate statistical analysis of metabolite data and integration of bioclimatic population clusters

The aov() function in R was used to perform separate one-way ANOVAs on the normalized metabolite data grouped by the four bioclimatic population clusters and *Ionopsidium* to investigate differences in the compound concentrations between control and cold treatment in the different clusters. To further investigate putative metabolomic differentiation between the clusters, PCAs were carried out on the normalized metabolite data using the dudi.pca() function implemented in R package 'ade4' (*Dray and Dufour, 2007*). We then performed DAPCs (*Jombart et al., 2010*) via the function dapc() embedded in the R package 'adegenet' (*Jombart, 2008*; *Jombart and Ahmed, 2011*). Input data were centered and scaled and prior to each DAPC a cross-validation was performed using the xvalDapc() function of the 'adegenet' package with 1000 replicates in order to find the optimal number of PCs to retain. The latter was determined from the lowest root mean squared error associated with the predictive success. Group priors were first defined by temperature treatment in order to analyze the general response to the different conditions. For further DAPCs, group priors were set to represent the four bioclimatic clusters and *Ionopsidium*. DAPCs were performed separately for compound data from the second measurement of either control plants (20°C) or cold treated plants (4°C). All discriminant functions were retained and DAPC results were visualized using the R function scatter() from the 'ade4' package.

## Acknowledgements

The authors thank our gardeners Thorsten Jakob, Bärbel Schwarz, and Frank Korn for excellent plant curation. Peter Sack, Lisa Kretz, Florian Michling, and Lua Lopez are acknowledged for assistance as is David Ibberson at Heidelberg Deep Sequencing Core facility and Rainer Schulz for help with cytogenetic compilations. This work was supported by the German Research foundation (DFG; KO2302/13 and KO2302/16 to MAK).

## Additional information

### Funding

| Funder | Grant reference number | Author |
| --- | --- | --- |
| Deutsche Forschungsgemeinschaft | KO2302/13 | Marcus A Koch |
| Deutsche Forschungsgemeinschaft | KO2302/16 | Marcus A Koch |

The funders had no role in study design, data collection and interpretation, or the decision to submit the work for publication.

### Author contributions

Eva Wolf, Data curation, Formal analysis, Investigation, Methodology, Validation, Visualization, Writing – original draft, Writing - review and editing; Emmanuel Gaquerel, Mathias Scharmann, Formal analysis, Investigation, Methodology, Validation, Writing – original draft; Levi Yant, Writing – original draft, Writing - review and editing; Marcus A Koch, Conceptualization, Data curation, Formal analysis, Funding acquisition, Investigation, Methodology, Project administration, Resources, Supervision, Validation, Visualization, Writing – original draft, Writing - review and editing

### Author ORCIDs

Emmanuel Gaquerel  http://orcid.org/0000-0003-0796-6417
Mathias Scharmann  http://orcid.org/0000-0001-8523-6888
Marcus A Koch  http://orcid.org/0000-0002-1693-6829

### Decision letter and Author response

Decision letter https://doi.org/10.7554/eLife.71572.sa1
Author response https://doi.org/10.7554/eLife.71572.sa2

## Additional files

**Supplementary files**

• Appendix 1—figure 2—source data 1. Geographical distribution of ploidy levels in documented Cochlearia accessions.

• Appendix 1—figure 4—source data 1. Measured 2C values (in picograms) of all analyzed Cochlearia samples as used for rank correlation analyses (2C value / total chromosome number).

• Appendix 1—figure 4—source data 2. Measured 2C values (in picograms) of Cochlearia samples excluding short-lived arctic diploids and the annual C. danica as putative outlier samples as used for rank correlation analyses (2C value / total chromosome number).

• Appendix 1—figure 6—source data 1. Best-scoring ML tree based on (nearly) complete plastid genomes and generated in RAxML with 1000 rapid bootstraps.

• Appendix 1—figure 7—source data 1. Best-scoring ML tree based on partial mitochondrial genomes and generated in RAxML with 1000 rapid bootstraps.

• Appendix 1—figure 8—source data 1. Tanglegram nexus-file (plastome versus mitochondrial sequence data).

• Appendix 1—figure 12—source data 1. Best-scoring ML tree based on transcriptome-wide nuclear SNPs (298,978 variant sites) and generated in RAxML with an ascertainment bias correction and 1000 rapid bootstraps.

• Appendix 1—figure 13—source data 1. STRUCTURE result C. bavarica at K=2 (uncorrelated allele frequency model).

• Appendix 1—figure 13—source data 2. STRUCTURE result C. bavarica at K=2 (correlated allele frequency model).

• Appendix 1—figure 19—source data 1. Bootstrapped (100 bootstrap replicates) maximum likelihood consensus tree generated with TreeMix with one migration/admixture event.

• Appendix 1—figure 20—source data 1. 19 WorldClim bioclimatic variables for 28 Cochlearia accessions.

• Appendix 1—figure 21—source data 1. 11 WorldClim bioclimatic variables for 28 Cochlearia accessions.

• Appendix 1—figure 22—source data 1. Detailed summary output of one-way ANOVAs for all metabolites for bioclimatic clusters (1–4) and Ionopsidium (5).

• Appendix 1—figure 30—source data 1. Metabolite contributions to discriminant function number 1 (separating control and cold conditions) in DAPC analysis as illustrated in *Figure 4d* (based on all metabolomic measurements and grouped by treatment) sorted from highest to lowest discriminating power.

• Supplementary file 1. *Cochlearia* species list. List of accepted *Cochlearia* taxa according to *BrassiBase* (*Kiefer et al., 2014*), in alphabetical order together with information on cytotype, ploidy level, ecology and distribution area.

• Supplementary file 2. Survey of georeferenced cytogenetic data.

• Supplementary file 3. Results of chromosome counting and/or flow cytometry measurements. 2C values are given in picograms (pg). Asterisks (*) indicate progeny from open pollination. Chromosome counts marked with 'FB' were made on flower buds, generated by Dr. Martin Lysak and Dr. Terezie Mandáková at the Central European Institute of Technology (CEITEC, Brno).

• Supplementary file 4. Publications considered for cytogenetic literature review. List of publications (and literature cited therein) screened for *Cochlearia* chromosome counts and/or genome size measurements.

• Supplementary file 5. Rank Correlation Analyses. (1) Correlation between 2C value and chromosome number (full data set, 78 accessions analyzed in total, including putative outliers), (2) Correlation between 2C value and chromosome number excluding Arctic diploids and *C. danica* as putative outliers (59 individuals, 29 accessions, 11 taxa analyzed in total), (3) Correlation between DNA content per chromosome and total chromosome number (full data set, 78 accessions analyzed in total, including putative outliers), (4) Correlation between DNA content per chromosome and total chromosome number excluding Arctic diploids and *C. danica* as putative outliers (59 individuals, 29 accessions, 11 taxa analyzed in total).

• Supplementary file 6. Detailed information on NGS data output and data analyses.

• Supplementary file 7. Information on accessions selected for Approximate Bayesian Computation (ABC). For detailed information on selected accessions see *Supplementary file 6*.

• Supplementary file 8. Selected population genetic statistics observed for EUR and ARC populations of diploid *Cochlearia*. Each statistic is an average over 5601 independent contigs. These statistics together with 103 additional summary statistics formed the basis of the coalescent modeling analyses.

• Supplementary file 9. Results of model choice via ABC-RF for four models differing in the history of gene flow. Models explored for the coalescent modeling framework: SI = strict isolation, CM = continuous migration from the population split to the present, OSC = ongoing secondary contact with gene flow starting after population split and continuing up to the present, PSC = past secondary contact with gene flow starting after population split and stopping before the present. The effect of the choice of prior is shown by two different upper bounds of the effective population sizes.

• Supplementary file 10. Parameter estimates for the SI and OSC models with two choices of priors for the effective population sizes (upper bound of N, $N_{max}$). The upper part shows medians with 5th and 95th percentiles of the approximated posterior distributions. Population sizes (N) in thousands, time (t) in thousands of generations, and migration rates in log10 of the proportion of immigrants per generation. The lower part shows goodness of fit of these parameter estimates from posterior predictive checks (PPCs), as the standardized euclidean distances between observed summary statistics and predicted values.

• Supplementary file 11. Accessions included in the metabolomic analyses. Where possible, missing coordinates were extracted (c.e.) based on information of the respective locality, otherwise coordinates are marked as n/a.

• Supplementary file 12. SPSS output for KMO and Bartlett's Test.

• Supplementary file 13. List of 40 selected metabolic compounds. Compounds were selected for integration based on annotatability (via the reference collection of the Golm Metabolome Database [GMD, http://gmd.mpimp-golm.mpg.de/] using the AMDIS program [Automated Mass Spectral Deconvolution and Identification System; https://www.amdis.net/]) and reproducibility of peak detection throughout the data set. Metabolites are sorted by general compound classes.

• Supplementary file 14. Metabolic compound matrix (raw/normalized).

• Supplementary file 15. Standard plants used for flow cytometric analyses. Respective 2C-values for *Raphanus sativus*, *Solanum lycopersicum* L. cv. 'Stupické polní rané' and *Glycine max* (L.) MERR. cv. '

Polanka' as given in *Dolezel et al., 2007*.

• Supplementary file 16. Flow cytometry summary table.

• Supplementary file 17. Information on mitochondrial DNA sequence reference contigs. Eight mitochondrial de novo consensus sequences generated from sample Cmica_0979 (*Cochlearia micacea*) and chosen as reference sequences for mitochondrial genome mappings, with information on contig length and annotated gene content.

• Supplementary file 18. Prior distributions of the explored ABC models. SI = strict isolation, CM = continuous migration from the population split to the present, OSC = ongoing secondary contact with gene flow starting after population split and continuing up to the present, PSC = past secondary contact with gene flow starting after population split and stopping before the present. Population sizes (N) in natural units, time (t) in number of generations ago, and migration rates (m) as the fraction of the recipient population made up of immigrants per generation. N parameters were also explored with a higher upper bound, 3 million.

• Supplementary file 19. SPSS Correlation Matrix of bioclimatic variables. Correlation table (generated using SPSS version 28) for all 19 standard topo-climatic variables as downloaded for 28

*Cochlearia* accessions from the high-resolution climate data WorldClim grids at a resolution of 30″ (~1 km²/pixel).

• Transparent reporting form

### Data availability

All data has been submitted and uploaded to: GeneBank: Genomic data - https://www.ncbi.nlm.nih.gov/bioproject/PRJEB21320, and plastomes under LT629868 - LT629930 and LN866844 - LN866848 Cytogenetic data: FlowRepository under identifier FR-FCM-Z3FY DRYAD: Mitochondrial consensus sequences as well as metabolite profiling data and input files for NGS data analyses are available at Dryad https://doi.org/10.5061/dryad.fbg79cnsn.

The following dataset was generated:

| Author(s) | Year | Dataset title | Dataset URL | Database and Identifier |
|---|---|---|---|---|
| Koch MA | 2020 | The evolutionary history of the genus Cochlearia | https://www.ncbi.nlm.nih.gov/bioproject/PRJEB21320 | NCBI BioProject, PRJEB21320 |
| Koch MA | 2021 | Data from: Evolutionary footprints of a cold relic in a rapidly warming world | http://dx.doi.org/10.5061/dryad.fbg79cnsn | Dryad Digital Repository, 10.5061/dryad.fbg79cnsn |
| Wolf E, Gaquerel E, Scharmann M, Yant L, Koch MA | 2021 | Ionopsidium megalospermum chloroplast genome, isolate Imega_1776 | https://www.ncbi.nlm.nih.gov/nuccore/LT629930 | NCBI GenBank, LT629930 |
| Wolf E, Gaquerel E, Scharmann M, Yant L, Koch MA | 2021 | Cochlearia borzaeana chloroplast genome, complete sequence, isolate Cbor_1063 | https://www.ncbi.nlm.nih.gov/nuccore/LN866844 | NCBI GenBank, LN866844 |
| Wolf E, Gaquerel E, Scharmann M, Yant L, Koch MA | 2021 | Ionopsidium acaule chloroplast genome, complete sequence, isolate Iacau_1072 | https://www.ncbi.nlm.nih.gov/nuccore/LN866848 | NCBI GenBank, LN866848 |
| Wolf E, Gaquerel E, Scharmann M, Yant L, Koch MA | 2021 | Cochlearia borzaeana chloroplast genome, complete sequence, isolate Cbor_1063 | https://www.ncbi.nlm.nih.gov/nuccore/LT629868 | NCBI GenBank, LT629868 |

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

## Appendix 1

### Insights into the evolution of polyploid *Cochlearia* taxa

Previous studies have indicated that the genus *Cochlearia* has a complex and putatively reticulate evolutionary background (*Koch et al., 1996*; *Koch et al., 1998*; *Koch et al., 1999*; *Koch, 2002*; *Koch et al., 2003*; *Kochjarová et al., 2006*; *Cieslak et al., 2007*) which is mirrored today in the cytogenetic and ecotypic complexity of the species complex. Since morphological differentiation is often insufficient for species identification within *Cochlearia*, the different taxa are ideally characterized by ploidy level and ecotype. A summary of previous evolutionary hypotheses regarding ploidy levels and ecology as revealed from previous studies, is given with *Appendix 1—figure 1*. Based on the two different base chromosome numbers (n=6 and n=7), a range of ploidy levels from diploid to octoploid has evolved, and although the evolution of polyploid taxa within the genus *Cochlearia* has been studied intensively, especially based on cytogenetic analyses (*Gill, 1973*; *Gill, 1976*; *Gill et al., 1978*; *Koch et al., 1996*; *Nordal and Laane, 1996*; *Koch et al., 1998*; *Koch, 2002*; *Kochjarová et al., 2006*), many aspects are still unknown and require further investigation.

The polyploid taxa can be categorized into coastal and inland polyploids. Along the coasts of continental Europe and Great Britain, we find tetraploid, hexaploid, and octoploid species. According to BrassiBase (*Kiefer et al., 2014*), these are *C. officinalis* L. (2n=4x=24), *C. scotica* Druce (2n=4x=24), *C. danica* L. (2n=6x=42), and *C. anglica* L. (2n=8x=48 or 2n=10x=60) as well as three subspecies of *C. officinalis*, namely *C. officinalis* subsp. *integrifolia* (Hartm.) Nordal & Stabbetorp (2n=4x=24), *C. officinalis* subsp. *norvegica* Nordal & Stabbetorp (2n=4x=24) and *C. officinalis* subsp. *officinalis* L. (2n=4x=24) (see *Supplementary file 1*). Their ecological preferences (e.g., regarding salt tolerance) are slightly differint, yet distribution ranges are often overlapping and many cases of hybridization between coastal taxa have been observed even across ploidy levels (*Saunte, 1955*; *Fearn, 1977*; *Pegtel, 1999*). The near absence of interspecific fertility barriers observed throughout the genus has also been demonstrated experimentally (*Crane and Gairdner, 1923*; *Gill, 1971a*; *Gill, 1971b*; *Gill, 1973*; *Gill, 1976*; *Gill et al., 1978*). In combination with a high level of morphological plasticity in response to the habitat type (*Elkington, 1984*; *Nordal, 1988*), this makes the coastal species complex very challenging from a taxonomical point of view.

The group of inland polyploids found in continental Europe comprises *C. bavarica* Vogt (2n=6x=36), *C. polonica* Fröhl. (2n=2x=36), *C. tatrae* Borbás (2n=6x=42), and *C. borzaeana* (Com. et Nyár.) Pobed. (2n = 8x=42/48). All of them are highly endangered, represent a present-day locally restricted distribution range and are considered as endemics. Besides these, there are tetraploid taxa found at inland stations in Great Britain. Here, populations with a karyotype of 2n=4x=24 have been described as either inland populations of *C. officinalis* (*Clapham et al., 1990*; *Koch et al., 1998*) or as a separate subspecies *C. pyrenaica* ssp. *alpina* (Bab.) Dalby (e.g. *Gill, 2008*) with closer affinities to the *C. pyrenaica* complex (*Gill, 2008*). In the following, these populations are referred to as *C. pyrenaica* ssp. *alpina*, abbreviated as *C. alpina*, while diploid inland populations, which have also been described as *C. alpina* (Bab.) H.C. Watson are treated as *C. pyrenaica* (*Gill, 1971b*). Another tetraploid British inland species is *C. micacea* E.S. Marshall, with a karyotype of 2n=4x=26 (*Gill et al., 1978*; *Clapham et al., 1990*). It is endemic to few high-altitude locations with an alpine character in Scotland.

New insights regarding the evolution of polyploid *Cochlearia* taxa gained in this study, based on both cytogenetic and phylogenomic data, are discussed below for respective taxa. In general, the observed infraspecific variation for chloroplast genomes, as shown before based on cpDNA restriction site variation analysis (RFLP) for several polyploid coastal species *C. officinalis*, *C. danica* and *C. anglica* (*Koch et al., 1996*), and also revealed in the present study by the lack of monophyly in all polyploid coastal taxa, can be best explained by reticulate evolution and multiple origin of these polyploids (*Koch et al., 1996*).

*Cochlearia officinalis* (2n=4x=24), *C. scotica* (2n=4x=24)

The tetraploid *Cochlearia officinalis* s. str. is a biennial to perennial species widely distributed along the northern coastlines of continental Europe and Great Britain, where it is mainly found on gravelly beaches, sea cliffs, or drier areas of salt-marshes (*Gill et al., 1978*; *Nordal and Stabbetorp, 1990*) and rarely in past mining areas on heavy metal contaminated soils along creeks

(*Koch et al., 1998*). Whereas former studies suggested an autopolyploid origin from relatives of the only recent Central European diploid coastal species, *C. aestuaria* (*Gill et al., 1978*; *Elkington, 1984*; *Koch et al., 1998*), the results presented herein are not in favor of this hypothesis. All samples of *C. officinalis* included in a STRUCTURE analysis of 62 *Cochlearia* samples (based on 400,071 transcriptome-wide SNPs, K=3) show signatures of admixture between the Arctic and the Central European gene pool (*Figure 3*, *Appendix 1—figure 15*), whereas *C. aestuaria* is purely grouped within the Central European gene pool. Likewise, in the SplitsTree analysis of all *Cochlearia* samples and *Ionopsidium* (*Appendix 1—figure 11*), *C. officinalis* was grouped in between diploid samples from the Arctic gene pool and the diploid samples from Northern Spain including *C. aestuaria*. In the plastid phylogeny, *C. officinalis* samples do not form a monophyletic lineage but are found both in a basal lineage of coastal polyploids (*Appendix 1—figure 6*, blue lineage) and in the derived purple lineage, containing mainly samples from Great Britain as well as *C. aestuaria*. Taken together, these results indicate the repeated formation of the coastal tetraploid *C. officinalis* via two plausible scenarios: (a) autopolyploidization from formerly existing northern European coastal diploids with a strong Arctic impact, or (b) allopolyploidization between *aestuaria*-like diploids and northern diploids forced to come into contact likely during past glacial maxima.

Compared to *C. officinalis* s. str., the British tetraploid *C. scotica* (2n=4x=24) has been described to have a more compact phenotype with shorter inflorescences. It has also been considered as a morphological variant of *C. officinalis* (*Gill, 1971a*; *Gill, 1973*) given the overall weak morphological distinctiveness. Whereas one of the *C. scotica* samples included in our study was placed next to Austrian and Swiss diploids in the organellar phylogenies (*Appendix 1—figures 6 and 7*), indicating mechanisms such as chloroplast capture or incomplete lineage sorting, no clear distinction between analyzed samples of *C. officinalis* and the two *C. scotica* samples was possible based on the results of the nuclear transcriptome SNP data analyses via STRUCTURE and SplitsTree. Therefore, *C. scotica* might rather be considered as a small growing morphological variant of *C. officinalis* as suggested before.

*Cochlearia alpina* (2n=4x=24) and *C. micacea* (2n=4x=26)

Based on the different analyses of organellar and nuclear genomes presented in this study, no strong conclusions regarding the evolutionary background of the British tetraploid species *C. alpina* (2n=4x=24) and *C. micacea* (2n=4x=26) can be drawn, but some reasonable speculations can be made.

In both organellar phylogenies, *C. alpina* did not form a monophyletic clade, suggesting repeated formation of this tetraploid inland species. This hypothesis is supported by the scattered placement of *C. alpina* samples in the SplitsTree network (*Figure 2*, *Appendix 1—figure 11*). Whereas Calp_0828 is grouped within British tetraploid samples, Calp_0165 and Calp_1127 show closest associations with diploid *C. pyrenaica*. Therefore, putative events of autopolyploidization from *C. pyrenaica* cannot be ruled out, although the organellar phylogenies also reveal a close relation to the coastal species complex.

The two samples of *C. micacea* analyzed in here, originating from the Scottish mountains Beinn an Dòthaidh (Cmica_0979) and Ben Lawers (Cmica_0983), respectively, are monophyletic in both organellar ML analyses (*Appendix 1—figures 6 and 7*). Both phylogenies as well as the SplitsTree network reveal a close association of *C. micacea* with the coastal species complex and especially with *C. officinalis*. Yet, the species-specific karyotype of 2n=26, as described by *Gill, 1973* and later on questioned by *Nordal and Stabbetorp, 1990*, could not be confirmed for the respective accessions (*Supplementary file 2*). Therefore, the former hypothesis of a formation via primary tetrasomy from autotetraploid *C. officinalis* (*Gill, 1973*) cannot be discussed without further cytogenetic studies.

Based on the results from SplitsTree and STRUCTURE analyses, a clear distinction between *C. alpina* and *C. micacea* is not possible and with regard to the described karyotypic uncertainties, a separate taxonomic status of the two species is questionable (see also *Gill, 2008*). For continental Europe, a formerly wide inland distribution of *C. officinalis*-type plants has been suggested (*Koch, 2002*; see below for *C. bavarica*, *C. polonica*/*C. tatrae*/*C. borzaeana*). Assuming a similar picture for Great Britain, and having in mind the phylogenetic association of both species with *C. officinalis*, it stands to reason to treat both taxa as inland populations of *C.

*officinalis*, as already done before for some *C. alpina* populations (*Clapham et al., 1990*; *Koch et al., 1998*).

*Cochlearia bavarica* (2n=6x=36)

In the existing literature on the inland hexaploid *C. bavarica* (2n=6x=36), endemic to two isolated areas in the Alpine foothills in southern Germany, there is a general consensus about its evolution via allopolyploidization from diploid *C. pyrenaica*-type plants and tetraploid *C. officinalis*-type plants (*Vogt, 1985*; *Koch et al., 1996*; *Koch et al., 1998*; *Abs, 1999*; *Koch, 2002*). A separate STRUCTURE analysis based on 103,874 variants extracted from the variant calling for all samples and including the two sequenced *C. bavarica* samples as well as representatives of the putative parental species further supports this hypothesis (*Appendix 1—figure 13*). Based on the generated divergence time estimates (*Appendix 1—figure 9*), a very recent origin of the species in the 17th/18th century, promoted by the cultivation of *C. officinalis* in Bavarian monasteries as suggested by *Vogt, 1985*, can be ruled out as split times between *C. bavarica* and *C. pyrenaica* samples were dated to postglacial ~10/~11 kya. Therefore, an allopolyploidization event predating anthropogenic impact, from *C. officinalis*-type plants that were formerly distributed inland in Central Europe, as suggested by Koch 002 is more likely. The two *C. bavarica* samples, representing the two recent metapopulations, do not form a monophyletic clade in the organellar phylogenies but are linked to geographically close *C. pyrenaica* samples from respective distribution areas (*Appendix 1—figure 14*). Yet, in the SplitsTree network, the two *C. bavarica* samples are grouped closely together, both connected to a *C. pyrenaica* sample from Baden-Württemberg (Hohenlohe). This result supports the idea of a single ancient origin followed by genetic differentiation over geographic distances and later disruption of the continuous distribution into two refuge distribution areas as suggested bch based on isozyme data showing a strong geographic partitioning.

*Cochlearia polonica* (2n=6x=36), *C. tatrae* (2n=6x=42), *C. borzaeana* (2n=8x=48)

The three polyploid inland species *C. polonica* (2n=6x=36), *C. tatrae* (2n=6x=42) and *C. borzaeana* (2n=8x=48), all endemic to different regions in Eastern Europe, are characterized by different karyotypes, yet similar allopolyploidization events have been suggested for all of them (*Bajer, 1951*; *Koch et al., 1996*; *Koch et al., 1998*; *Koch et al., 2003*; *Kochjarová et al., 2006*). The nuclear genomes indicate a shared evolutionary history and group all them into a monophyletic clade (*Appendix 1—figure 12*). As described for *C. bavarica* (see above), *C. pyrenaica*-like diploids and *C. officinalis*-like tetraploids have been proposed as putative parental species. In the STRUCTURE analysis of all 62 *Cochlearia* samples, the three polyploid taxa show similar patterns of admixture between the European and the Arctic gene pool and with the *C. danica*-specific cluster (*Appendix 1—figure 15*). This is also reflected in the SplitsTree network, where the three taxa are grouped next to the *Ionopsidium*/*C. tridactylites*/*C. danica* cluster (*Appendix 1—figure 11*). In the plastid phylogeny, the three species are dispersed, that way indicating independent polyploidization events (*Appendix 1—figure 6*). Given the basal positions of both *C. tatrae* and *C. borzaeana* in the plastid phylogeny, the latter grouped together with Spanish *C. danica* and Arctic diploids (2n=14), one hypothesis for the formation of these early diverged (presumably last interglacial) European polyploids would be an earlier southward migration of Arctic diploids, that later on went extinct in Europe but served as parental taxa in the presumed allopolyploidization. This would explain the Arctic footprint in the STRUCTURE results, and formerly described affinities between *C. tatrae* and *C. groenlandica* based on chloroplast DNA analyses (*Koch et al., 1996*) support this hypothesis. Still, based on the data presented herein, no final conclusions on the complex reticulate evolutionary history of the three species can be made.

*Cochlearia danica* (2n=6x=42)

Although the annual coastal species *C. danica* (2n=6x=42) shares a common base number of n = 7 with hexaploid *C. tatrae* and the Arctic diploids like the widespread *C. groenlandica*, closer phylogenetic relationships with these species have been rejected by former studies based on chloroplast DNA analyses (*Koch et al., 1996*) or the special cytogenetic characteristics of *C. danica* three chromosome sets of varying size (*Gill, 1976*). Instead, allopolyploid formation from tetraploid *C. officinalis* relatives and *C. aestuaria*-like coastal diploids has been suggested (*Koch et al., 1996*; *Koch et al., 1998*). Our SplitsTree network revealed a close association of *C. danica* with *Ionopsidium* and *C. tridactylites* (*Figure 3*, *Appendix 1—figure 11*). The connection with the

Arctic species complex is further supported by the plastid phylogeny where a Spanish *C. danica* is placed in the second most basal lineage together with *C. borzaeana* (2n=6x=42/48) and the two Arctic diploid species *C. groenlandica* and *C. sessilifolia* (2n=2x=14; see *Appendix 1—figure 6*). However, in the STRUCTURE analysis of all *Cochlearia* samples, *C. danica* forms a clearly separated cluster and only one Scottish sample (Cdan_0654) shows signs of admixture with the Arctic gene pool likely as a result of recent introgression from other coastal species. Thus, the origin of *C. danica* remains somewhat cryptic but an early allopolyploidization event involving n=7 parental taxa with subsequent southward migration along Western European coasts seems likely. A respective genetic coastal gradient from North to South has been demonstrated earlier (*Koch et al., 1998*). Regarding the frequently described naturally occurring hybrids between *C. danica* and other coastal taxa (*Pegtel, 1999*), subsequent hybridization and introgression following the initial formation of the species can be assumed.

*Cochlearia anglica* (2n=8x=48)

The octoploid coastal species *C. anglica* (2n=8x=48), which is distributed from northern France to southern Sweden, was hypothesized to be an auto- or allopolyploid from a *C. officinalis*-like tetraploid (*Koch et al., 1996*). Initially, two accessions of *C. anglica* were included in the present NGS study, yet one accession (Cang_1157, Scotland, north of Creetown) was finally determined to be a putative hybrid between *C. anglica* and *C. officinalis* (*C. x hollandica*) based on chromosome counts and genome size measurements (see *Supplementary file 3*), whereas no cytogenetic data are available for the other accession (Cang_1023, France, Département Morbihan, La Trinité-sur-Mer). Hence, possibly no true *C. anglica* was included in the study and results for Cang_1023 must be handled with caution. The latter shows signatures of admixture with the *C. danica*-specific cluster in the STRUCTURE results (*Appendix 1—figure 15*), which would argue against an autopolyploidization from *C. officinalis* as none of the included *C. officinalis* samples shows this pattern. The association with *C. danica* is further supported by the SplitsTree network (*Appendix 1—figure 11*), yet as stated above, given the described karyotypic uncertainties, no conclusions on the origin of *C. anglica* can be drawn from these results and further studies are needed to verify the formation of this octoploid species.

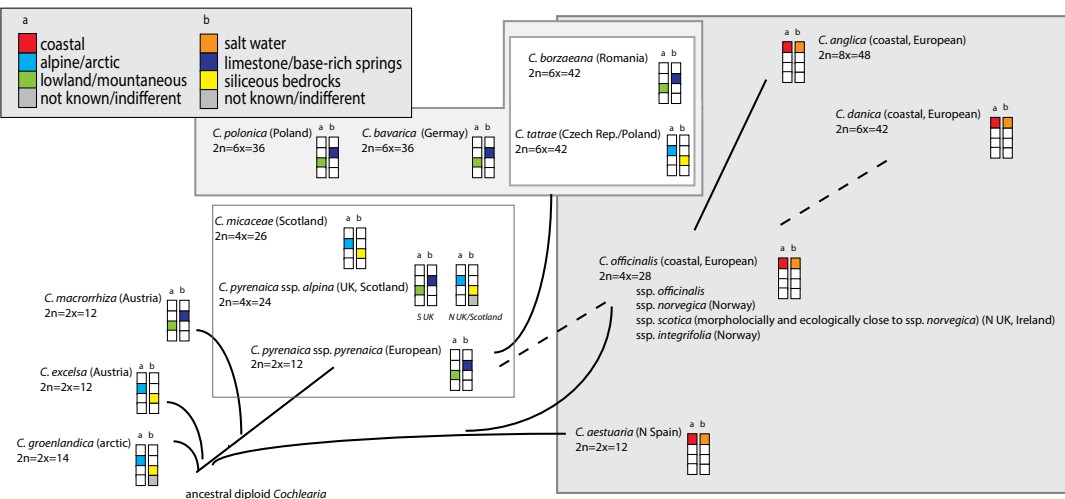

**Appendix 1—figure 1.** Illustration of previous evolutionary hypotheses. A summary of hypotheses regarding the evolution of the genus *Cochlearia* combined with information on ploidy levels and ecology. The latter is given as a combination of (a) species distribution, and (b) substrate specificity.

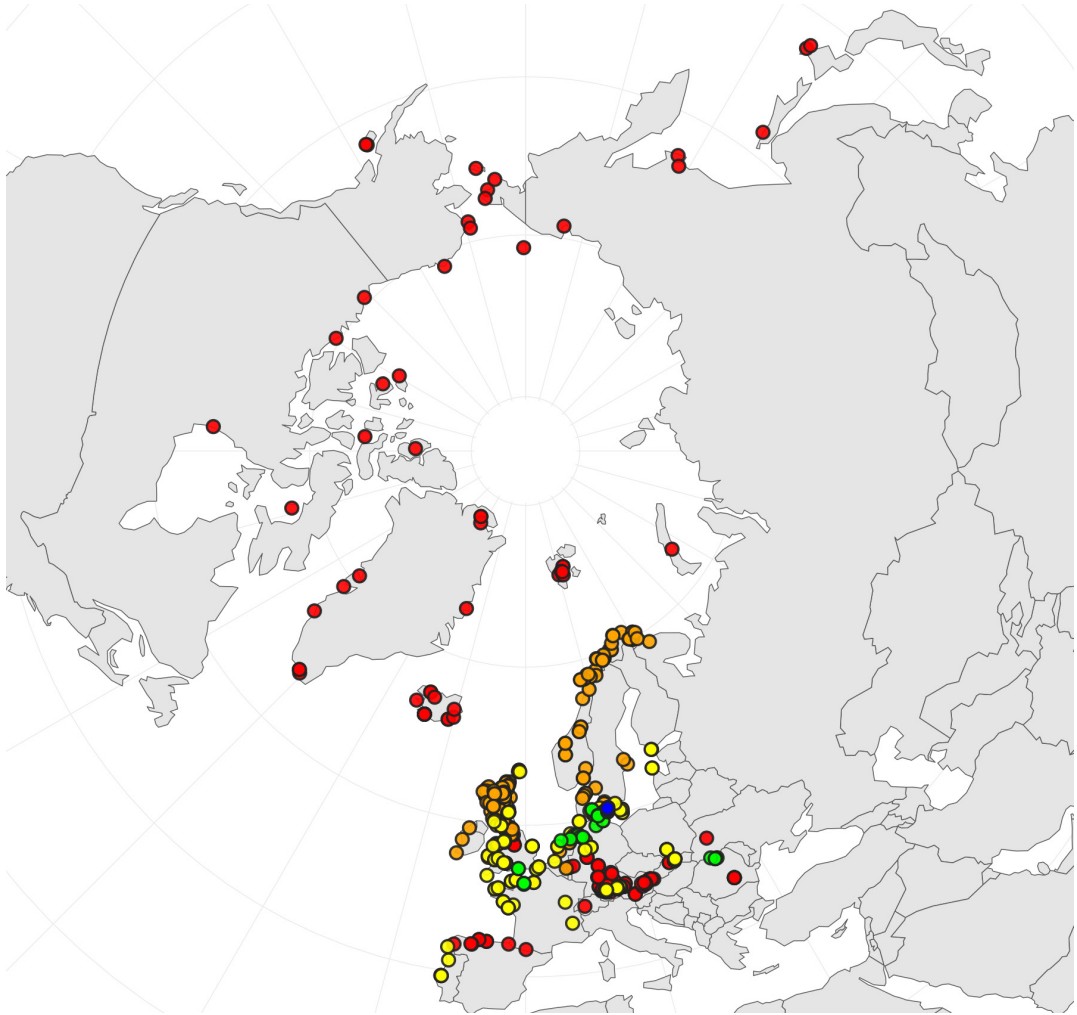

**Appendix 1—figure 2.** Geographic distribution of ploidy levels in *Cochlearia* (n=564) as revealed via a review of published and own chromosome counts (***Supplementary file 2*** and ***Appendix 1—figure 2—source data 1***). Diploids are colored in red, tetraploids in orange, hexaploids in yellow, octoploids in green, and dodecaploids in blue.

The online version of this article includes the following source data for appendix 1—figure 2:

• **Appendix 1—figure 2—source data 1.** Geographical distribution of ploidy levels in documented Cochlearia accessions.

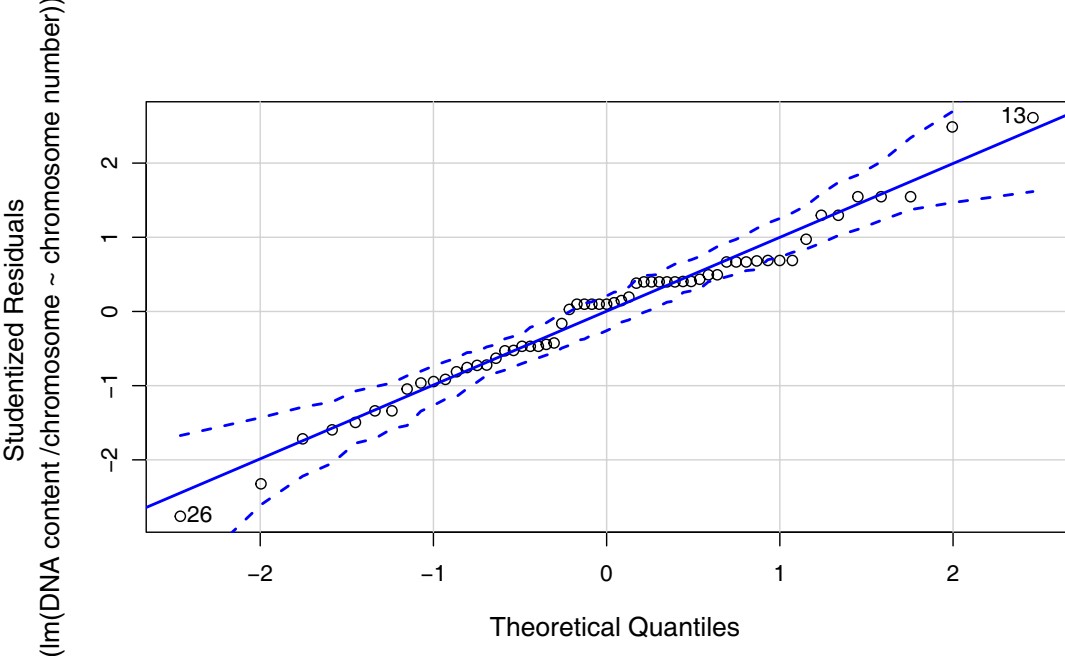

**Appendix 1—figure 3.** Quantile-Quantile Plot generated using the qqPlot() function in R version 4.0.3 (confidence level for point-wise envelope set to 0.99) as a visual check used to assess data normality for linear regression analysis shown in *Figure 1d* (data given with *Figure 1—source data 4*).

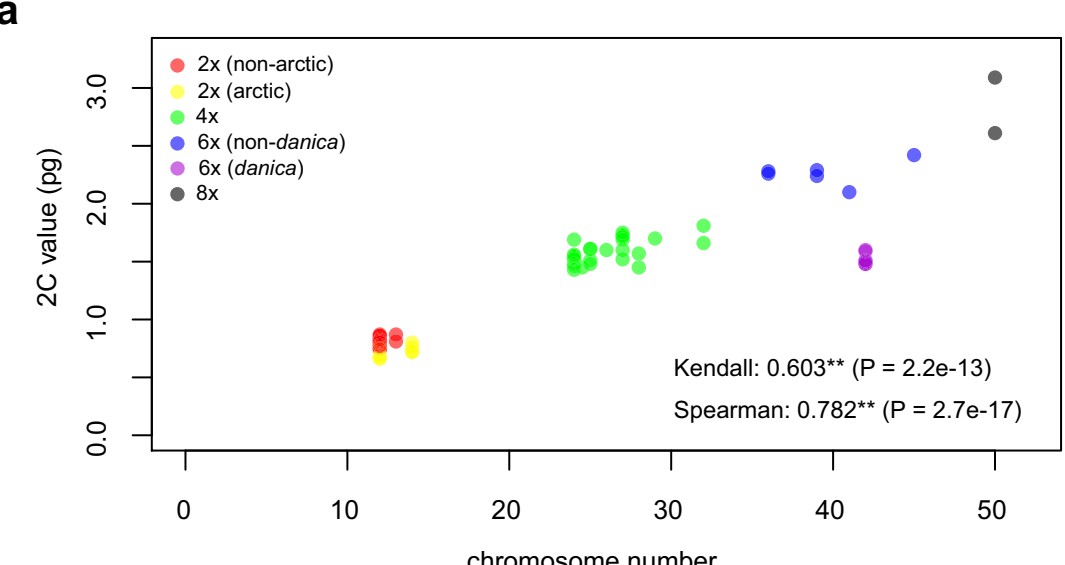

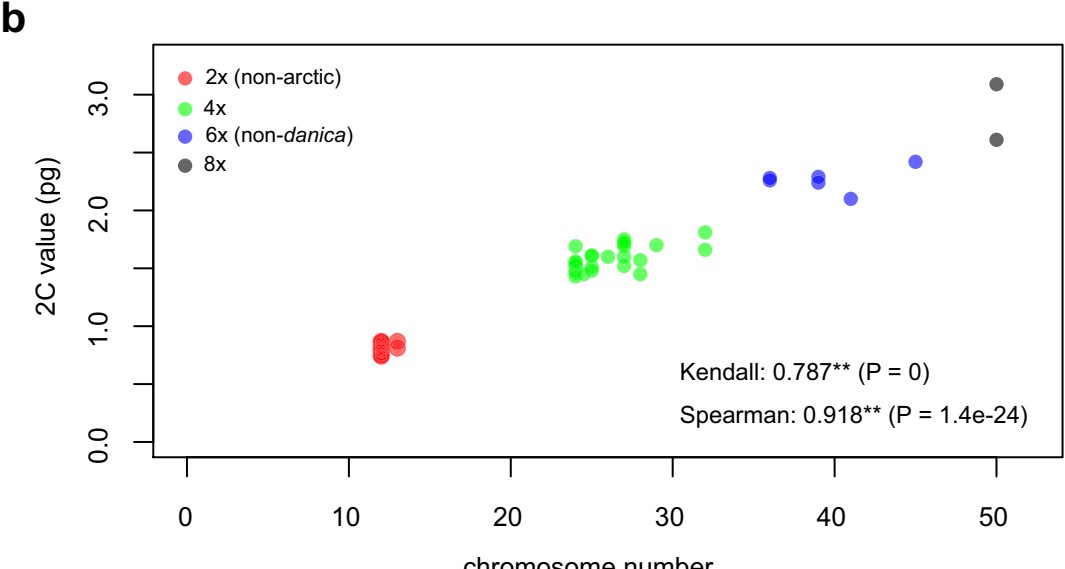

**Appendix 1—figure 4.** Genome size increase with increasing ploidy levels. Measured 2C value (given in picograms) relative to respective total chromosome numbers. (**a**) Full data set (red [21 counts]: 2n=2x (non-Arctic), yellow [nine counts] 2n=2x [Arctic], green [30 counts]: 2n=4x, blue [six counts]: 2n=6x [excluding *C. danica*], purple [10 counts]: 2n=6x [*C. danica*], dark grey [two counts]: 2n=8x) showing a significant increase of genome size with increasing total chromosome numbers as revealed by Kendall and Spearman rank correlation analyses (78 individuals from 38 accessions representing 14 taxa analyzed in total; *Appendix 1—figure 4—source data 1*). (**B**), Data set excluding short-lived Arctic diploids (yellow) and annual *C. danica* (purple) as putative outliers, showing a significant genome size increase with increasing chromosome numbers as revealed by Kendall and Spearman rank correlation analyses (59 individuals, 29 accessions, 11 taxa analyzed in total; *Appendix 1—figure 4—source data 2*).

The online version of this article includes the following source data for appendix 1—figure 4:

• **Appendix 1—figure 4—source data 1.** Measured 2C values (in picograms) of all analyzed Cochlearia samples as used for rank correlation analyses (2C value / total chromosome number).

• **Appendix 1—figure 4—source data 2.** Measured 2C values (in picograms) of Cochlearia samples

excluding short-lived arctic diploids and the annual C. danica as putative outlier samples as used for rank correlation analyses (2C value / total chromosome number).

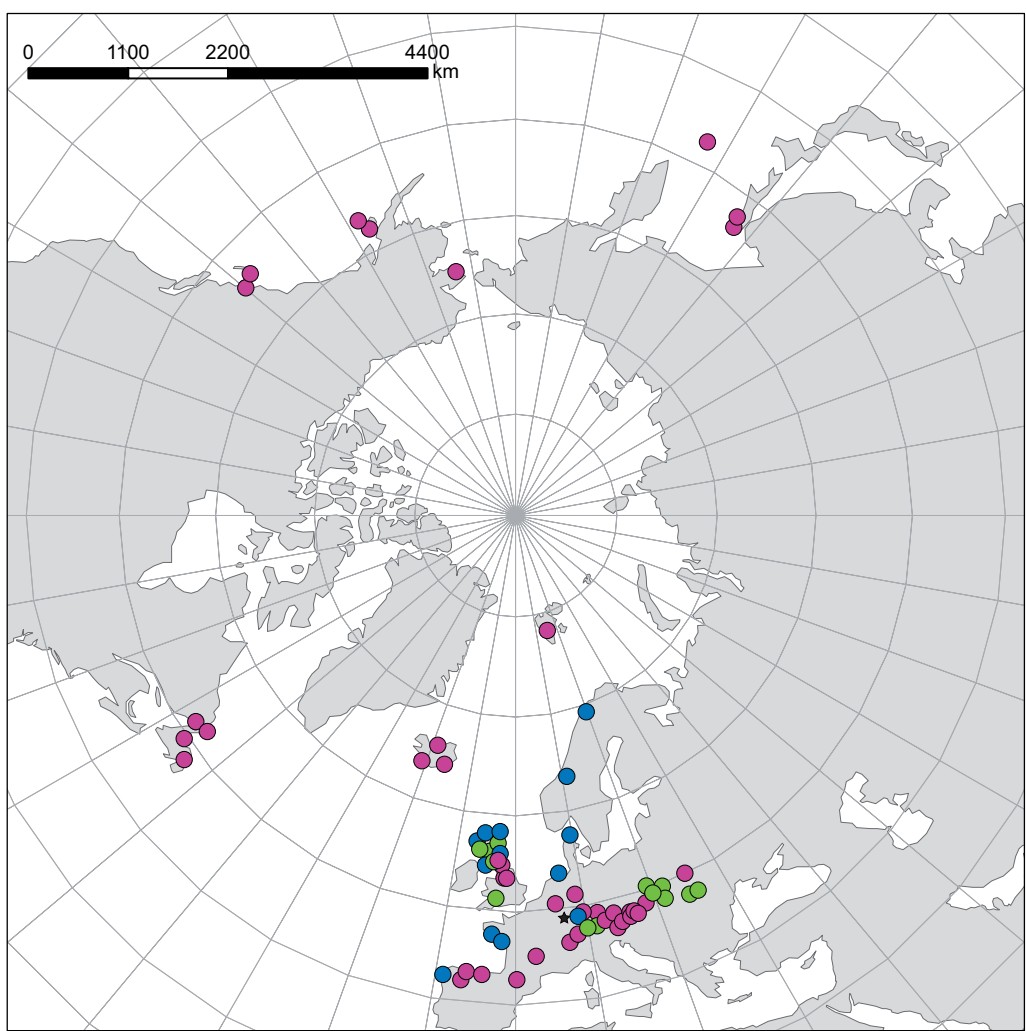

**Appendix 1—figure 5.** NGS samples—geographical distribution and ploidy levels/ecotypes. Geographical distribution of *Cochlearia* NGS samples. Colors indicate different ploidy levels/ecotypes. Diploid samples are shown in pink, polyploid inland samples in green, and polyploid coastal samples in blue. The asterisk marks an inland sample of *C. danica* which is a coastal species rapidly dispersed inland along highways.

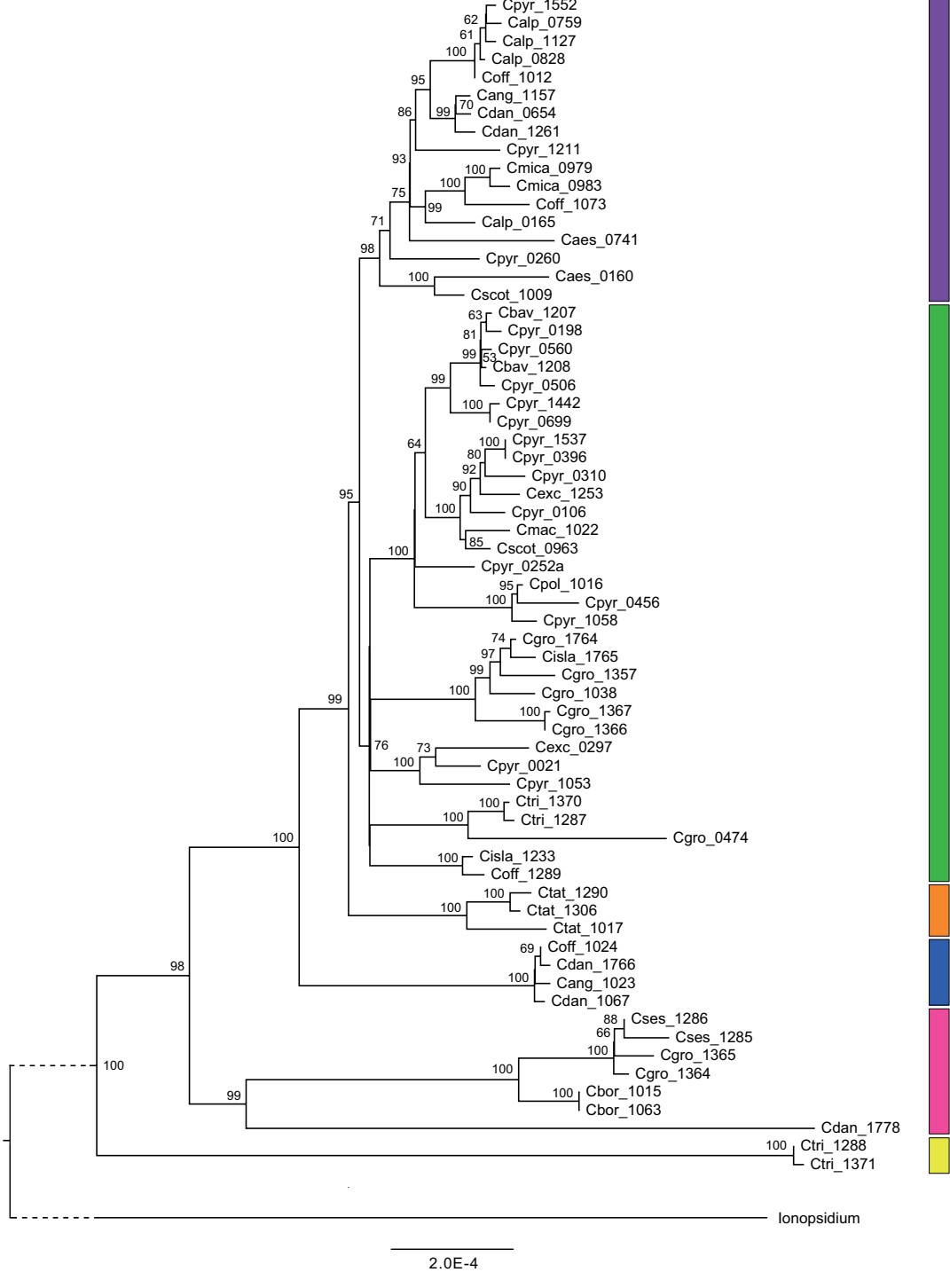

**Appendix 1—figure 6.** Plastid maximum likelihood (ML) phylogeny. ML phylogram based on complete chloroplast genomes from the tribe Cochlearieae generated with RAxML (***Appendix 1—figure 6—source data 1***). Bootstrap support (1000 replicates) above 50% is shown near the respective nodes. For illustration purpose, the *Ionopsidium* outgroup lineage is collapsed and condensed. Evolutionary lineages within *Cochlearia* are displayed as colored bars (corresponding to ***Figure 2***). Taxa included in evolutionary lineages: yellow lineage: *C. tridactylites*; pink lineage: *C. danica, C. borzaeana, C. groenlandica, C. sessilifolia*; blue lineage: *C. danica, 'C. anglica'* (or *C. hollandica), C. officinalis*; orange lineage: *C. tatrae*; green lineage: *C. officinalis, c. islandica, C.*

*Appendix 1—figure 6 continued on next page*

*Appendix 1—figure 6 continued*

*groenlandica, C. tridactylites, C. pyrenaica, C. excelsa, C. polonica, C. scotica, C. macrorrhiza, C. bavarica*; purple lineage: *C. scotica, C. aestuaria, C. pyrenaica, C. alpina, C. officinalis, C. micacea, C. danica, 'C. anglica/hollandica.'*

The online version of this article includes the following source data for appendix 1—figure 6:

• **Appendix 1—figure 6—source data 1.** Best-scoring ML tree based on (nearly) complete plastid genomes and generated in RAxML with 1000 rapid bootstraps.

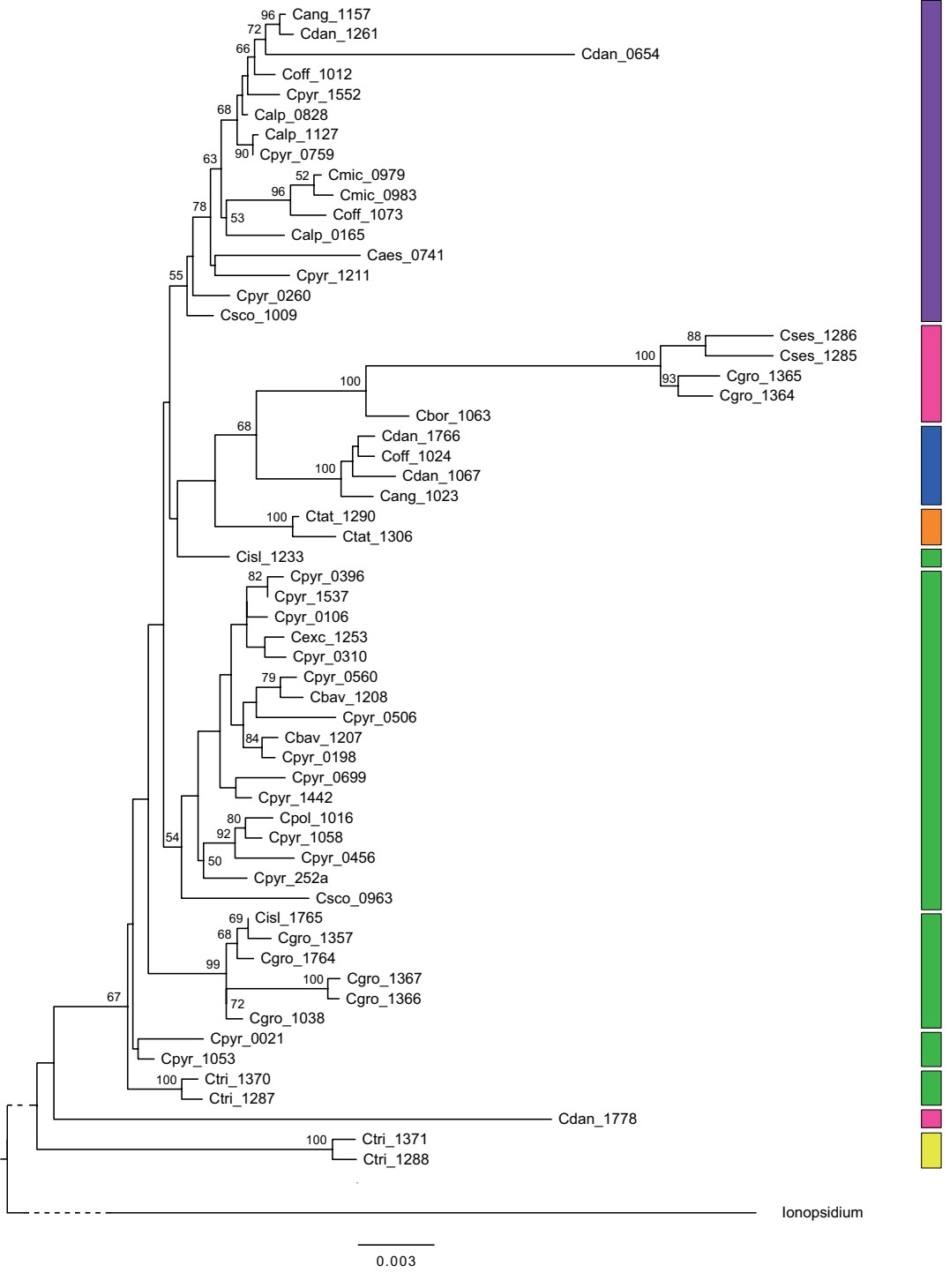

**Appendix 1—figure 7.** Mitochondrial maximum likelihood (ML) phylogeny. ML tree (*Appendix 1—figure 7—source data 1*) based on 58 partial mitochondrial genomes of genus *Cochlearia* and one outgroup sample of genus *Ionopsidium* (*Ionopsidium megalospermum*, sample-ID: Imega_1776). The latter is shown condensed for a better illustration of *Cochlearia* samples. Bootstrap support (1000 replicates) above 50% is shown near the respective nodes and colored blocks to the right correspond to respective chloroplast lineages as displayed in *Figure 2*.

The online version of this article includes the following source data for appendix 1—figure 7:

- **Appendix 1—figure 7—source data 1.** Best-scoring ML tree based on partial mitochondrial

genomes and generated in RAxML with 1000 rapid bootstraps.

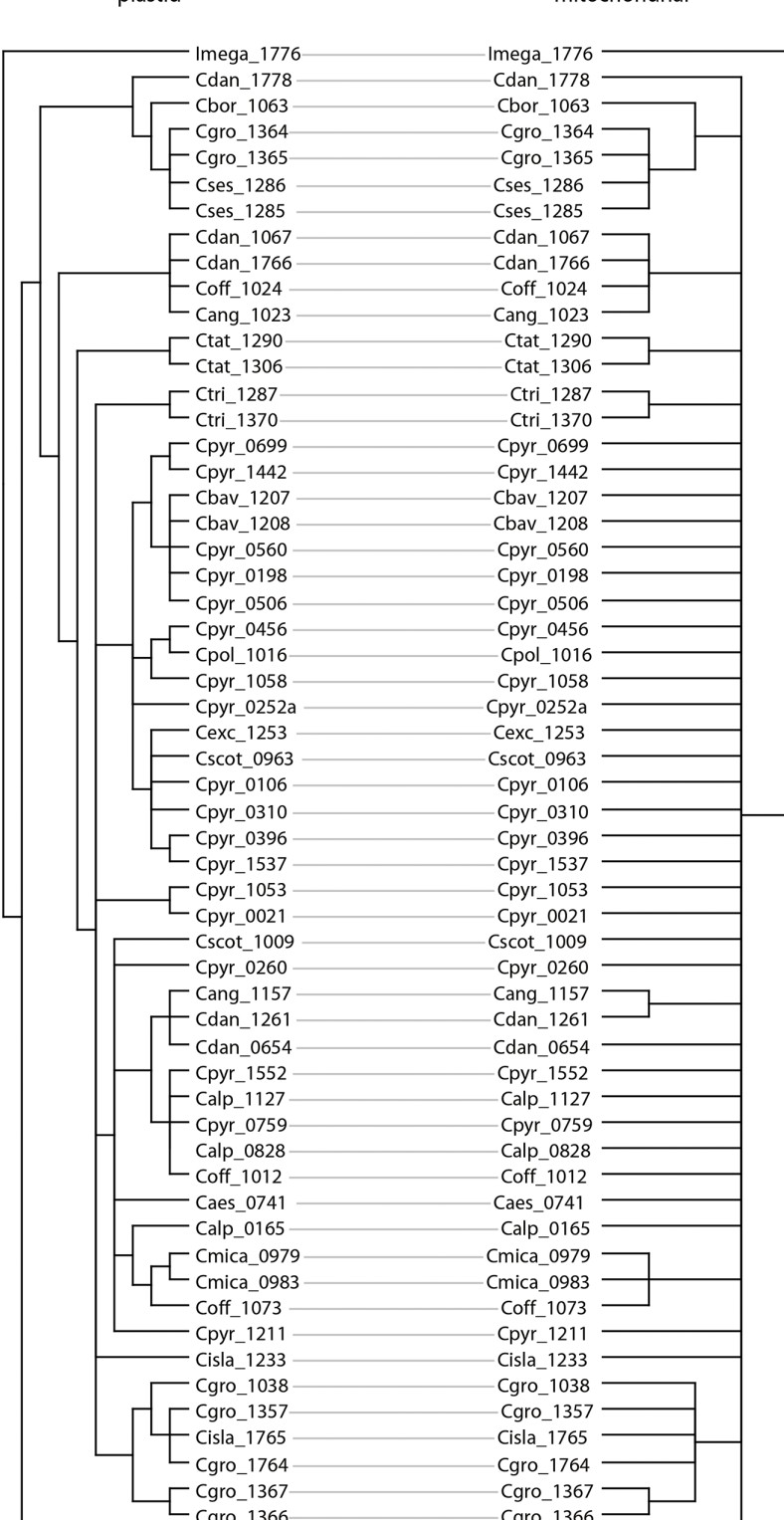

**Appendix 1—figure 8.** Tanglegram of plastid and mitochondrial phylogenies. The tanglegram was generated with dendroscope (*Huson and Scornavacca, 2012*) and branches with bootstrap support below 95% were collapsed prior to the analysis (*Appendix 1—figure 8—source data 1*).

The online version of this article includes the following source data for appendix 1—figure 8:

• **Appendix 1—figure 8—source data 1.** Tanglegram nexus-file (plastome versus mitochondrial sequence data).

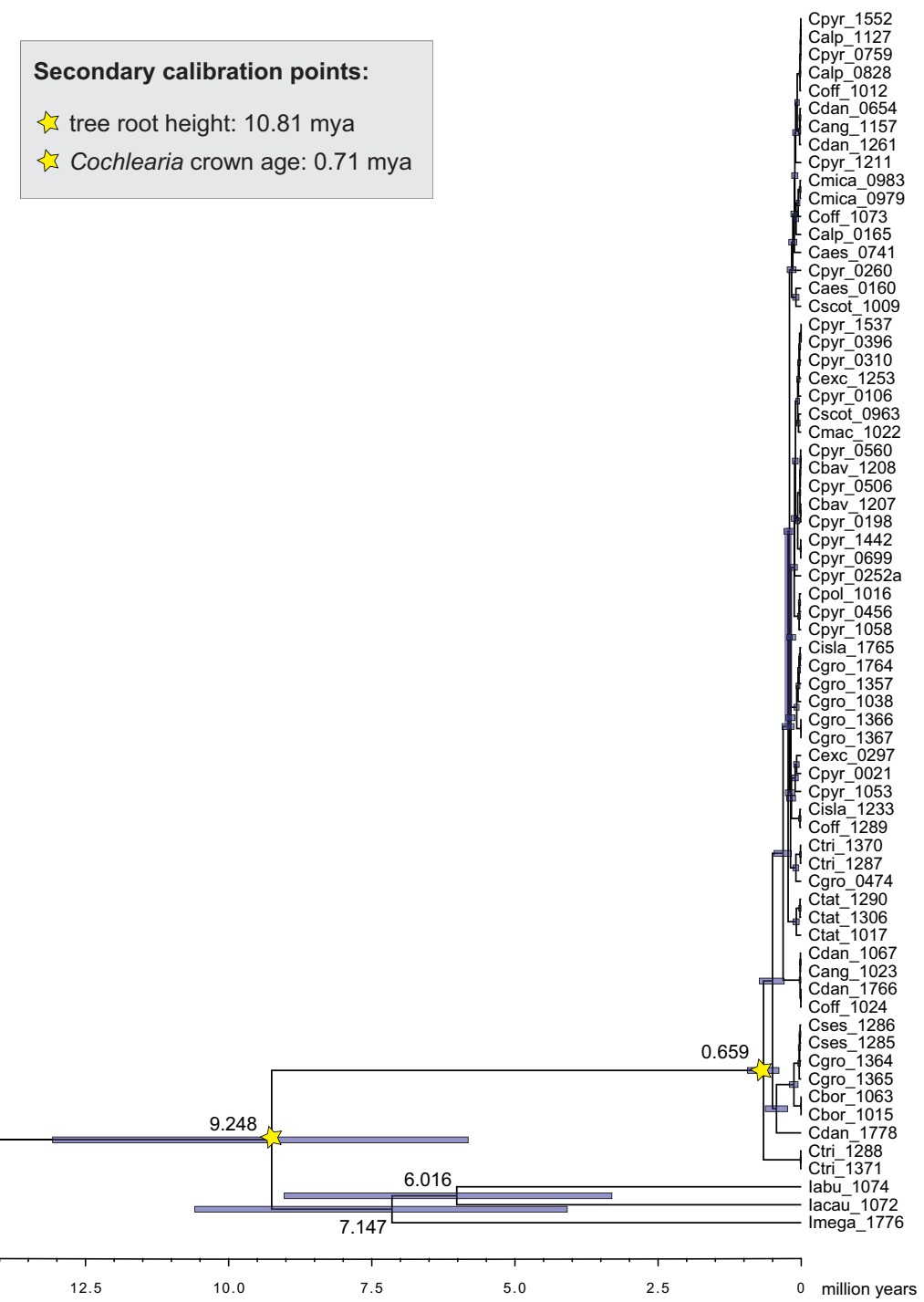

**Appendix 1—figure 9.** Cochlearieae maternal timeline. Cochlearieae BEAST chronogram based on whole chloroplast genome sequence data (*Figure 2—source data 1*). Secondary calibration points are indicated accordingly. Divergence times for the *Ionopsidium/Cochlearia* split, the genus *Ionopsidium,* and the *Cochlearia* crown age are indicated with 95% confidence intervals. Divergence time estimates within genus *Cochlearia* are given with *Appendix 1—figure 10.*

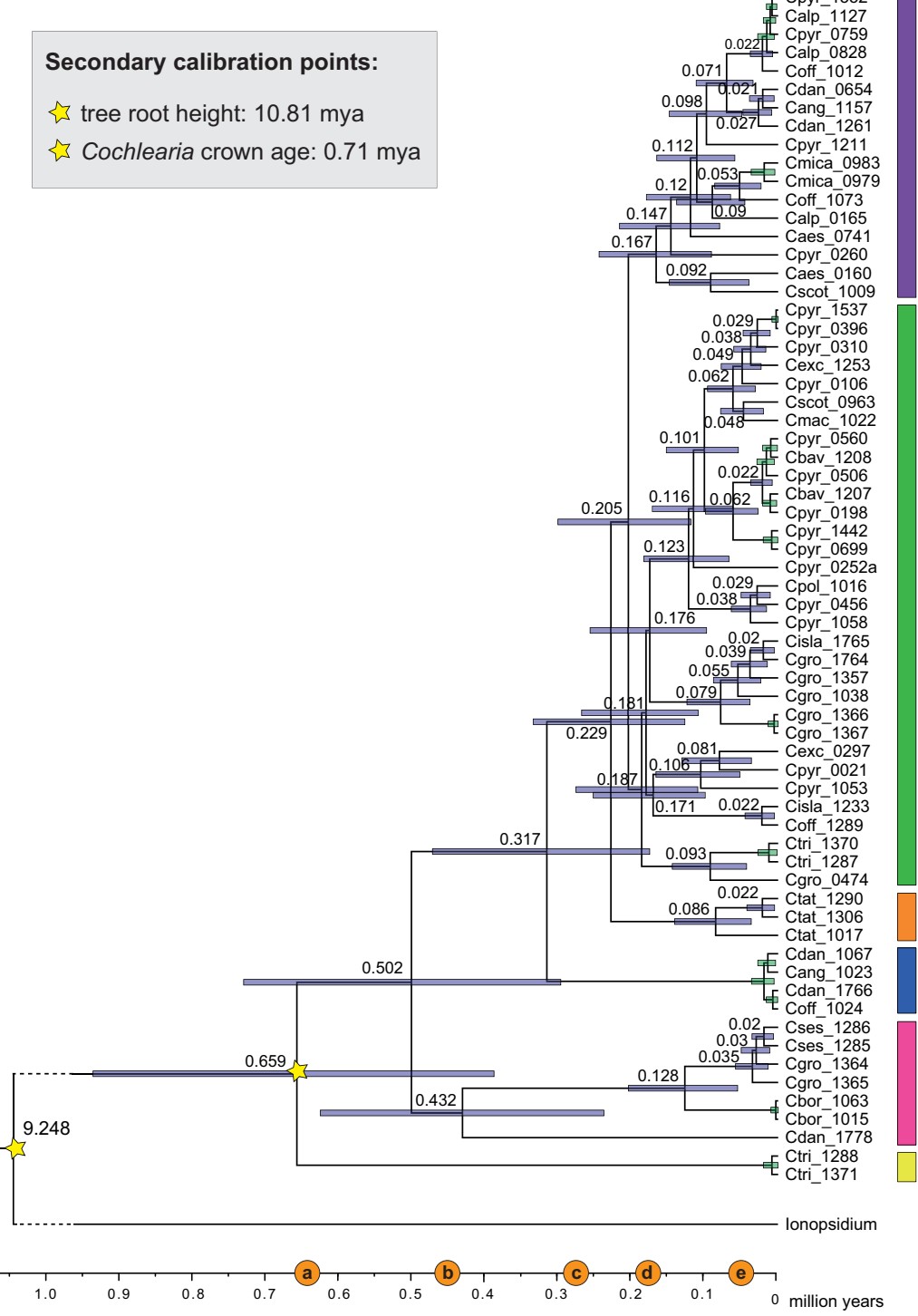

**Appendix 1—figure 10.** *Cochlearia* divergence time estimates. BEAST chronogram based on complete chloroplast genomes (*Figure 2—source data 1*). For illustration purpose, the *Ionopsidium* outgroup lineage is collapsed and condensed. The two secondary calibration points are indicated accordingly and divergence times older than 0.02 mya age are indicated with 95% confidence intervals. Green node bars indicate divergence times younger than 0.02 mya. Major evolutionary lineages within *Cochlearia* are displayed as colored bars (corresponding to *Figure 3*). Taxa included in evolutionary lineages: yellow lineage: *C. tridactylites*; pink lineage: *C. danica, C. borzaeana, C.*

*Appendix 1—figure 10 continued*
groenlandica, *C. sessilifolia*; blue lineage: *C. danica*, '*C. anglica/C. hollandica*,' *C. officinalis*; orange lineage: *C. tatrae*; green lineage: *C. officinalis, C. islandica, C. groenlandica, C. tridactylites, C. pyrenaica, C. excelsa, C. polonica, C. scotica, C. macrorrhiza, C. bavarica*; purple lineage: *C. scotica, C. aestuaria, C. pyrenaica, C. alpina, C. officinalis, C. micacea, C. danica,* '*C. anglica/hollandica*.' Letters displayed on the timeline indicate high glacial periods as follows: (a) 640 ky, end of Günz glacial; (b) 450 ky, begin of Mindel glacial; (c) 250–300 ky, Mindel-Riss inter-glacial; (d) 150–200 ky, Riss glacial; (e) 30–80 ky, Würm glacial.

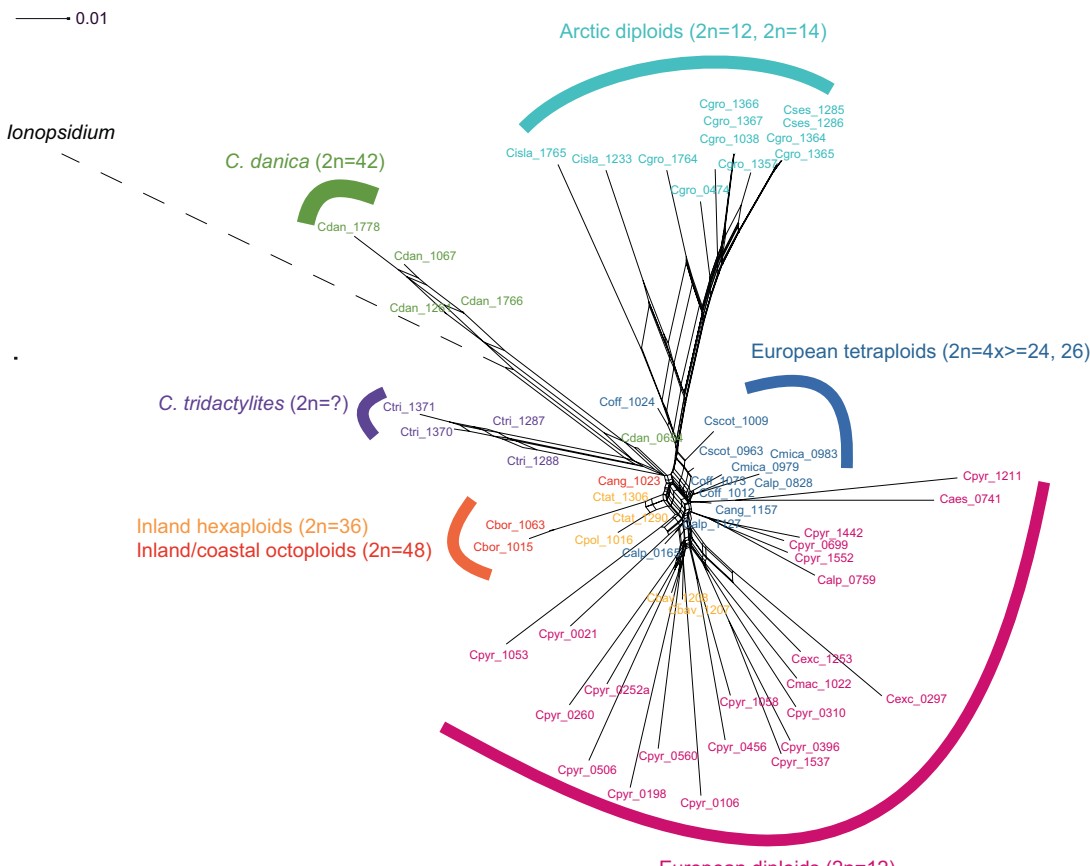

**Appendix 1—figure 11.** Detailed SplitsTree output. Result of a SplitsTree analysis of *Cochlearia* samples and *Ionopsidium* (outgroup) using the NeighborNet algorithm based on uncorrected p-distances (*Figure 3—source data 1*). For illustration purpose, the *Ionopsidium* branch is condensed.

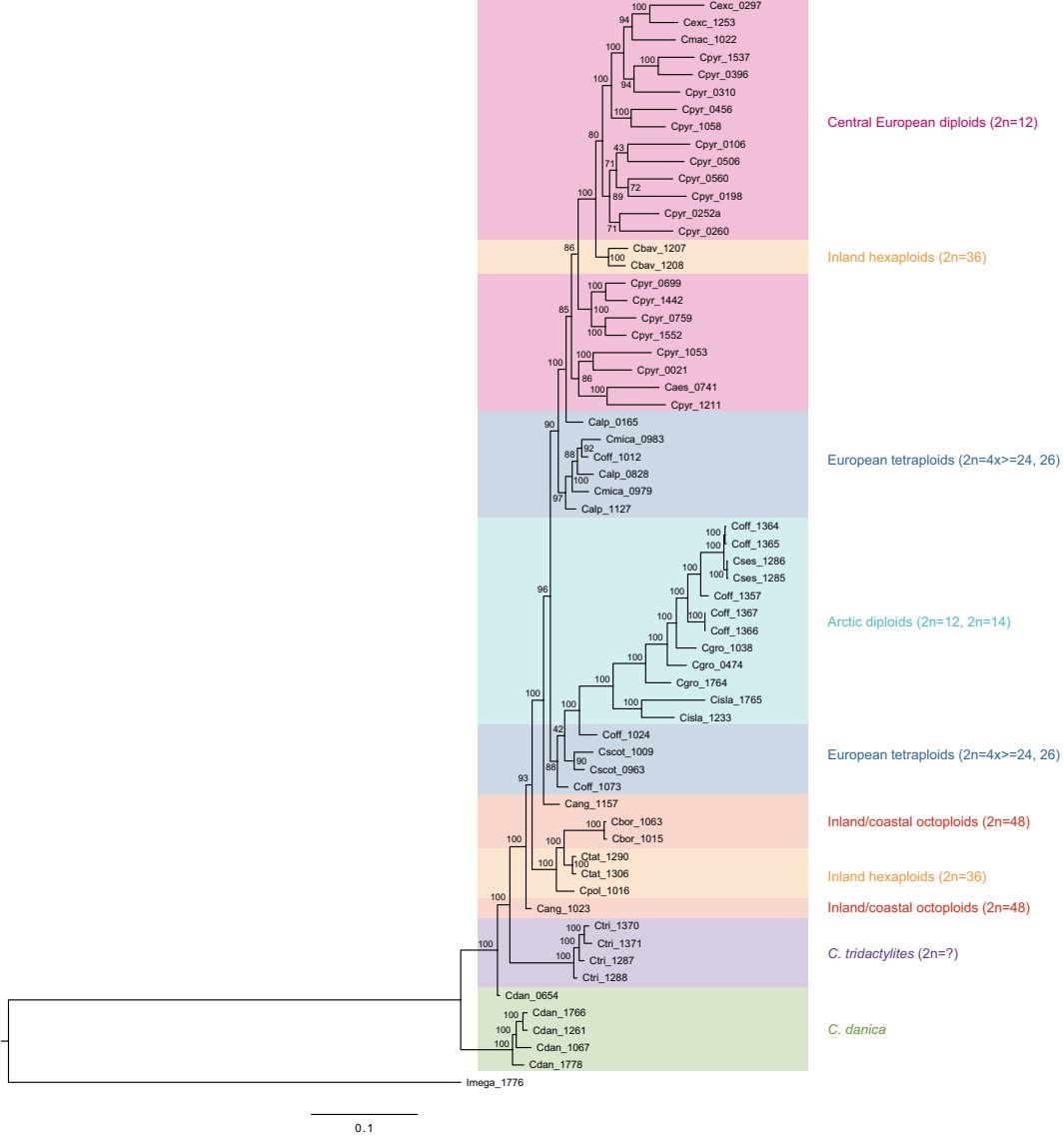

**Appendix 1—figure 12.** Maximum likelihood tree based on transcriptome-wide nuclear SNPs. The tree was generated with RAxML and the analysis (based on 298,978 variant sites) was performed with an ascertainment bias correction in order to account for sampling only variable sites (*Appendix 1—figure 12—source data 1*). Bootstrap support (1000 replicates) is shown near the respective nodes and coloring of sample groups corresponds to respective colors in the SplitsTree output as visualized in *Figure 3a*.

The online version of this article includes the following source data for appendix 1—figure 12:

• **Appendix 1—figure 12—source data 1.** Best-scoring ML tree based on transcriptome-wide nuclear SNPs (298,978 variant sites) and generated in RAxML with an ascertainment bias correction and 1000 rapid bootstraps.

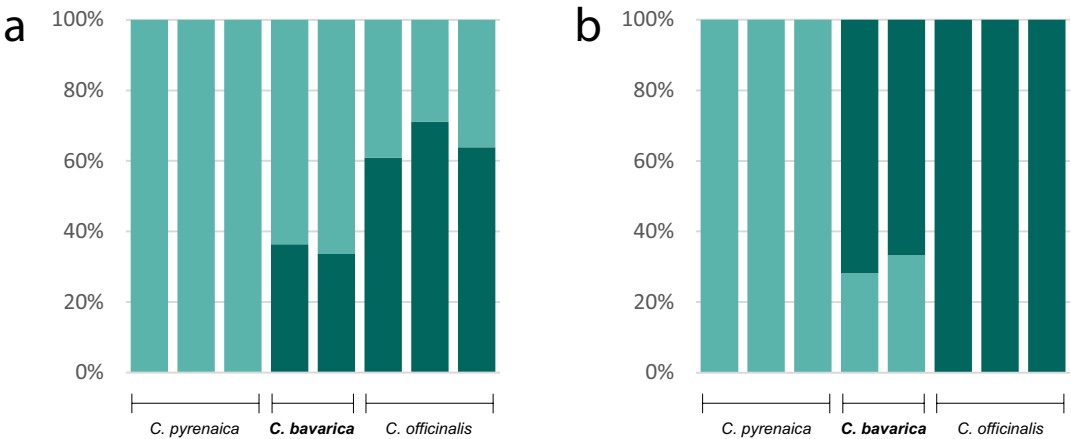

**Appendix 1—figure 13.** Genetic assignment analyses investigating the allopolyploid origin of the hexaploid species *Cochlearia bavarica* (2n=6x=36). Results of two STRUCTURE analyses based on nuclear transcriptome data (coding and non-coding; 103,874 variants) of two *C. bavarica* samples and samples from putative parental species *C. pyrenaica* (2n=2x=12) and *C. officinalis* (2n=4x=24). The analyses were performed under the *uncorrelated* (**a**; Appendix1-file13_SourceData1) and the *correlated* (**b**; Appendix1-file13_SourceData2) allele frequency models respectively, both assuming K=2.

The online version of this article includes the following source data for appendix 1—figure 13:

• **Appendix 1—figure 13—source data 1.** STRUCTURE result C. bavarica at K=2 (uncorrelated allele frequency model).

• **Appendix 1—figure 13—source data 2.** STRUCTURE result C. bavarica at K=2 (correlated allele frequency model).

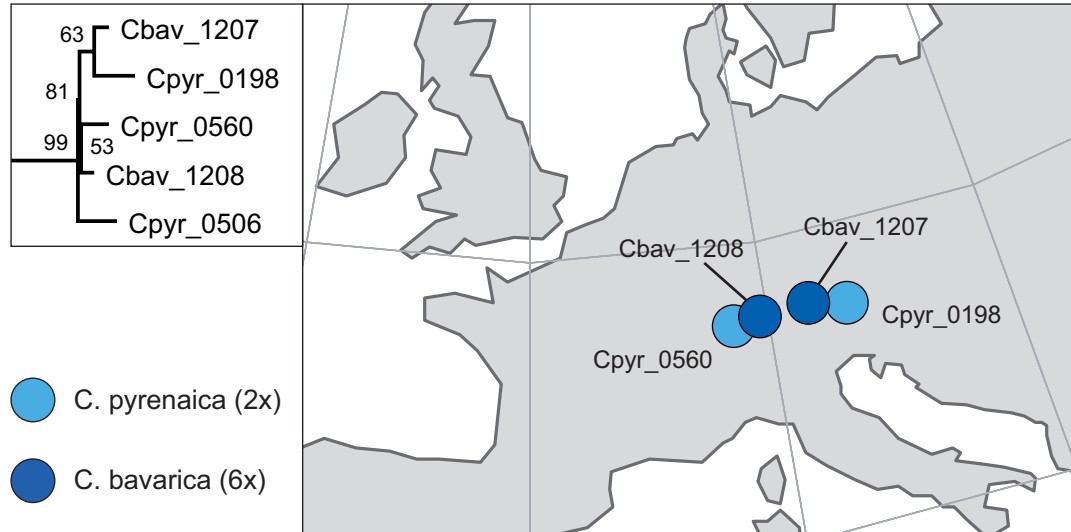

**Appendix 1—figure 14.** Phylogenetic placement and geographical occurrence of hexaploid *Cochlearia bavarica*. Geographically close populations of *C. pyrenaica* as putative maternal plants in past allopolyploidization processes of *C. bavarica* as revealed from the maximum likelihood plastid phylogeny (see *Appendix 1—figure 6*). Results are consistent with earlier population-based analyses using isozyme studies from multiple populations (*Koch, 2002*).

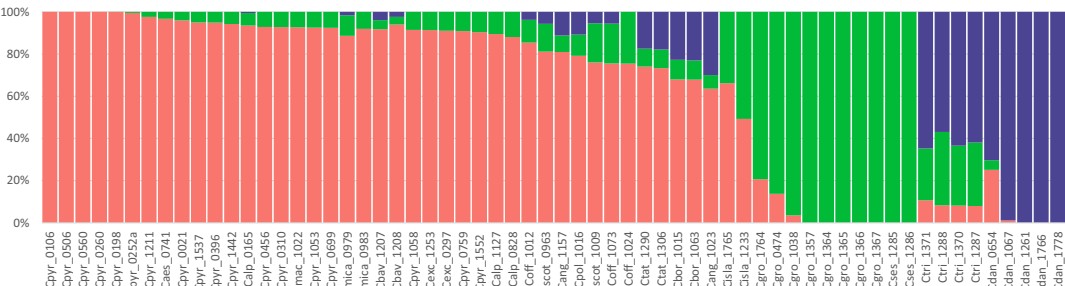

**Appendix 1—figure 15.** Genetic assignment analysis of the genus *Cochlearia* (all samples). Result of a STUCTURE analysis for all 62 *Cochlearia* samples assuming K=3 (*correlated allele frequencies*) based on 101,386 SNPs detected within 1,425,819 callable loci throughout the transcriptome (coding and non-coding). Colors correspond to the geographical representation as shown in *Figure 3*, *Figure 3—source data 2*. Every bar represents one individual and sample names are given below respective bars.

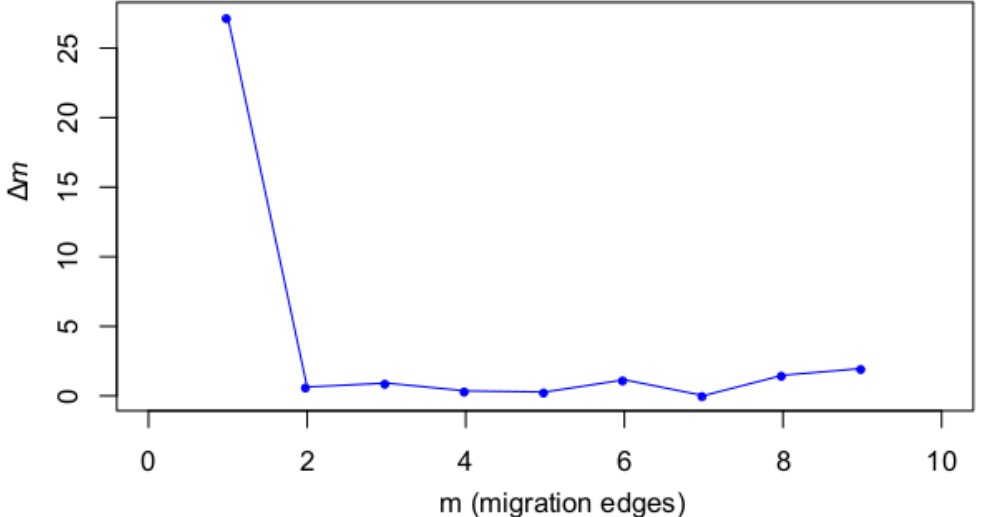

**Appendix 1—figure 16.** Optimal number of migration edges for TreeMix analysis. The optimal value of m (migration edges, x-axis) was assessed according to the Evanno method (deltaM, y-axis) across 10 replicates of each m from m=1 to m=10.

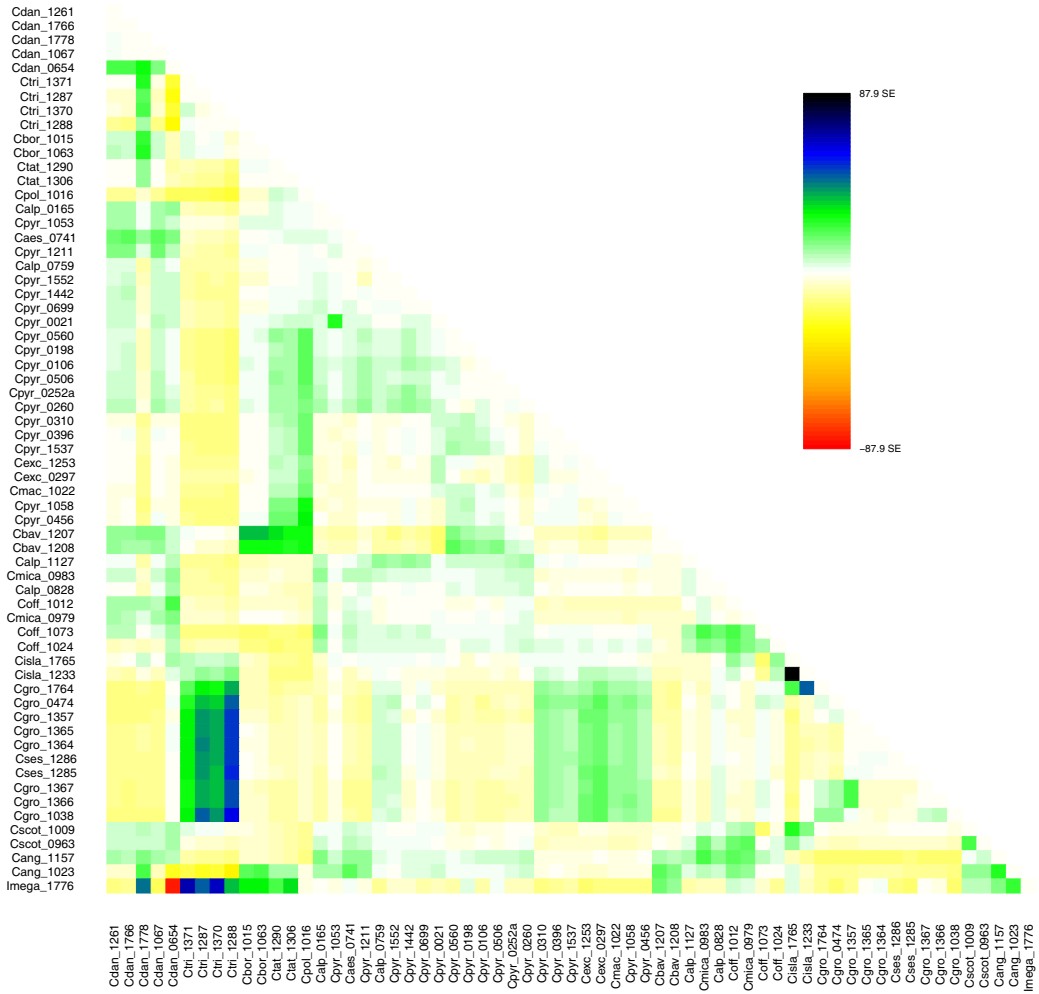

**Appendix 1—figure 17.** TreeMix residuals m=0. Residual plot of a TreeMix analysis with m = 0. Individuals (populations) that are not well-modeled can be identified by residuals deviating from zero with positive residuals indicating an underestimate of the observed covariance. Here, the fit of the model might be improved by additional migration edges.

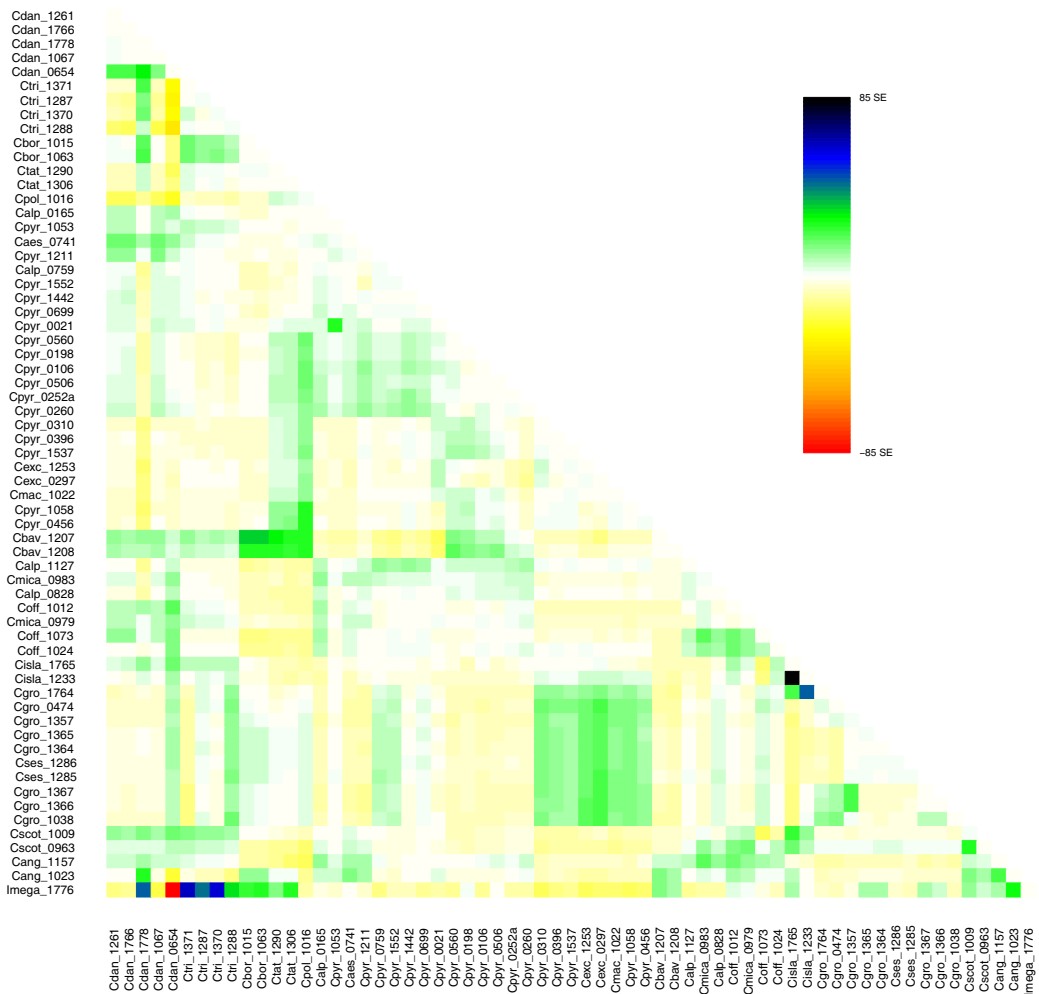

**Appendix 1—figure 18.** TreeMix residuals m=1. Residual plot of a TreeMix analysis with m=1 (best m, Evanno method; *Evanno et al., 2005*). Individuals (populations) that are not well-modeled can be identified by residuals deviating from zero with positive residuals indicating an underestimate of the observed covariance. Here, the fit of the model might be improved by additional migration edges.

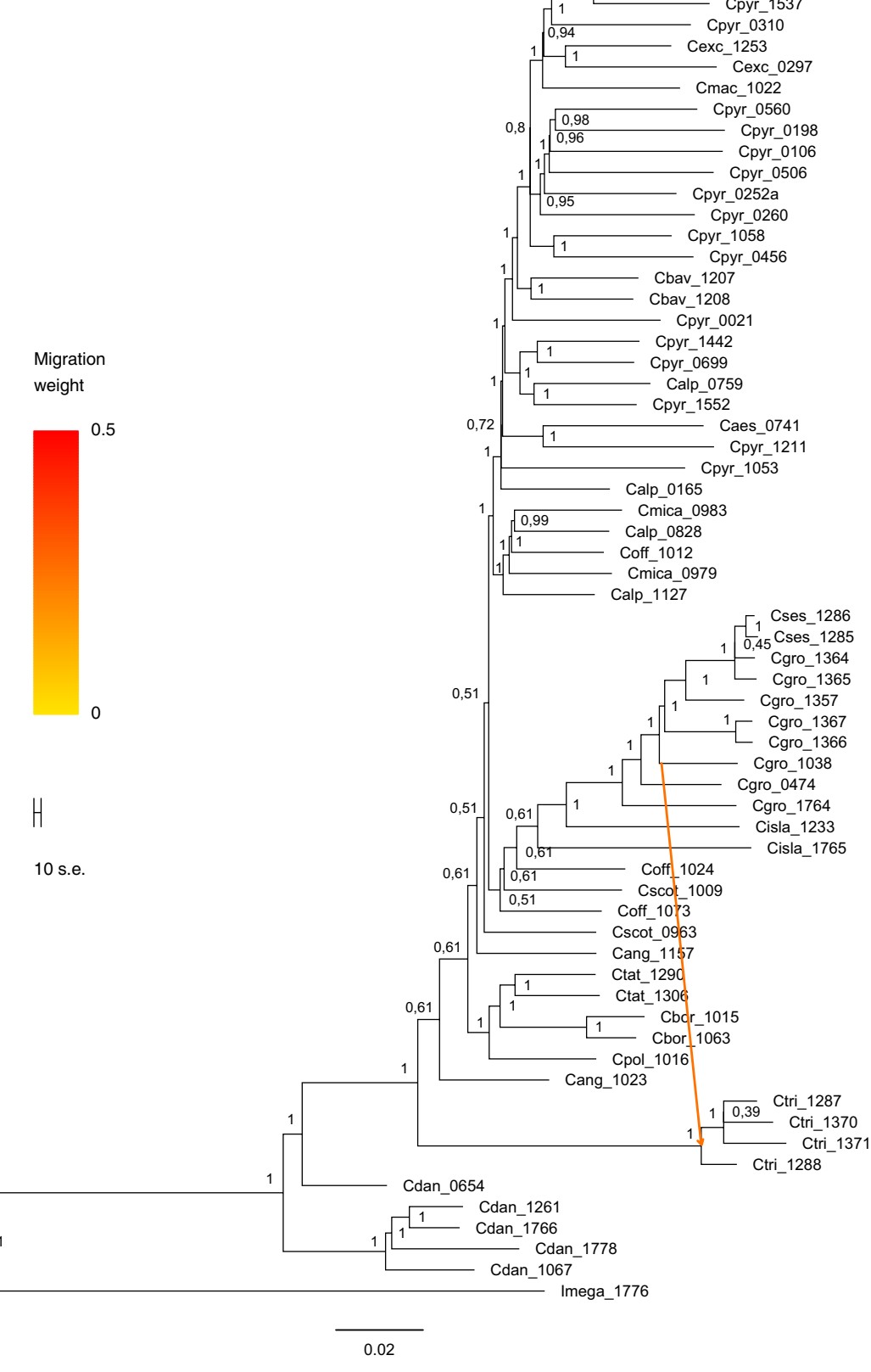

**Appendix 1—figure 19.** TreeMix bootstrapped tree with migration edge (m=1). The maximum likelihood consensus tree (_SourceData1)*Appendix 1—figure 19—source data 1* was generated with TreeMix (*Pickrell and Pritchard, 2012*) with one migration/admixture event (best m, Evanno

*Appendix 1—figure 19 continued*

method; ***Appendix 1—figure 16***). TreeMix analysis indicates that migration occurred from Arctic *C. groenlandica* (2n=14) to Eastern Canadian *C. tridactylites* (2n=?). Migration arrow is colored by migration weight and indicates the direction of gene flow. The consensus tree was inferred from 100 bootstrap replicates using SumTrees (***Sukumaran and Holder, 2010***).

The online version of this article includes the following source data for appendix 1—figure 19:

• **Appendix 1—figure 19—source data 1.** Bootstrapped (100 bootstrap replicates) maximum likelihood consensus tree generated with TreeMix with one migration/admixture event.

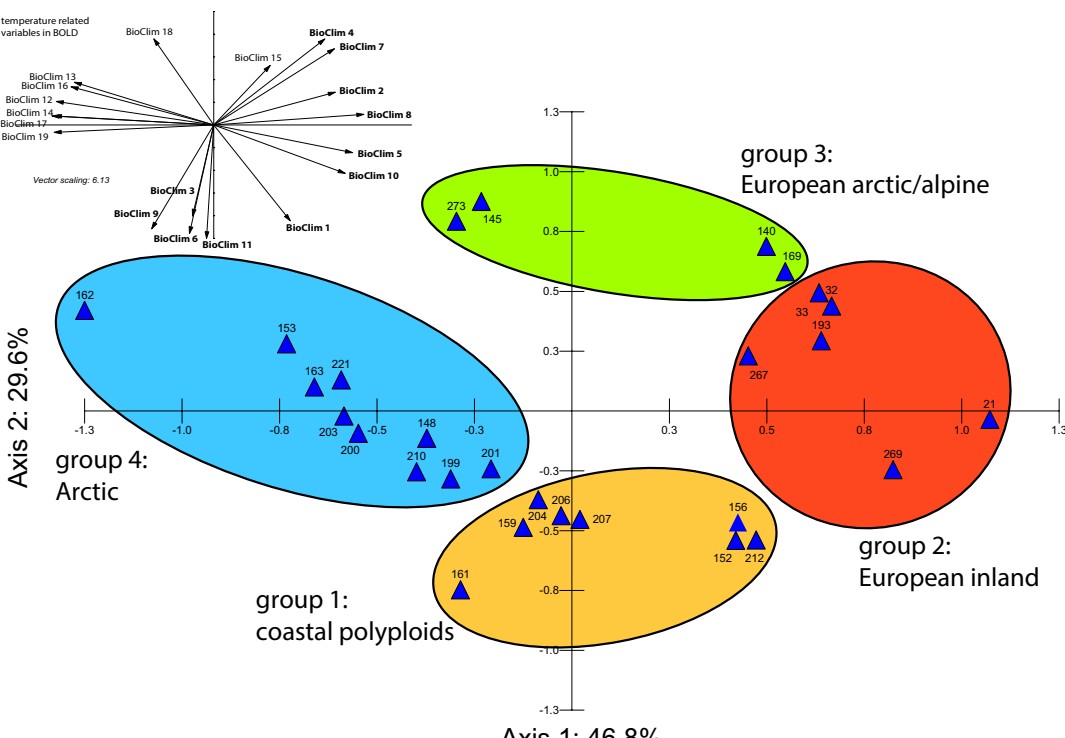

**Appendix 1—figure 20.** Principal component analysis (PCA) based on all 19 bioclimatic variables. PCA result based on all 19 WorldClim bioclimatic variables for 28 *Cochlearia* accessions (***Appendix 1—figure 20—source data 1***); biplot of the first two axes explaining 76.4% of the total variance. Group colors correspond to bioclimatic clusters (***Figure 4***) as also defined by hierarchical cluster analysis.

The online version of this article includes the following source data for appendix 1—figure 20:

• **Appendix 1—figure 20—source data 1.** 19 WorldClim bioclimatic variables for 28 Cochlearia accessions.

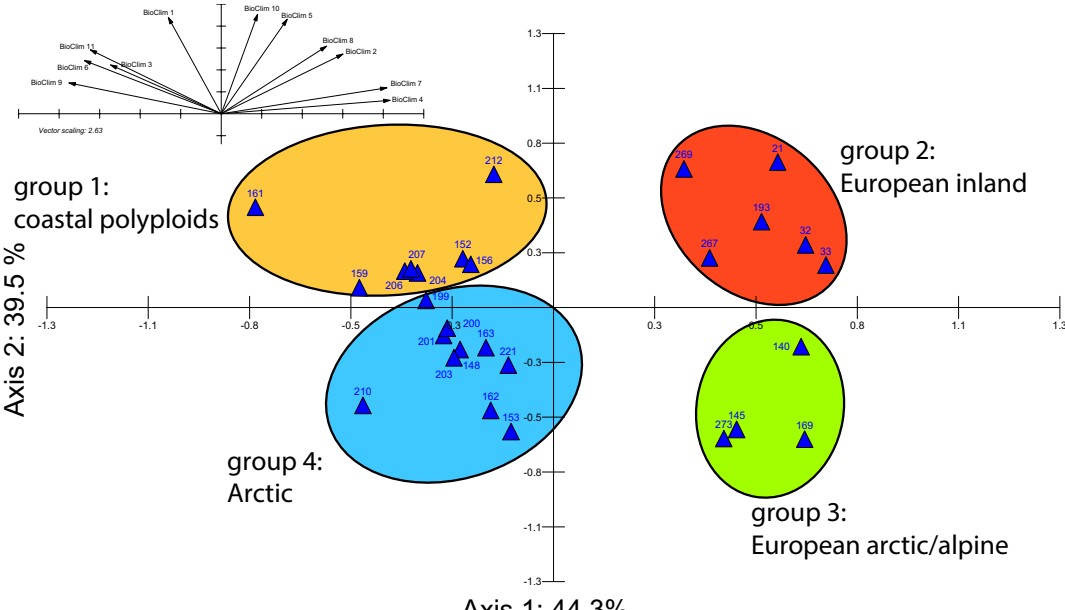

**Appendix 1—figure 21.** Principal component analysis (PCA) based on bioclimatic variables. PCA result based on WorldClim bioclimatic variables 1–11 (temperature-related) for 28 *Cochlearia* accessions (*Appendix 1—figure 21—source data 1*); biplot of the first two axes explaining 83.8% of the total variance. Group colors correspond to bioclimatic clusters (*Figure 4*) as also defined by hierarchical cluster analysis.

The online version of this article includes the following source data for appendix 1—figure 21:

• **Appendix 1—figure 21—source data 1.** 11 WorldClim bioclimatic variables for 28 Cochlearia accessions.

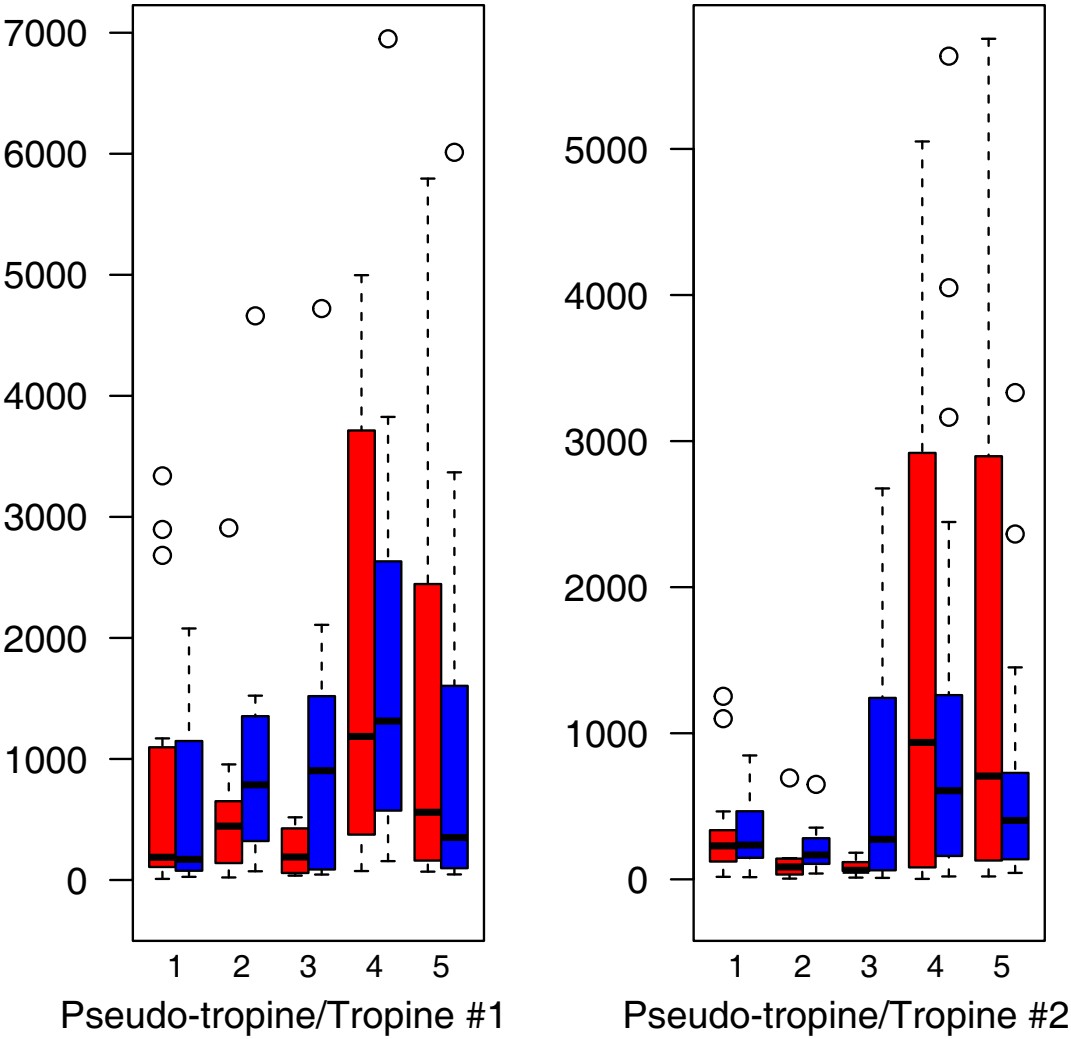

**Appendix 1—figure 22.** Alkaloid relative levels (normalized to the ribitol internal standard) under control/cold conditions. Boxplots showing means of normalized levels to the ribitol internal standard (y-axis) of analyzed alkaloids in plants from four bioclimatic *Cochlearia* clusters (x-axis: 1–4; see *Figure 4a*) and (b) and *Ionopsidium* (x-axis: 5) as measured by gas chromatography-mass spectrometry after growing under control (18°C/20 °C, red) or cold (5°C, blue) conditions for 20 days. No significant differences between the two treatments within each cluster were revealed via one-way ANOVA (*Appendix 1—figure 22—source data 1*).

The online version of this article includes the following source data for appendix 1—figure 22:

• **Appendix 1—figure 22—source data 1.** Detailed summary output of one-way ANOVAs for all metabolites for bioclimatic clusters (1–4) and Ionopsidium (5).

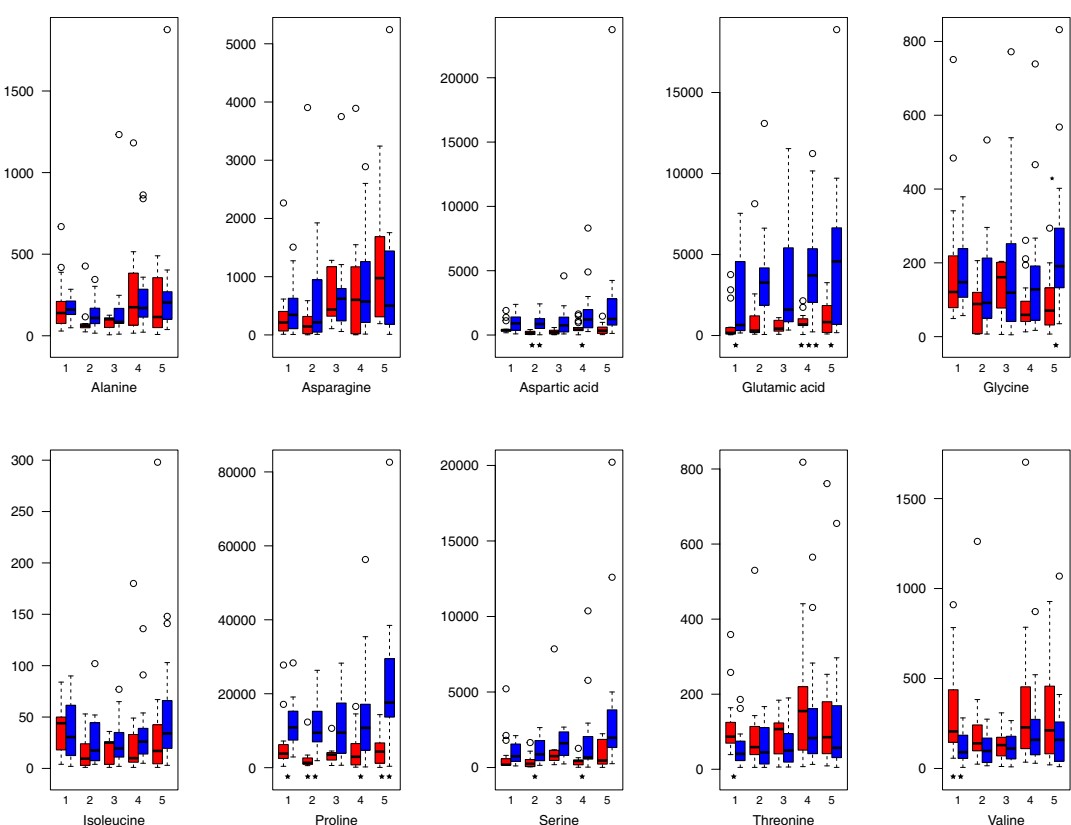

**Appendix 1—figure 23.** Amino acid relative levels (normalized to the ribitol internal standard) under control/cold conditions. Boxplots showing means of normalized levels to the ribitol internal standard (y-axis) of analyzed amino acids in plants from four bioclimatic *Cochlearia* clusters (x-axis: 1–4; see *Figure 4a* and b) and *Ionopsidium* (x-axis: 5) as measured by gas chromatography-mass spectrometry after growing under control (18°C/20°C, red) or cold (5°C, blue) conditions for 20 days. Significant differences between the two treatments within each cluster as revealed via one-way ANOVA are indicated by asterisks (*p≤0.05; **p≤0.02; ***p≤0.001; *Appendix 1—figure 22—source data 1*).

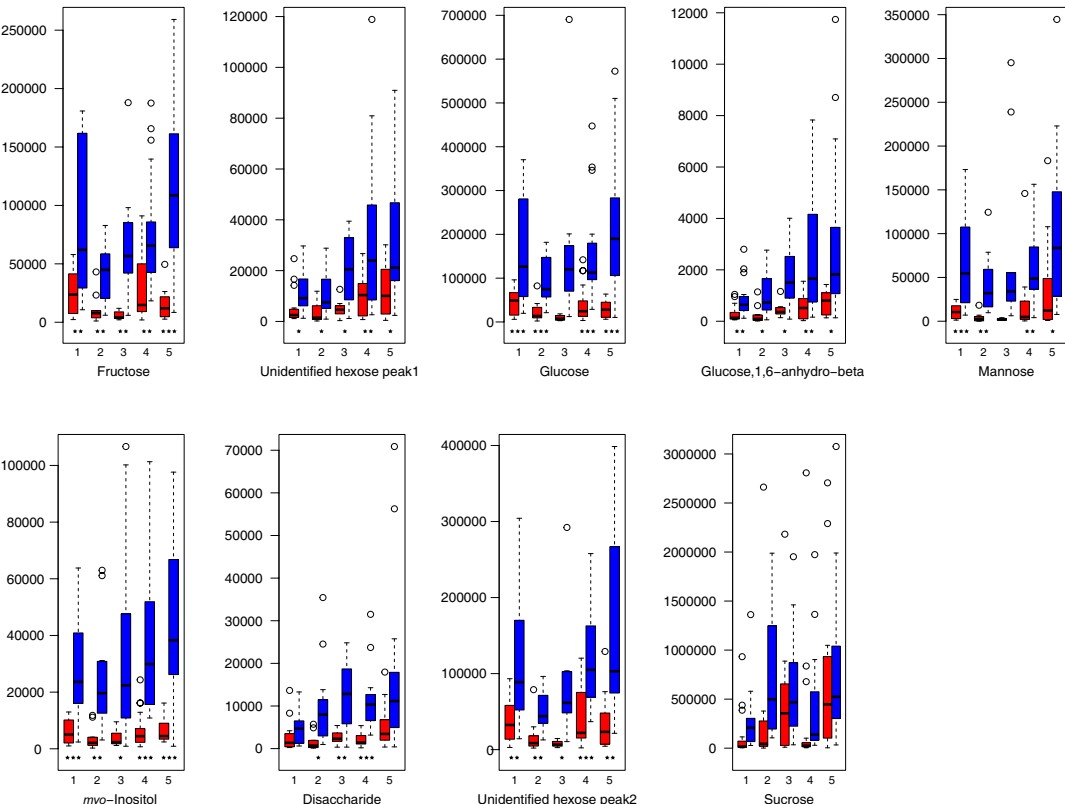

**Appendix 1—figure 24.** Carbohydrate relative levels (normalized to the ribitol internal standard) under control/cold conditions. Boxplots showing means of normalized levels to the ribitol internal standard (y-axis) of analyzed carbohydrates in plants from four bioclimatic *Cochlearia* clusters (x-axis: 1–4; see *Figure 4a* and b) and *Ionopsidium* (x-axis: 5) as measured by gas chromatography-mass spectrometry after growing under control (18°C/20°C, red) or cold (5°C, blue) conditions for 20 days. Significant differences between the two treatments within each cluster as revealed via one-way ANOVA are indicated by asterisks (*p≤0.05; **p≤0.02; ***p≤0.001; *Appendix 1—figure 22—source data 1*).

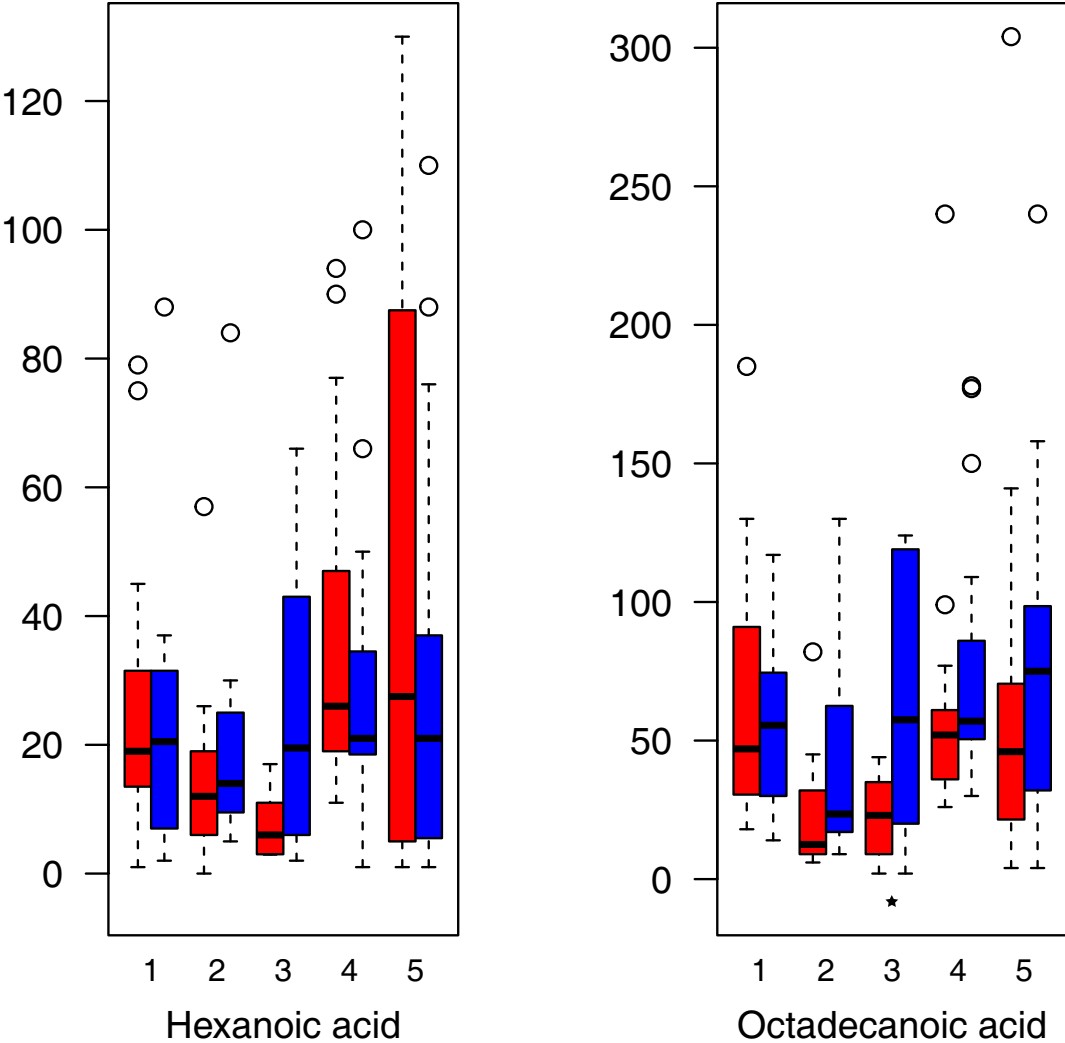

**Appendix 1—figure 25.** Fatty acid relative levels (normalized to the ribitol internal standard) under control/cold conditions. Boxplots showing means of normalized levels to the ribitol internal standard (y-axis) of analyzed fatty acids in plants from four bioclimatic *Cochlearia* clusters (x-axis: 1–4; see *Figure 4a* and b) and *Ionopsidium* (x-axis: 5) as measured by gas chromatography-mass spectrometry after growing under control (18°C/20°C, red) or cold (5°C, blue) conditions for 20 days. Significant differences between the two treatments within each cluster as revealed via one-way ANOVA are indicated by asterisks (*p≤0.05; **p≤0.02; ***p≤0.001; *Appendix 1—figure 22—source data 1*).

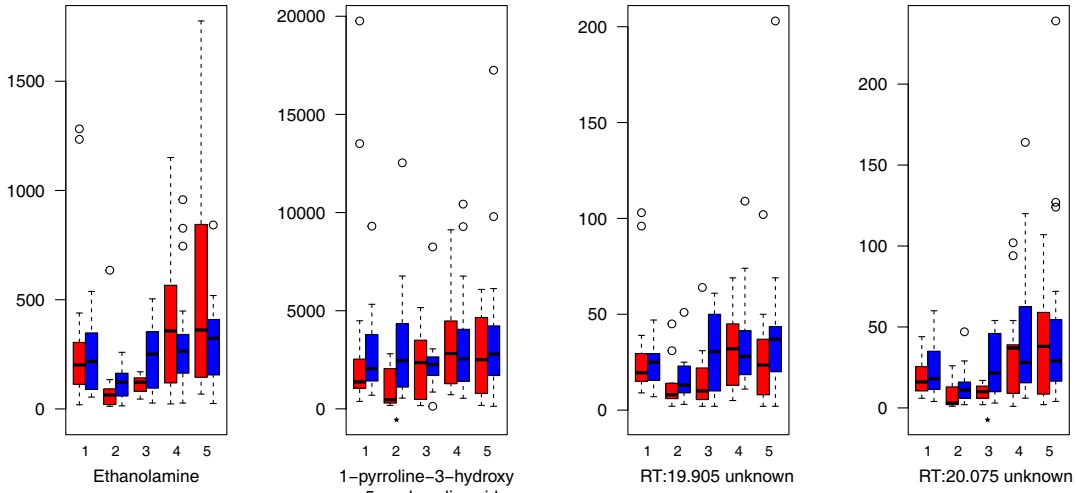

**Appendix 1—figure 26.** Miscellaneous compounds relative levels (normalized to the ribitol internal standard) under control/cold conditions. Boxplots showing means of normalized levels to the ribitol internal standard (y-axis) of analyzed miscellaneous compounds in plants from four bioclimatic *Cochlearia* clusters (x-axis: 1–4; see *Figure 4a* and b) and *Ionopsidium* (x-axis: 5) as measured by gas chromatography-mass spectrometry after growing under control (18°C/20°C, red) or cold (5°C, blue) conditions for 20 days. Significant differences between the two treatments within each cluster as revealed via one-way ANOVA are indicated by asterisks (*p≤0.05; **p≤0.02; ***p≤0.001; *Appendix 1—figure 22—source data 1*). RT, Retention Time in minutes.

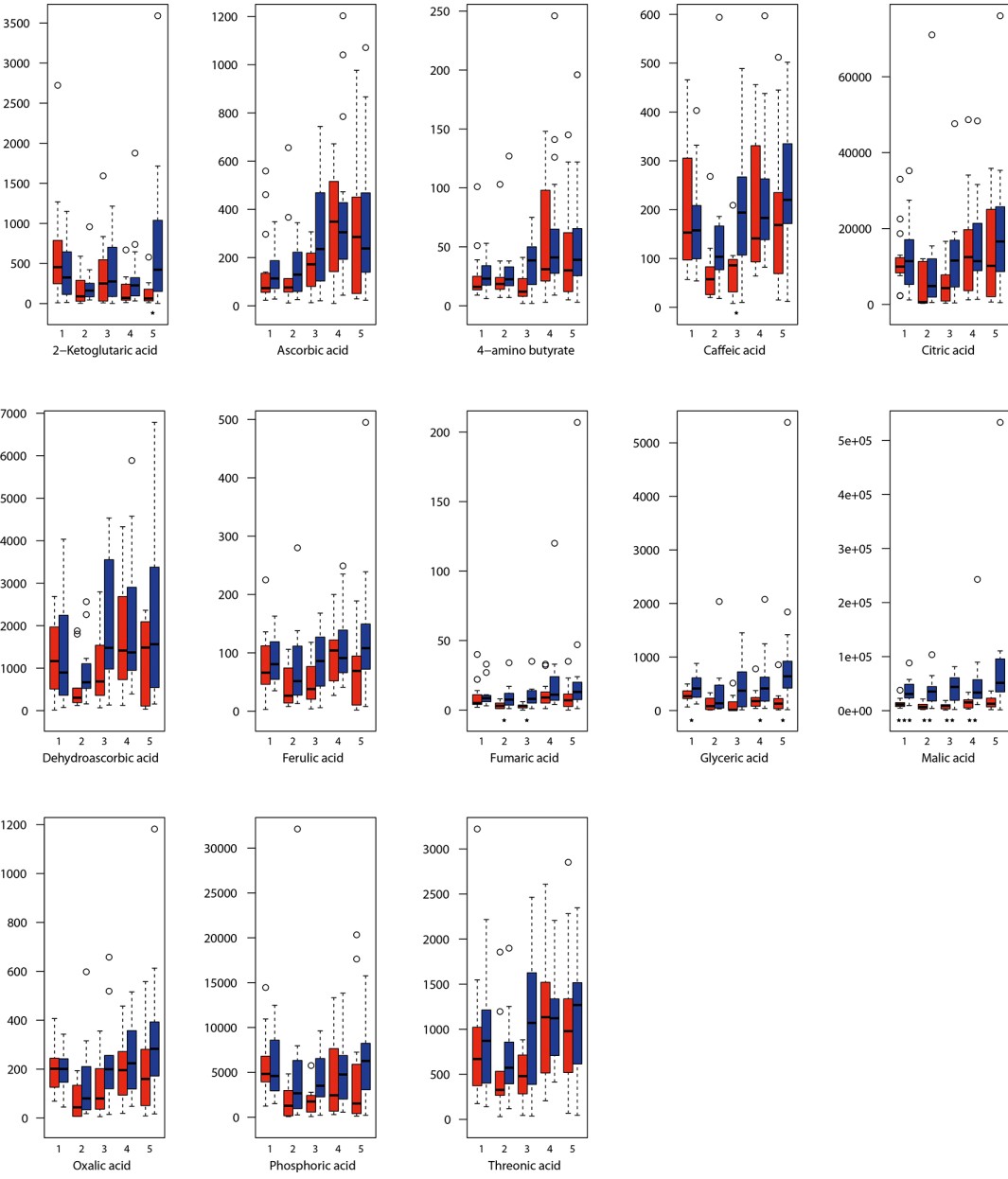

**Appendix 1—figure 27.** Organic acid relative levels (normalized to the ribitol internal standard) under control/cold conditions. Boxplots showing means of normalized levels to the ribitol internal standard (y-axis) of analyzed organic acids in plants from four bioclimatic *Cochlearia* clusters (x-axis: 1–4; see *Figure 4a* and b) and *Ionopsidium* (x-axis: 5) as measured by gas chromatography-mass spectrometry after growing under control (18°C/20°C, red) or cold (5°C, blue) conditions for 20 days. Significant differences between the two treatments within each cluster as revealed via one-way ANOVA are indicated by asterisks (*p≤0.05; **p≤0.02; ***p≤0.001; *Appendix 1—figure 22—source data 1*)

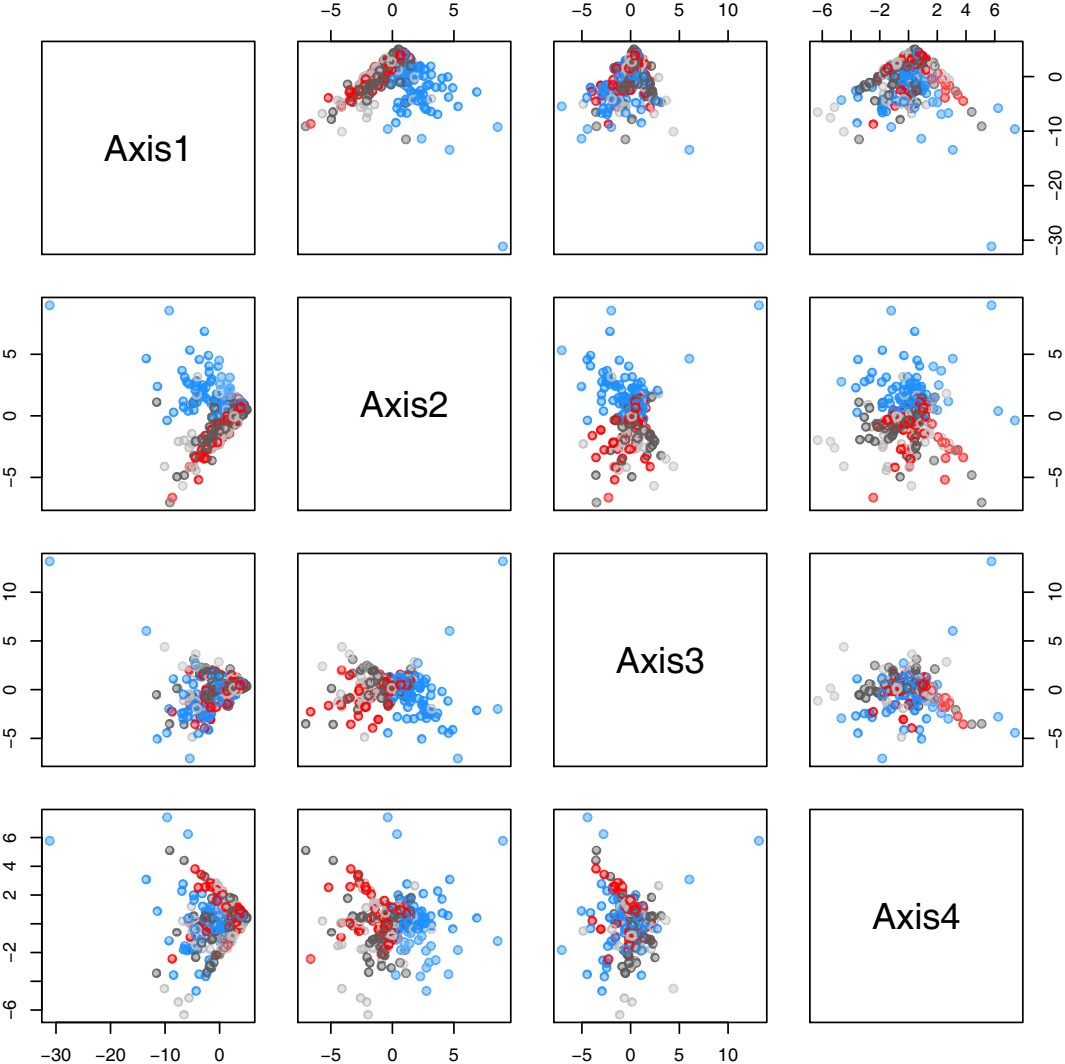

**Appendix 1—figure 28.** PCA plot of metabolite measurements grouped by treatment. The first four principal components (Axis1–4) of the normalized metabolite data set grouped by treatment plotted against each other. Colors are representing metabolite extractions as illustrated in *Figure 4c*. Light/dark grey: plants after the first round of metabolite extractions under control conditions (18°C/20°C) prior to temperature treatment; red: control plants after another 20 days under control conditions; blue: plants after 20 days of cold treatment (5°C).

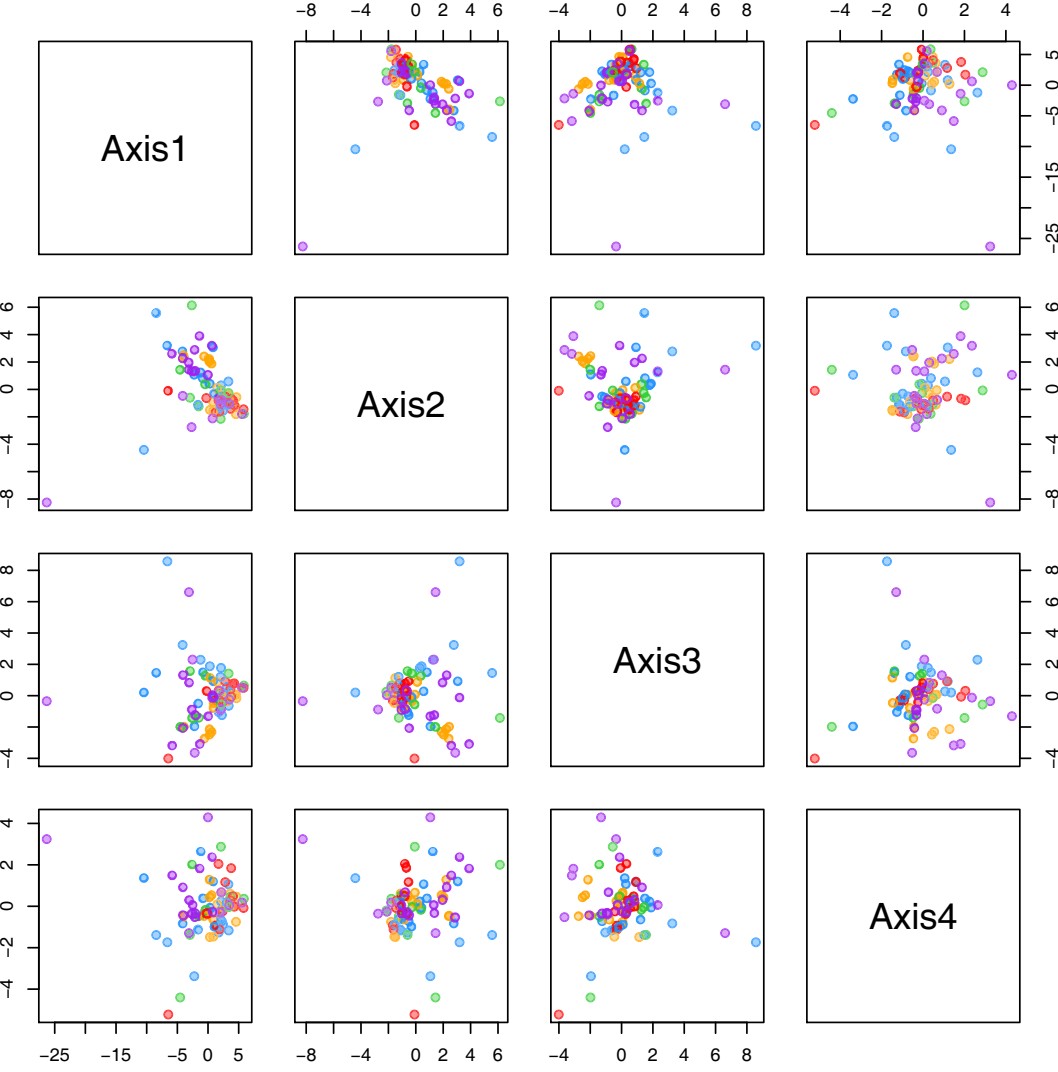

**Appendix 1—figure 29.** PCA plot of metabolite measurements after cold treatment grouped by bioclimatic clusters. The first four principal components (Axis1–4) of the normalized metabolite data set grouped by bioclimatic *Cochlearia* clusters plotted against each other. Colors are representing the four bioclimatic clusters as illustrated in *Figure 4a* and *Ionopsidium* (purple).

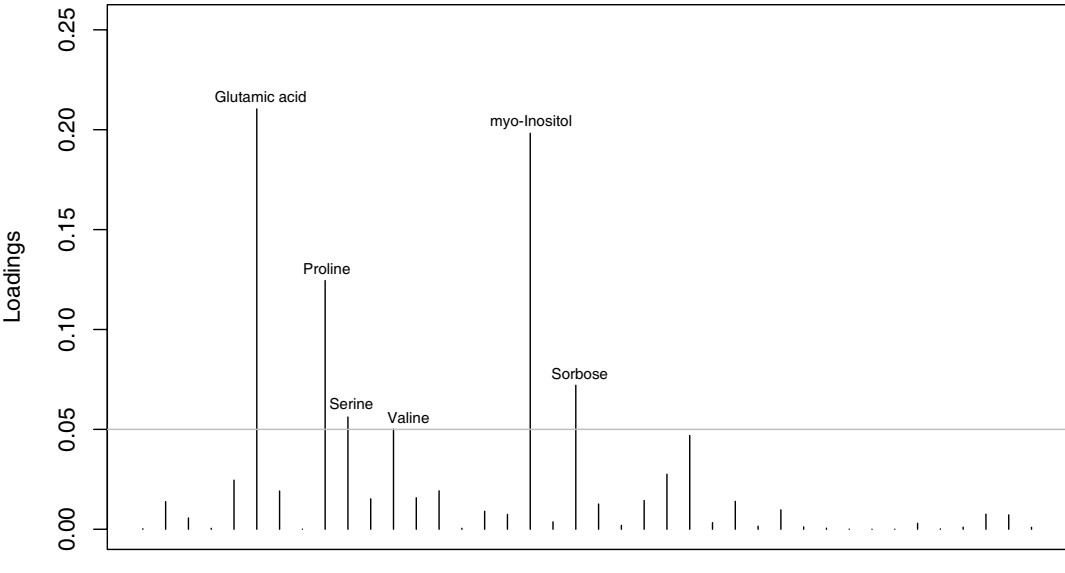

**Appendix 1—figure 30.** Metabolites with highest discriminating power in DAPC grouped by treatment. Loading plot of metabolite contributions to discriminant function number 1 (separating control and cold conditions) in DAPC analysis as illustrated in *Figure 4d* (based on all metabolomic measurements and grouped by treatment). The grey line indicates a threshold of 0.05 and metabolites above this threshold are labeled accordingly. Detailed metabolite contributions given with *Appendix 1—figure 30—source data 1*.

The online version of this article includes the following source data for appendix 1—figure 30:

• **Appendix 1—figure 30—source data 1.** Metabolite contributions to discriminant function number 1 (separating control and cold conditions) in DAPC analysis as illustrated in *Figure 4d* (based on all metabolomic measurements and grouped by treatment) sorted from highest to lowest discriminating power.

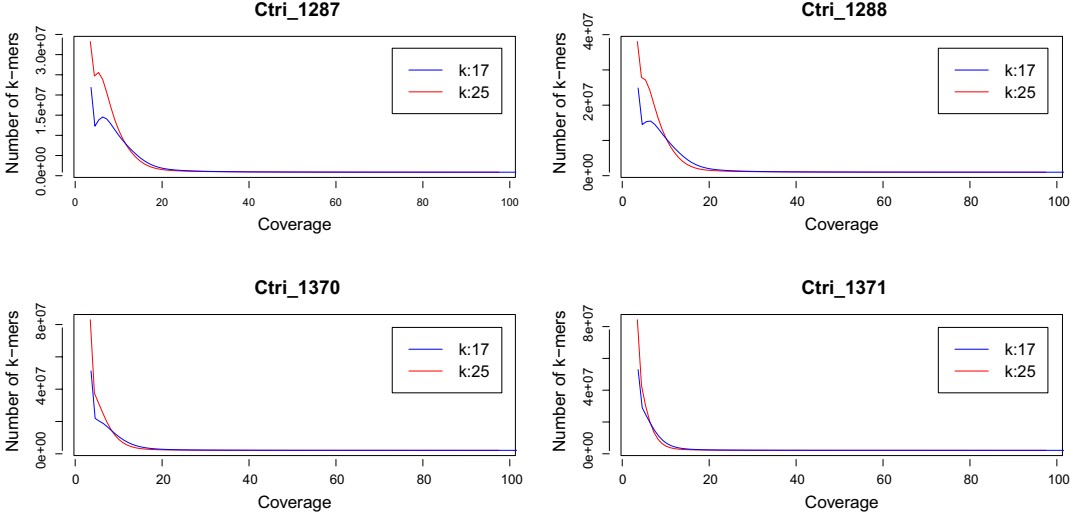

**Appendix 1—figure 31.** Examples on gating and peak analysis of flow cytometry measurements. Gating (**a**) and peak analysis (**b**) were performed via the Partec FloMax software. Gating was applied as a simple polygon-based gating, defined in the fluorescence channel 1 (FL1) versus side scatter
*Appendix 1—figure 31 continued on next page*

*Appendix 1—figure 31 continued*

(SSC) dot plot, in order to remove the generally high amount of cellular debris. Samples were stained with propidium iodide (PI) as indicated on the x-axis label. Additional information on corresponding statistics is given with Supplementary Data Set 4.

## Appendix 2

### Ploidy estimation for *C. tridactylites*

Without any published chromosome counts or genome size measurements nor fresh leaf material for flow cytometry analyses accessible, we performed a ploidy estimation for *C. tridactylites* based on *k*-mer analyses of quality trimmed sequencing reads. Yet, given the shallow sequencing depth, the curve of the generated *k*-mer histograms did not show a normal distribution for any of the sequenced *C. tridactylites* samples. Only two samples exhibited observable peaks but the curve shapes did not allow for accurate genome size estimates (*Appendix 2—figure 1*). As a consequence of these uncertainties and given the fact that results of the nuclear genome data analyses indicated that *C. tridactylites* could likewise be a polyploid of hybrid origin, we finally excluded the species from the set of known diploids and respective analyses of diploid samples.

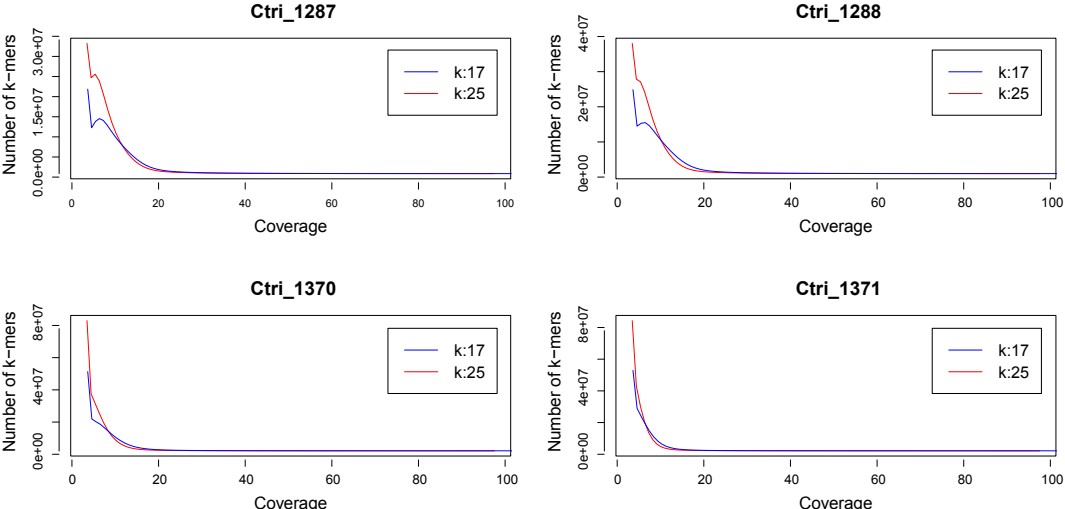

**Appendix 2—figure 1.** *k*-mer analysis of sequenced *Cochlearia tridactylites* samples. Distribution of *k*-mers 17 and 25 estimated using Jellyfish (*Marçais and Kingsford, 2011*).

## Appendix 3

### Further explanations on ABC model choice

Lack of power for the distinction of gene flow is unlikely to be a specific problem of the *Cochlearia* case, or a methodological shortcoming, but may instead reflect a fundamental property of populations with low divergence levels. It has been shown that genetic data from low-divergence populations may fit equally well to a model without any gene flow and recent population split time, and an alternative model with strong gene flow and ancient split time (*Hey et al., 2015*), which can lead to very high false-positive rates for the inference of gene flow in low divergence data (*Cruickshank and Hahn, 2014*). Although this problem was so far addressed mainly in the context of a particular inference method, the IMa2 program (*Hey, 2010*), it is likely to be relevant in general, and it may affect traditional rejection-ABC model choice in particular. This method approximates the relative posterior probability of a model and priors by the frequency of simulated data sets that are similar to the observed data. However, low genetic divergence will necessarily be obtained more frequently under models with gene flow than under models without gene flow, all other priors of the models being equal. In other words, low divergence is a-priori more probable with gene flow than without it, and thus rejection-ABC for low divergence data sets can provide little insight beyond this intuitive expectation. In contrast to rejection-ABC model choice, the method we used here, ABC model choice with Random Forests (ABC-RF; *Marin and Pudlo, 2015*), is not affected by this fallacy, and can instead indicate a lack of power for model choice of low-divergence data sets between models with and without gene flow ('posterior probability' between 0.25 and 0.75). This is because ABC-RF estimates the probability to choose the correct model specifically for the observed data, rather than the model's relative probabilities of obtaining the observed data. We believe that ambiguity (lack of power) is preferable over clear but positively misleading rejection-based ABC model choice outcomes.

