## [Editor Report]

This work has the potential to be of broad interest to scientists seeking to understand the evolutionary dynamics of plants during past periods of rapid climate change. Specifically, within the target genus of *Cochlearia*, the results indicate increased rates of speciation and diversification in response to pronounced glacial cycles. Future work to establish more direct mechanistic links between the results and conclusions will improve our understanding of adaptation and speciation.

---

## [Decision Letter]

**Decision letter after peer review:**

Thank you for submitting your article "Evolutionary footprints of a cold relic in a rapidly warming world" for consideration by *eLife*. Your article has been reviewed by 3 peer reviewers, including Daniel J Kliebenstein as Reviewing Editor and Reviewer #1, and the evaluation has been overseen by Patricia Wittkopp as the Senior Editor. The following individual involved in review of your submission has agreed to reveal their identity: John T Lovell (Reviewer #2).

Essential revisions:

1) The reviewers noted a number of places where the central claims seemed to be overly strong and needed tempering. Please work to clarify what is and is not fully supported by the data.

2) All three reviewers and several editors agreed that the writing makes it sometimes difficult to understand how the data supports the argument/claim being made. The audience is a very general audience and it will greatly help to make the link between data and claims as linear, simple and direct as possible. Essentially try to walk the general audience from data to claim so that the reader can understand the work as much as the authors.

*Reviewer #1 (Recommendations for the authors):*

A better commentary on the ability to use or not use metabolomics for genetic arguments should be included.

*Reviewer #2 (Recommendations for the authors):*

Below are specific comments regarding sections of the manuscript that I feel could use some more clarification etc.

[1] – Framing tests of adaptation –

The main take home is that *Cochlearia* diverged from more temperate relatives around the start of the Pleistocene, and given its' current poleward and higher elevation distribution, this is evidence that speciation coincided with and was possibly caused by, a changing climate. This, on its own is an interesting taxonomic and phylogeographic discovery, and I believe this result, and the phylogenetics supporting it, to be very strong. The authors then use the multiple lineages within *Cochlearia* as separate lines of evidence to speculate about parallel and adaptive evolution. In some cases, this would also offer strong evidence. However, here, where evolution is known to be highly reticulate, it is not clear that the species really do offer independent lines of evidence for adaptation. Further, how in a system like this can one infer 'parallel' evolution? Couldn't any observed conserved phenotypes (or distributions) be due to recent admixture? Indeed, in several cases, the authors present evidence showing strong and recent gene flow between the modestly diverged genetic clusters.

[2] – Nuclear vs. cp/mt genome phylogeny –

The bulk of the phylogenetic analyses are supported by single-locus chloroplast or mito genome alignments. Yet it appears that the topologies between the nuclear and cytoplasmic genomes differ significantly (SI Figure 11 vs. Figure 2). Given these data and the known reticulate evolution of this system, it is not clear to me how much one can learn from the cyto genomes. Maybe a more explicit reasoning for this would help because, it appears to me that focusing on a single maternally inherited locus could significantly bias the observations. For example, in lines 139-140, a polytomy is provided as evidence of rapid radiation. Couldn't this also be driven by hybridization/chloroplast capture across a diverse set of genetic backgrounds?

[3] – Metabolomics –

I had trouble understanding how these results supported the phylogeographic patterns. The hypothesis of parallel adaptation to cold is really not testable by surveying a single group that are all cold adapted and an outgroup that is not adapted. Perhaps if PICs or some other measure was used to control for phylogenetic relatedness and strong associations between climate and metabolic responses were observed independent of phylogeny. But even in this case, it’s hard to tell how this would be able to detect whether "early evolved tolerance facilitated" the widespread distribution in northern habitats. Overall, the authors do a reasonable job contextualizing these results as merely speculative; however, I am unsure that the metabolomics data supports a speculation of adaptive mechanisms. Maybe outlining more specific hypotheses would help here.

[4] – Figures –

Some thoughts to improve the visualization:

(1) Figures1, 2, and 3 show similar points, but three separate projections. The projection in Figure 2 is the easiest for me to look at, but either way, a single projection system across all maps might help.

(2) There are no legends on the figures and the different genetic groups are often described in the text by their colors in the figures. This made it difficult for me to follow. Perhaps simply adding a consistent legend on the figures would help, or giving the different groups informative names beyond their colors?

[5] – interpretation –

Overall, I felt that in many places the interpretation and speculation of what results mean in terms of adaptation went well outside of the scope of the analysis. I detail a few of these situations below, but this is not exhaustive:

(1) Line 374 and elsewhere … is the hypothesis here that smaller chromosome sizes in polyploids is due to non-functionalization and subsequent duplicate gene loss. Certainly possible, but far from certain. Smaller chromosomes could be due to a number of factors, most likely the activity of TEs.

(2) Line 360 … more chrs = more meiosis = more recombination = more adaptation. Possible, but far from causal. Plus, n = 7 is ancestral, so it’s more likely that a lower recombination landscape evolved from an ancestrally larger one.

*Reviewer #3 (Recommendations for the authors):*

Lines 35 and 36: It might be helpful to point to some key references that outline this idea (e.g. Aitken et al., 2008; Gienapp et al., 2008).

Aitken, S. N., Yeaman, S., Holliday, J. A., Wang, T., and Curtis‐McLane, S. (2008). Adaptation, migration or extirpation: climate change outcomes for tree populations. Evolutionary applications, 1(1), 95-111.

Gienapp, P., Teplitsky, C., Alho, J. S., Mills, J. A., and Merilä, J. (2008). Climate change and evolution: disentangling environmental and genetic responses. Molecular ecology, 17(1), 167-178.

Lines 60-64: The logic here is a bit hard to follow. In the first part, I understand invoking Exposito-Alonso et al., 2018 to discuss the historical sharing of drought survival alleles between the Mediterranean and Scandinavian drought-resistant genetic groups, but I think it would be beneficial to more explicitly connect the dots between these results and the set-up of the author's system that leads to the hypothesis of "long-lasting footprint of drought adaptation".

Lines 71-76: The content of this paragraph seems a little disjointed as the topic sets up a taxonomically focused paragraph followed by a conservation concern. I'm wondering if it might helpful to absorb some of these points into other paragraphs.

Line 218: It would be helpful to provide plots/tables of δ k (in the supplemental) that helped you make these choices.

Lines 235-239: It might worth discussing any concerns about how unequal sample sizes between the groups might account from some of the metrics indicating higher genetic diversity in EUR.

Lines 291-292: Given the preceding sentences, this feels like it comes up a little suddenly. Is the hypothesis here that the only gene flow between the diploids is facilitated through a polyploid?

Lines 330-332: These results refer to a panel of supplementary figures with a mix of significant/non-significant results. It might be useful to highlight a few key results in the main text that support this point.

Lines 337-339: It might be good to be a little careful with the wording when discussing DAPC results. I understand the sentiment of this point, but DAPC really isn't testing for effect sizes or a treatment result.

Line 339: Explicitly discussing why these results are surprising would be helpful for interpretation.

---

## [Author Response]

Essential revisions:1) The reviewers noted a number of places where the central claims seemed to be overly strong and needed tempering. Please work to clarify what is and is not fully supported by the data.2) All three reviewers and several editors agreed that the writing makes it sometimes difficult to understand how the data supports the argument/claim being made. The audience is a very general audience and it will greatly help to make the link between data and claims as linear, simple and direct as possible. Essentially try to walk the general audience from data to claim so that the reader can understand the work as much as the authors.

Following above (and below) given comments we revised the manuscript according to these two essential comments and details are found below.

Reviewer #1 (Recommendations for the authors):A better commentary on the ability to use or not use metabolomics for genetic arguments should be included.

This comment re‐phrases the comment with the public review and is followed, of course. We also cited now the excellent paper presented by Weiszmann et al., (2020) exemplying the potential power of metabolome analyses across biogeographic and continental‐scale gradients (here, even within one single species – namely *Arabidopsis thaliana*).

Reviewer #2 (Recommendations for the authors):Below are specific comments regarding sections of the manuscript that I feel could use some more clarification etc.[1] – Framing tests of adaptation –The main take home is that Cochlearia diverged from more temperate relatives around the start of the Pleistocene, and given its' current poleward and higher elevation distribution, this is evidence that speciation coincided with and was possibly caused by, a changing climate. This, on its own is an interesting taxonomic and phylogeographic discovery, and I believe this result, and the phylogenetics supporting it, to be very strong. The authors then use the multiple lineages within Cochlearia as separate lines of evidence to speculate about parallel and adaptive evolution. In some cases, this would also offer strong evidence. However, here, where evolution is known to be highly reticulate, it is not clear that the species really do offer independent lines of evidence for adaptation. Further, how in a system like this can one infer 'parallel' evolution? Couldn't any observed conserved phenotypes (or distributions) be due to recent admixture? Indeed, in several cases, the authors present evidence showing strong and recent gene flow between the modestly diverged genetic clusters.

Thank you for the thoughtful comments. We agree in general, and recognize challenges to strongly infer parallelism in the face of admixture. In this respect we think It is useful to note the variety of degrees of admixture in our study: many species (strongly defined by ecology/distribution and cytotype) here are actually geographically isolated entities and not subjected to “recent” admixture. In fact, for one taxon only a postglacial hybrid origin is shown (*C. bavarica*), and for other taxa such as *C. tatrae* (hexaploid) we work to show an ancient connection between diploid gene pools in the text – but no recent geneflow. Therefore, it is indeed meaningful that highly endemic diploid *C. excelsa* and hexaploid *C. tatrae* evolved in two different high mountain ranges “in parallel”. However, we do note that recent geneflow is mostly obvious along coastal lines involving salt adapted species (between hexaploid *C. danica*, tetraploid *C. officinals*, *C. anglica* – and eventually resulting in those numerous aneuploids), but this very recent geneflow is out of the scope of this study.

In this respect we absolutely do agree: it depends on species and *Cochlearia* lineages being selected how powerful the system is to study parallel and/or convergent evolution. We endeavor in our revision to reflect this more subtle but important point – and we fully agree that there we have often have to speculate about “parallel evolution”.

Please consider also Suppl. Figure 1 illustrating the close association of substrate adaptation and distribution, which is thereby presenting another summary of convergently evolving traits.

[2] – Nuclear vs. cp/mt genome phylogeny –The bulk of the phylogenetic analyses are supported by single-locus chloroplast or mito genome alignments. Yet it appears that the topologies between the nuclear and cytoplasmic genomes differ significantly (SI Figure 11 vs. Figure 2). Given these data and the known reticulate evolution of this system, it is not clear to me how much one can learn from the cyto genomes. Maybe a more explicit reasoning for this would help because, it appears to me that focusing on a single maternally inherited locus could significantly bias the observations. For example, in lines 139-140, a polytomy is provided as evidence of rapid radiation. Couldn't this also be driven by hybridization/chloroplast capture across a diverse set of genetic backgrounds?

The reviewer is right: here we provide two different perspectives. Indeed, the maternal (cytoplasmic) data are mostly used to document maternal and, thereby, also biogeographic patterns of major dispersal routes since these organelles are inherited maternally and “propagated via seeds” and its genomes do not recombine significantly.

We note that we believe the organization is explicit regarding the data: the chapter mentioned by the reviewer is in fact headed “Organeller phylogenies …” and in response to the reviewer we framed this section now more explicitly. However, yes, we do also fully agree that we should comment and compare to the genomic perspective here, too. We do also agree that we may not use terms such as “rapid radiation” referring to speciation processes and being more precise and have thus removed instances of ‘rapid’. We rephrased this and also explained limits of the cytonuclear as well as the nuclear phylogenies; e.g. lines 186‐190, 295‐299.

The reviewer mentioned SI Figure 11. This is indeed true and the same is true with TreeMix analysis (SI Figure 15, congruent to Figure 11). Unfortunately, here TreeMix is not able to demonstrate early geneflow (e.g. hybridization and allo‐polyploidization), which has been already documented for some polyploids (e.g. in *C. bavarica*). This is why we largely refer to structure analysis recognizing significantly three gene pools in *Cochlearia* with varying degrees of admixture; and also neighbor‐net analysis (Figure 3) provides compelling evidence for “polyploid intermediacy”. Therefore, trees from the nuclear genome – as the reviewer correctly noted – do not show the reticulate patterns and may be even more misleading than cytonuclear trees. We phrased this accordingly to provide the reader with this information.

Please note: with additional SI Figures numbering changed with the revised version.

[3] – Metabolomics –I had trouble understanding how these results supported the phylogeographic patterns. The hypothesis of parallel adaptation to cold is really not testable by surveying a single group that are all cold adapted and an outgroup that is not adapted. Perhaps if PICs or some other measure was used to control for phylogenetic relatedness and strong associations between climate and metabolic responses were observed independent of phylogeny. But even in this case, its hard to tell how this would be able to detect whether "early evolved tolerance facilitated" the widespread distribution in northern habitats. Overall, the authors do a reasonable job contextualizing these results as merely speculative; however, I am unsure that the metabolomics data supports a speculation of adaptive mechanisms. Maybe outlining more specific hypotheses would help here.

Thank you for the comments: we modify the text accordingly and also suggest the reviewer note the interesting comments of reviewer 1, and our reply in considering our revision. In general, we agree that aspects of parallel evolution and adaptation processes are not to be tested explicitly with our metabolomic data.

[4] – Figures –Some thoughts to improve the visualization:(1) Figures1, 2, and 3 show similar points, but three separate projections. The projection in Figure 2 is the easiest for me to look at, but either way, a single projection system across all maps might help.

The comment is appreciated and we re‐draw figures using projection originally shown with Figure 2.

(2) There are no legends on the figures and the different genetic groups are often described in the text by their colors in the figures. This made it difficult for me to follow. Perhaps simply adding a consistent legend on the figures would help, or giving the different groups informative names beyond their colors?

This is in particular true for Figure 2. The only way to define plastids clades is to provide a code (I – VI plus a wider circumscription) and thereby avoiding referring to colours. However, in this case, it is difficult to get the information about geographical distribution as shown with the respective map in Figure 2. But we improved this aspect and whenever meaningful, some explanatory words within the text are given using a consistent terminology describing the distribution range.

[5] – Interpretation –Overall, I felt that in many places the interpretation and speculation of what results mean in terms of adaptation went well outside of the scope of the analysis. I detail a few of these situations below, but this is not exhaustive:

We appreciate the reviewer’s comments and work in the revision to better reflect where we speculate, for example in the discussion and conclusion.

(1) Line 374 and elsewhere … is the hypothesis here that smaller chromosome sizes in polyploids is due to non-functionalization and subsequent duplicate gene loss. Certainly possible, but far from certain. Smaller chromosomes could be due to a number of factors, most likely the activity of TEs.

We try not to speculate on the mechanism of chromosome size reduction beyond the observation of a highly fluid dynamic in this genus and the clear evidence of ‘polyploid drop. Indeed, many factors may be responsible, we agree; and we think it best to demur from more speculation, so we have reduced this.

(2) Line 360 … more chrs = more meiosis = more recombination = more adaptation. Possible, but far from causal. Plus, n = 7 is ancestral, so its more likely that a lower recombination landscape evolved from an ancestrally larger one.

This is true. We have no causal link or experimental evidence for this. We have thus rephrased this and added the respective limitation.

Reviewer #3 (Recommendations for the authors):Lines 35 and 36: It might be helpful to point to some key references that outline this idea (e.g. Aitken et al., 2008; Gienapp et al., 2008).Aitken, S. N., Yeaman, S., Holliday, J. A., Wang, T., and Curtis‐McLane, S. (2008). Adaptation, migration or extirpation: climate change outcomes for tree populations. Evolutionary applications, 1(1), 95-111.Gienapp, P., Teplitsky, C., Alho, J. S., Mills, J. A., and Merilä, J. (2008). Climate change and evolution: disentangling environmental and genetic responses. Molecular ecology, 17(1), 167-178.

We fully agree and are thankful for these suggestions.

Lines 60-64: The logic here is a bit hard to follow. In the first part, I understand invoking Exposito-Alonso et al., 2018 to discuss the historical sharing of drought survival alleles between the Mediterranean and Scandinavian drought-resistant genetic groups, but I think it would be beneficial to more explicitly connect the dots between these results and the set-up of the author’s system that leads to the hypothesis of “long-lasting footprint of drought adaptation”.

Yes, that was expressed a little bit vaguely; we have edited to clarify our point.

Lines 71-76: The content of this paragraph seems a little disjointed as the topic sets up a taxonomically focused paragraph followed by a conservation concern. I’m wondering if it might helpful to absorb some of these points into other paragraphs.

Definitely, thank you for that observation: that small taxonomic paragraph is somehow an orphan and we have found it a home with the discussion just above discussing chromosome number, using the conservation concern to link to the discussion of abiotic factors. In the introduction text the conservation concern was included to indicate that population are declining because of rapidly changing environment. Indeed, we are not referring to conservation issues with this contribution.

Line 218: It would be helpful to provide plots/tables of δ k (in the supplemental) that helped you make these choices.

Yes, this is right, and we included them with the respective scores with the suppl. Material.

Lines 235-239: It might worth discussing any concerns about how unequal sample sizes between the groups might account from some of the metrics indicating higher genetic diversity in EUR.

Based on early population‐based analyses using isozymes we found a very similar pattern in earlier studies (Koch et al., 1998). Of course, “genetic” markers are not comparable, but since we herein cover the entire distribution range of arctic taxa (also spanning different species) we do not expect a major bias caused by our sampling. But the reviewer is right, that we should mention it here.

Lines 291-292: Given the preceding sentences, this feels like it comes up a little suddenly. Is the hypothesis here that the only gene flow between the diploids is facilitated through a polyploid?

This is clarified. Our data indicate that polyploids such as *C. tatrae* show a genomic footprint of the 2n=14 gene pool. Consequently, this limits the maximum age of this taxon accordingly.

Lines 330-332: These results refer to a panel of supplementary figures with a mix of significant/non-significant results. It might be useful to highlight a few key results in the main text that support this point.

We agree and added a respective paragraph.

Lines 337-339: It might be good to be a little careful with the wording when discussing DAPC results. I understand the sentiment of this point, but DAPC really isn't testing for effect sizes or a treatment result.

We fully agree and tempered conclusions and deleted the term “strong”.

Line 339: Explicitly discussing why these results are surprising would be helpful for interpretation.

We also fully agree here, and also link in Dan Kliebenstein´s very interesting idea that pleiotropy in metabolic response networks may constrain selection, eventually indicating a potential mechanism of the (useful?) stasis in these networks allowing (parallel?) adaptation.